# Cryo-EM structure of the transposon-associated TnpB enzyme

Ryoya Nakagawa[1,11], Hisato Hirano[1,11], Satoshi N. Omura[1], Suchita Nety[2,3,4,5,6], Soumya Kannan[2,3,4,5,6], Han Altae-Tran[2,3,4,5,6], Xiao Yao[7], Yuriko Sakaguchi[7], Takayuki Ohira[7], Wen Y. Wu[8], Hiroshi Nakayama[9], Yutaro Shuto[1], Tatsuki Tanaka[1], Fumiya K. Sano[1], Tsukasa Kusakizako[1], Yoshiaki Kise[1,10], Yuzuru Itoh[1], Naoshi Dohmae[9], John van der Oost[8], Tsutomu Suzuki[7], Feng Zhang[2,3,4,5,6] & Osamu Nureki[1,10 ✉]

The class 2 type V CRISPR effector Cas12 is thought to have evolved from the IS200/IS605 superfamily of transposon-associated TnpB proteins[1]. Recent studies have identified TnpB proteins as miniature RNA-guided DNA endonucleases[2,3]. TnpB associates with a single, long RNA (ωRNA) and cleaves double-stranded DNA targets complementary to the ωRNA guide. However, the RNA-guided DNA cleavage mechanism of TnpB and its evolutionary relationship with Cas12 enzymes remain unknown. Here we report the cryo-electron microscopy (cryo-EM) structure of *Deinococcus radiodurans* ISDra2 TnpB in complex with its cognate ωRNA and target DNA. In the structure, the ωRNA adopts an unexpected architecture and forms a pseudoknot, which is conserved among all guide RNAs of Cas12 enzymes. Furthermore, the structure, along with our functional analysis, reveals how the compact TnpB recognizes the ωRNA and cleaves target DNA complementary to the guide. A structural comparison of TnpB with Cas12 enzymes suggests that CRISPR–Cas12 effectors acquired an ability to recognize the protospacer-adjacent motif-distal end of the guide RNA–target DNA heteroduplex, by either asymmetric dimer formation or diverse REC2 insertions, enabling engagement in CRISPR–Cas adaptive immunity. Collectively, our findings provide mechanistic insights into TnpB function and advance our understanding of the evolution from transposon-encoded TnpB proteins to CRISPR–Cas12 effectors.

CRISPR–Cas systems in prokaryotes provide adaptive immunity against foreign nucleic acids, and are divided into two classes (classes 1 and 2) and six types[1,4] (types I–VI). The class 2 systems include types II, V and VI, in which Cas9, Cas12 and Cas13, respectively, function as effector enzymes responsible for interference. Type II Cas9 effector proteins associate with dual RNA guides (CRISPR RNA (crRNA) and *trans*-activating crRNA (tracrRNA) or their artificially connected single-guide RNA (sgRNA)) and cleave double-stranded DNA (dsDNA) targets using the HNH and RuvC nuclease domains[5,6]. Type V Cas12 effector proteins are further divided into Cas12a–m subtypes and span diverse functionalities[7–15]. Although the Cas12 proteins commonly have a single RuvC nuclease domain, they share low sequence similarity outside this conserved region. Cas12 enzymes associate with either crRNA guides or dual RNA guides (crRNA and tracrRNA) to cleave dsDNA targets, using their single RuvC domain. Recent reports have demonstrated that the class 2 CRISPR effectors Cas9 and Cas12 evolved independently from two members of the IS200/IS605 transposon-encoded nuclease superfamily, IscB and TnpB, respectively[1–3]. Functional and

structural studies of IscB revealed that an associated ωRNA (obligate mobile element guided activity (OMEGA)) has a crucial role in recognizing the guide RNA–target DNA heteroduplex, enabling IscB to cleave its target DNA using the HNH and RuvC domains[16,17].

TnpB proteins are also RNA-guided DNA endonucleases[2,3]. TnpB associates with a single, long non-coding RNA (referred to as ωRNA, also known as right end element RNA (reRNA)). The gene encoding the ωRNA overlaps with the 3′ end of the *tnpB* gene and the non-coding right end element of the transposon (Fig. 1a), and TnpB cleaves dsDNA targets complementary to the ωRNA guide sequence by using its RuvC domain. In addition, TnpB requires a target-adjacent motif (TAM) upstream of the target sequence to cleave the target DNA, similar to the Cas enzymes, which require a protospacer-adjacent motif (PAM) sequence to cleave their target DNAs[18]. TnpB from *D. radiodurans* ISDra2 (hereafter referred to as TnpB for simplicity) consists of 408 residues (Extended Data Fig. 1) and has a similar domain organization to Cas12f, which is the smallest subtype among the type V Cas12 enzymes[11]. However, a recent molecular mass analysis revealed that TnpB functions

[1]Department of Biological Sciences, Graduate School of Science, The University of Tokyo, Tokyo, Japan. [2]Broad Institute of MIT and Harvard, Cambridge, MA, USA. [3]McGovern Institute for Brain Research at MIT, Massachusetts Institute of Technology, Cambridge, MA, USA. [4]Department of Biological Engineering, Massachusetts Institute of Technology, Cambridge, MA, USA. [5]Department of Brain and Cognitive Science, Massachusetts Institute of Technology, Cambridge, MA, USA. [6]Howard Hughes Medical Institute, Cambridge, MA, USA. [7]Department of Chemistry and Biotechnology, Graduate School of Engineering, The University of Tokyo, Tokyo, Japan. [8]Laboratory of Microbiology, Wageningen University and Research, Wageningen, The Netherlands. [9]Biomolecular Characterization Unit, RIKEN Center for Sustainable Resource Science, Saitama, Japan. [10]Curreio, The University of Tokyo, Tokyo, Japan. [11]These authors contributed equally: Ryoya Nakagawa, Hisato Hirano. ✉e-mail: nureki@bs.s.u-tokyo.ac.jp

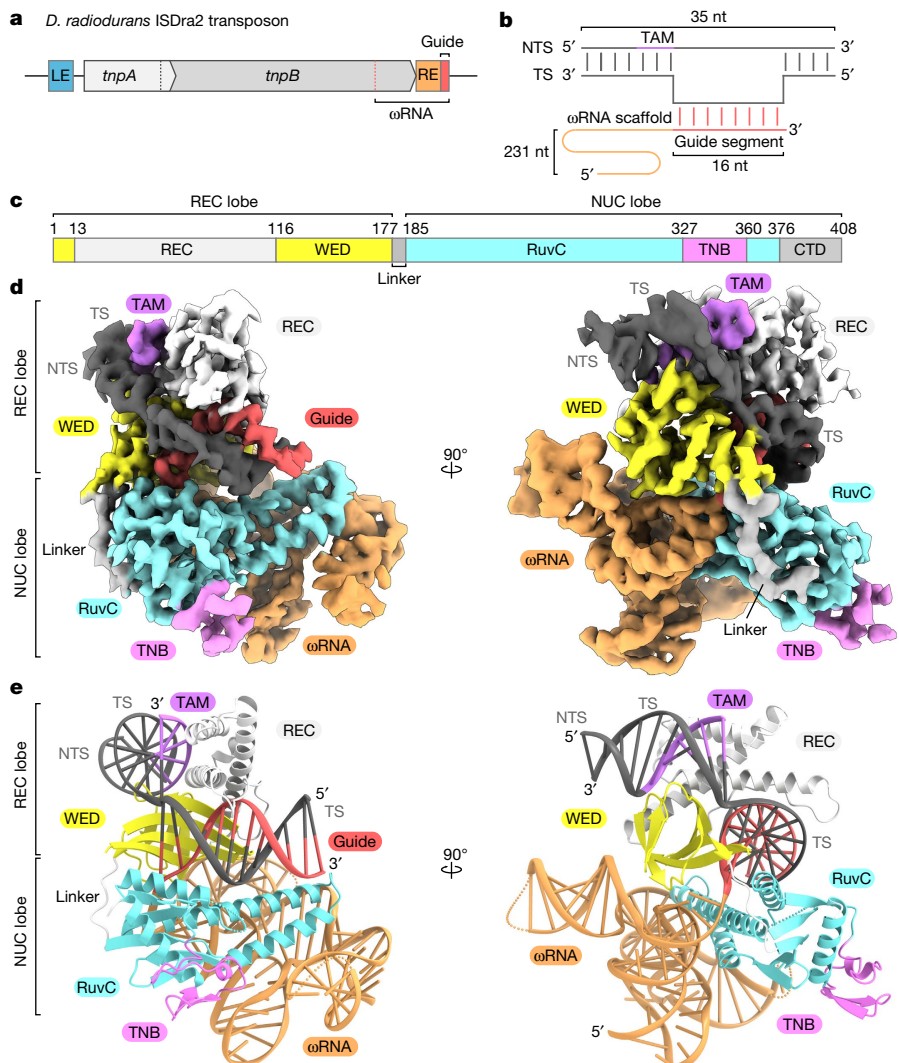

**Fig. 1 | Cryo-EM structure of the TnpB–ωRNA–target DNA ternary complex.**
**a**, Schematic of the *D. radiodurans* ISDra2 locus. The MGE consists of the *tnpA*
and *tnpB* genes flanked by the left end (LE) and right end (RE) elements of the
transposon. The ωRNA is derived from the 3′ end of the *tnpB* gene and the RE
element. **b**, Diagram of the ωRNA and target DNA used for cryo-EM analysis.
The target strand (TS) and non-target strand (NTS) each comprise 35 nucleotides,
and the non-target strand contains a TTGAT TAM sequence. The 247-nt ωRNA
was co-expressed and co-purified with TnpB. Nucleotides −231 to −117, −70 to

−49, −20 to −17 and 13 to 16 of the ωRNA, nucleotides −8 to 4 and 27 of the target
strand, and nucleotides −11* and 1* to 24* of the non-target strand were not
included in the final model. **c**, The domain structure of TnpB. CTD, C-terminal
domain. Residues 281 to 296 and 379 to 408 were not included in the final
model. **d**, Cryo-EM density map of the TnpB–ωRNA–target DNA complex.
**e**, The overall structure of the TnpB–ωRNA–target DNA complex. Disordered
regions are indicated as dotted lines.

as a monomer, whereas Cas12f functions as a dimer[3,19]. Moreover,
whereas Cas12f associates with dual RNA guides, TnpB associates with
the single ωRNA. Therefore, how the compact TnpB protein assembles
with its cognate ωRNAs to mediate the RNA-guided double-stranded
DNA breaks remains unknown.

## Structure of TnpB–ωRNA–target DNA

To elucidate the molecular mechanism of TnpB, we co-expressed TnpB
and its cognate 247-nucleotide (nt) ωRNA containing a 16-nt guide
segment at the 3′ end, and then purified the TnpB–ωRNA complex.
We reconstituted the ternary TnpB–ωRNA–target DNA complex by
mixing the purified TnpB–ωRNA complex and a 35-base pair (bp)
double-stranded DNA with phosphorothioate modifications within the
DNA backbone around the cleavage site and the TTGAT TAM sequence,
and analysed its ternary structure by cryo-EM (Fig. 1b). We obtained

a three-dimensional reconstruction of the ternary complex with an
overall resolution of 3.2 Å (Fig. 1c–e, Extended Data Fig. 2, Extended
Data Table 1 and Supplementary Fig. 1). The cryo-EM structure revealed
that a single TnpB molecule assembles with a single ωRNA molecule
to form a ribonucleoprotein effector complex, consistent with
the previous molecular mass analysis[3]. TnpB adopts a bilobed architec-
ture consisting of recognition (REC) and nuclease (NUC) lobes, which
are connected by a linker loop (Fig. 1c–e). The REC lobe comprises the
wedge (WED) and REC domains, and the NUC lobe consists of the RuvC
nuclease domain and the target nucleic acid-binding (TNB) domain.
The C-terminal domain (residues 376 to 408), which has little sequence
homology among TnpB proteins, is disordered in the present structure
except for its first three residues (Extended Data Fig. 1). The ωRNA–
target DNA heteroduplex is accommodated within a central channel
formed by the WED, REC and RuvC domains (Fig. 1d,e and Extended Data
Fig. 3a). The TAM-containing DNA duplex (the TAM duplex) is bound to

the cleft formed by the WED and REC domains, whereas the ωRNA scaffold binds to a surface formed by the WED and RuvC domains (Extended Data Fig. 3b–e). In the interior of the complex, the amino acids of TnpB and the nucleotides of the ωRNA and the target DNA are clearly visible in the density map. By contrast, we did not observe clear densities for the peripheral regions, such as the C-terminal domain (residues 379 to 408), ωRNA scaffold (nucleotides −70 to −49) and the TAM-distal DNA duplex (re-hybridized duplex), indicating the flexibility of these regions. Thus, residues 281 to 296 and 379 to 408 of TnpB, nucleotides −231 to −117, −70 to −49, −20 to −17 and 13–16 of the ωRNA, nucleotides −8 to 4 and 27 of the target strand, and nucleotides −11* and 1*–24* of the non-target strand were not included in the final model.

## Domain structure

The WED domain (residues 1 to 12 and 117 to 176) comprises a seven-stranded β-barrel flanked by an α-helix and adopts an oligonucleotide/oligosaccharide-binding fold (Extended Data Fig. 4a). The REC domain (residues 13 to 116) is inserted between the β1 and β2 strands of the WED domain and is composed of four α-helices. The RuvC domain (residues 185 to 326 and 360 to 375) has an RNase H fold, consisting of a five-stranded mixed β-sheet flanked by four α-helices. The RuvC domain of TnpB structurally resembles those of Cas12 enzymes, and the conserved D191, E278 and D361 residues form a catalytic centre similar to those of other Cas12 enzymes[19–21] (Extended Data Fig. 4a). The TNB domain (residues 327 to 359) is inserted between the β5 strand and the α4 helix and contains a CCCC-type zinc-finger, in which a zinc ion is coordinated by four conserved cysteine residues (C331, C334, C351 and C354) (Extended Data Fig. 4a,b). A structural comparison of TnpB with Cas12 enzymes revealed that TnpB represents the minimal domain organization common to all Cas12 enzymes[19–21] (Extended Data Fig. 4a).

## Characterization of the ωRNA

In the present structure, we observed a density for the 3′ end of the ωRNA (−116G to 16C), except for the peripheral region, whereas we could not detect a density for the 5′ end (−231G to −117T) (Fig. 2a,b and Extended Data Fig. 3c). Consistent with this observation, a denaturing gel analysis revealed that the purified TnpB was bound to 100–160 nt of the ωRNA (Extended Data Fig. 5a), even though it was co-expressed with a 247-nt ωRNA. To understand which part of the ωRNA sequence remains bound to the TnpB protein, we performed northern blotting analyses using five DNA probes (I–V) covering the entire ωRNA sequence (Extended Data Fig. 5b,c and Supplementary Fig. 2). In cells expressing ωRNA only, the total RNA did not yield any ωRNA band, strongly suggesting that the ωRNA is degraded by endogenous RNases in the absence of the TnpB protein (Extended Data Fig. 5c). By contrast, in cells co-expressing ωRNA and TnpB, three different RNAs, approximately 220, 160 and 130 nt in size, were found in the total RNA extract (Extended Data Fig. 5c). The approximately 220-nt RNA was detected by probes II–V and the approximately 160 and 130-nt RNAs were detected by probes III–V, respectively, indicating that these RNAs commonly contain the guide sequences on the 3′ end. The approximately 160 and 130-nt RNAs were also observed in the RNA extracted from the purified TnpB–ωRNA complex, whereas the approximately 220-nt RNA was barely observed, suggesting that the 220-nt RNA was degraded during the purification process or urea gel electrophoresis (Extended Data Fig. 5c). Next, we isolated the approximately 160-nt and 130-nt RNAs and performed liquid chromatography-mass spectrometry (LC–MS) analyses to determine the processing sites of these RNAs. The approximately 160-nt RNA treated with RNase A had a pGAACp fragment (Extended Data Fig. 5d), suggesting that this RNA was cleaved between −150A and −149G or −138U and −137G by TnpB and/or endogenous RNases. We could not identify the cleavage site of the approximately 130-nt RNA, since the region around −110 to −120 is difficult to analyse owing to its

GU-rich sequence. However, we did observe a clear density for −116G, but not −117T, suggesting that the approximately 130-nt RNA is cleaved between −117T and −116G. We concluded that the ωRNA is processed by TnpB or endogenous RNases at multiple sites at the 5′ end, and that at least the 130-nt fragment including the guide at the 3′ end of the ωRNA remains stably bound to the TnpB protein.

## Co-evolution of the *tnpB* gene and ωRNA

Previous studies revealed that many archaea and bacteria transcribe non-coding RNAs (ncRNAs) overlapping the 3′ end of the *tnpB* gene, suggesting that these overlapping ncRNAs have conserved roles in prokaryotes[2,22]. Indeed, the ωRNA of ISDra2 TnpB overlaps the 3′ end of the *tnpB* gene (residues 335 to 408 and −231G to −10U) (Extended Data Fig. 6a). However, our structural and biochemical analyses showed that the ωRNA bound to the TnpB was processed on the 5′ side, as described above. Indeed, the truncation of the 5′ region of the ωRNA (Δ−231G to −117T (Δ5′ region)) had no effect on the TnpB-mediated DNA cleavage, indicating that this region of the ωRNA (−231G to −117T) is not required for cleavage (Extended Data Fig. 6b,c). Conversely, the C-terminal domain (residues 376 to 408) has relatively low sequence homology among TnpB proteins (Extended Data Fig. 1), and is disordered in the present structure. Our in vitro cleavage assay revealed that a C-terminal truncation mutant (Δ376 to 408 (ΔCTD)) efficiently cleaves the target DNA, although the protein stability is slightly decreased compared with the wild-type TnpB (Extended Data Fig. 6b–d). This result indicated that the C-terminal domain is not required for the RNA-guided target DNA cleavage among TnpB proteins. Thus, our structure revealed that the TnpB C-terminal region (residues 376 to 408 overlapping with −109G to −10U) is disordered and not involved in the DNA cleavage, whereas the 5′ region of the ωRNA (−231G to −117T, overlapping with residues 336 to 373) is not crucial for the target DNA cleavage. Therefore, except for a few nucleotides, the functionally important regions of the *tnpB* gene and ωRNA do not overlap, suggesting that although ωRNA expression and processing may require co-expression with the TnpB protein, the co-evolution of these two elements is less constrained than previously predicted and avoids the overlap of functionally essential gene regions (Extended Data Fig. 6a).

## ωRNA architecture

The present structure demonstrated that the TnpB–ωRNA scaffold adopts an unexpected architecture compared with that expected from the primary sequence[2] (Fig. 2a,b). The ωRNA (−116G to 16C) consists of the 16-nt guide segment and 116-nt RNA scaffold, comprising four stems (stems 1–4) and a pseudoknot (PK). Notably, −5U to −3C base pair with −103A to −105G, rather than with the predicted −30A to −32G, and −6G and −2A form non-canonical base pairs with −102U and −106G, respectively, to construct the PK (Fig. 2a,b and Extended Data Fig. 3c). The PK coaxially stacks with stem 1 to form a continuous helix. Nucleotides −35U to −32G base pair with −81A to −84C, and −36G and −31A form non-canonical base pairs with −80U and −85G, respectively, to form stem 2 (Fig. 2a,b and Extended Data Fig. 3d). In addition, −91U to −86C base pair with −111C to −116G in stem 1, thereby contributing to the triple helix formation (Fig. 2a,b and Extended Data Fig. 3e). As expected from the nucleotide sequence, stem 3a contains a 7-bp duplex—pairing −77A to −71U with −42U to −48A—with a loop, whereas stem 3b (−70G to −49A) is unresolved, suggesting the intrinsic flexibility of this region (Fig. 2a,b and Extended Data Fig. 3d).

Previous studies demonstrated that a truncation of the disordered region of the Cas12f sgRNA improved genome editing efficiency in mammalian cells[23,24]. To test this idea in TnpB, we constructed an ωRNA truncated mutant, in which the 5′ end of the ωRNA (nucleotides −231G to −117T) was deleted (referred to as Trim1). The Trim1 mutant induced indels at efficiencies similar to or higher than those of the full-length

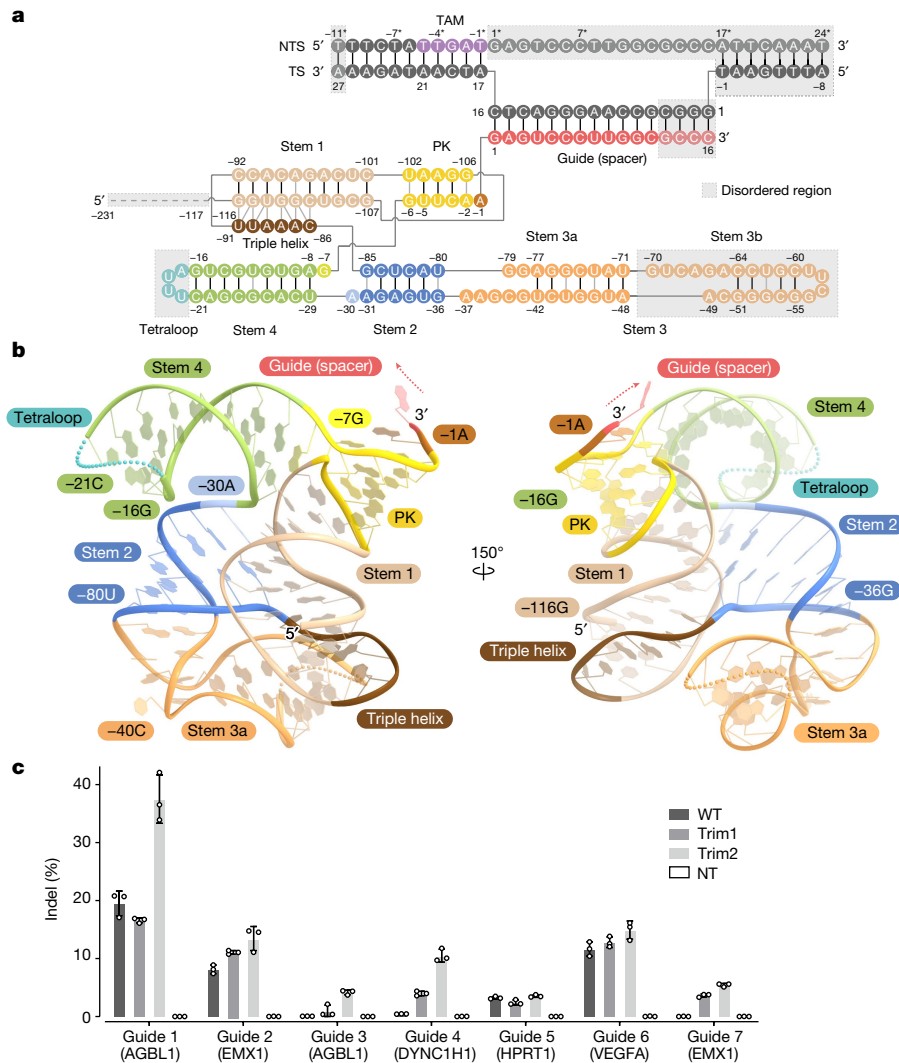

**Fig. 2 | ωRNA architecture. a**, Schematic of the ωRNA and target DNA. Nucleotides −231 to −117, −70 to −49, −20 to −17, and 13 to 16 of the ωRNA, nucleotides −8 to 4 and 27 of the target strand, and nucleotides −11* and 1* to 24* of the non-target strand were disordered and not included in the model. These disordered regions are enclosed in grey boxes. **b**, Structure of the ωRNA scaffold. The disordered regions are indicated as dotted lines. **c**, Insertion–deletion mutation (indel) formation efficiencies of TnpB with wild-type (WT) ωRNA, ωRNA with deleted 5′ region (Trim1), ωRNA lacking both 5′ region and stem 3b (Trim2) and non-targeting ωRNA (NT) at seven endogenous target sites in HEK293FT cells. Data are mean ± s.d. (*n* = 3 biologically independent samples).

ωRNA (Fig. 2c). An additionally truncated mutant, in which stem 3b (−70G to −49A) was deleted by connecting −71U and −48A with a GAAA linker (referred to as Trim2), exhibited further enhanced genome editing activity (Fig. 2c). These results confirmed the utility of TnpB in combination with the Trim2 ωRNA as a compact genome-engineering tool.

A structural comparison of the ωRNA with the guide RNAs of Cas12 enzymes revealed the presence of a structurally conserved core region[19,20] (Extended Data Fig. 7a). The guide RNAs of Cas12 enzymes commonly have PK structures preceding the guide sequences, despite their distinct architectures, whereas the ωRNA of TnpB forms the conserved PK structure. These core regions are recognized by their cognate TnpB and Cas12 proteins in similar manners (Extended Data Fig. 7b) (described below). A structural comparison of the ωRNA with the sgRNA of Cas12f also revealed similarities between them. The guide, PK and stem 4 regions of the ωRNA structurally resemble the guide, PK (repeat–antirepeat 1) and stem 5 (repeat–antirepeat 2) regions of the Cas12f sgRNA (Extended Data Fig. 7a). Furthermore, other regions of the ωRNA form similar stem loops to those of the Cas12f sgRNA and interact with the TnpB protein. Therefore, our structure revealed

that the ωRNA functions as a 'natural' sgRNA, in which a crRNA-like region and a tracrRNA-like region are connected by a UUUA tetraloop (−20U to −17A) (Extended Data Fig. 7c).

## ωRNA recognition

TnpB recognizes the ωRNA via its WED and RuvC domains, mainly through interactions with its sugar-phosphate backbone (Fig. 3a and Extended Data Fig. 8a). The stem 1 and triple helix structure of the ωRNA are recognized by the α1 and α2 helices of the RuvC domain through electrostatic interactions between its sugar-phosphate backbone and highly conserved basic residues (Fig. 3b). Stems 2 and 3a extensively interact with the RuvC domain (Fig. 3c and Extended Data Fig. 8a). Nucleotides −36G, −73U and −74C form hydrogen-bonding interactions with R232, Q227 and R231 respectively, and the nucleobase of −35U forms a stacking interaction with R238 (Fig. 3c). Stem 4 is recognized by the WED domain primarily through electrostatic interactions between its upper stem region and strands β4 and β5 of the WED domain (Fig. 3d). The conserved PK region of the ωRNA is sandwiched

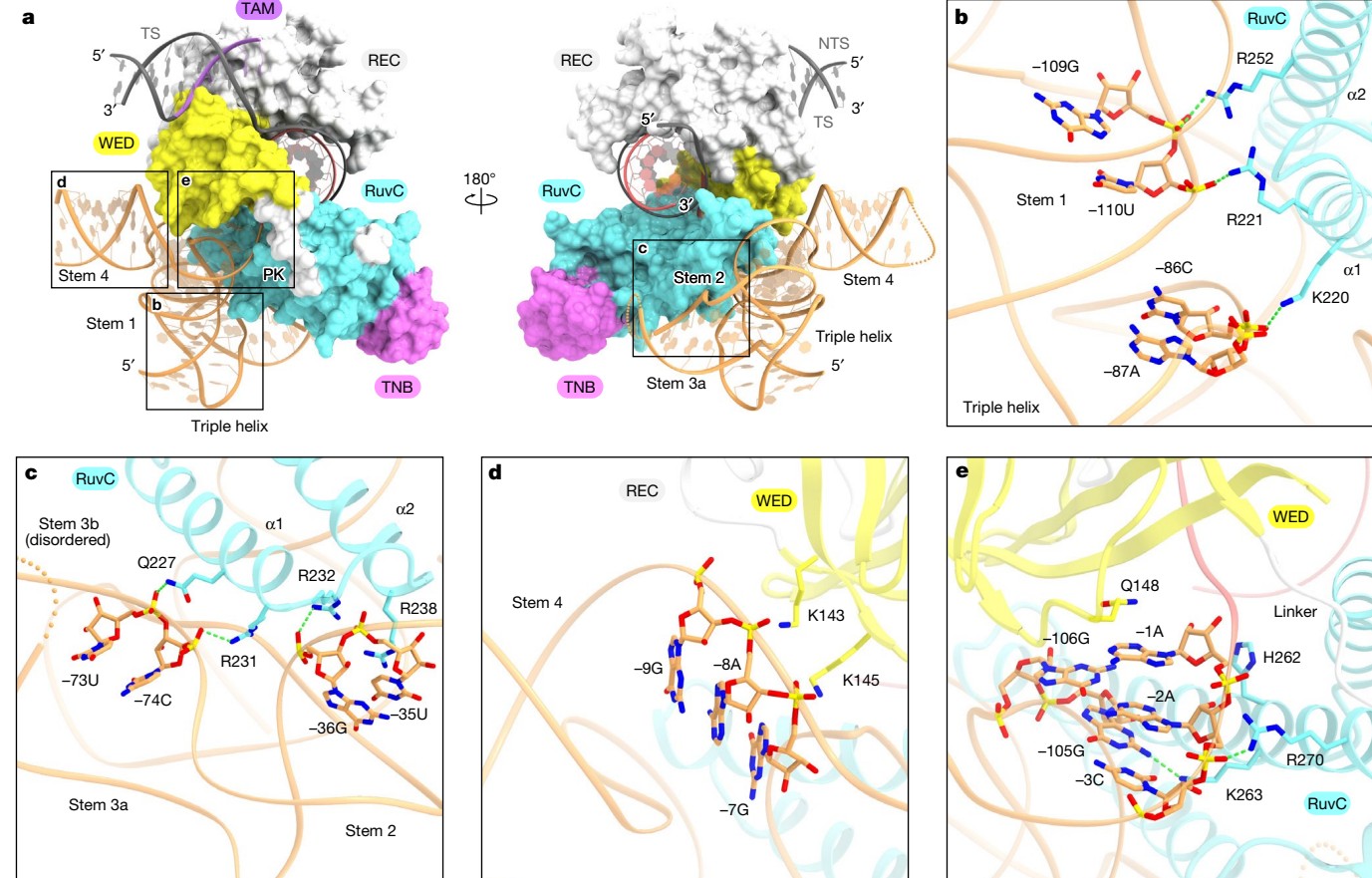

**Fig. 3 | ωRNA recognition. a**, Recognition of the ωRNA scaffold by the TnpB protein. TnpB is shown as a surface model. The ωRNA scaffold is recognized mainly through the WED and RuvC domains. **b–e**, Recognition of the stem 1 and triple helix structure (**b**), stems 2 and 3a (**c**), stem 4 (**d**) and the PK architecture (**e**) of the ωRNA scaffold. Hydrogen bonding and electrostatic interactions are shown as dashed lines.

between and extensively recognized by the WED and RuvC domains (Fig. 3e and Extended Data Fig. 7b). The −105G:−3C base pair in the PK is recognized by K263 through a hydrogen bond. The nucleobase of −1A between the PK and the guide segment is sandwiched by Q148 and the −106G:−2A non-canonical base pair, and the ribose moiety of −1A forms a hydrophobic interaction with H262 (Fig. 3e). Notably, Cas12 enzymes recognize the conserved PK structure in their cognate guide RNAs by the WED and RuvC domains in a similar manner[19,20] (Extended Data Fig. 7b), consistent with the notion that the core regions of the ωRNA and the guide RNAs are highly conserved among the TnpB and Cas12 enzymes and are important for the target DNA cleavage.

## TAM recognition

The TTGAT TAM duplex is recognized by the WED and REC domains (Fig. 4a and Extended Data Figs. 3b and 8a). The nucleobase of −1*dT in the non-target strand forms van der Waals interactions with Y52 and S56 in the REC domain (Fig. 4b). The nucleobase of −2*dA in the non-target strand forms a hydrogen bond with Q80 in the REC domain, and the methyl group of dT18 in the target strand forms van der Waals interactions with F77 in the REC domain (Fig. 4b). O6 and N7 of −3*dG in the non-target strand form a bidentate hydrogen bond with the side chain of K76 in the REC domain, which is anchored via a stacking interaction with F77 (Fig. 4c). The 5′-methyl group of −4*dT forms van der Waals interactions with the side chains of F77 in the REC domain and T123 in the WED domain, whereas the nucleobase of −5*dT hydrogen bonds with N124 in the WED domain (Fig. 4c). The Y52A, K76A, Q80A and T123A mutations abolished the cleavage activity of TnpB, whereas the S56A, F77A

and N124A mutations substantially reduced the DNA cleavage activity of TnpB, confirming the functional importance of these residues for the TAM recognition (Fig. 4d). Together, our structural and functional analyses revealed that TnpB forms sequence-specific contacts with both target and non-target strands to achieve TAM recognition.

## Target DNA recognition

The guide RNA–target DNA heteroduplex is accommodated within a positively charged central channel formed by the REC and RuvC domains and is recognized through interactions with its sugar-phosphate backbone (Fig. 4a and Extended Data Figs. 3a and 8a). The backbone phosphate group between dA17 and dC16 in the target strand is recognized by Q121 and N156 in the WED domain, and the first 1G:16dC base pair stacks with N4 and Y309/W313 in the WED and RuvC domains, respectively (Fig. 4e), as also observed in Cas12 enzymes[20]. These interactions facilitate the target DNA unwinding and the guide RNA–target DNA heteroduplex formation. The displaced single-stranded non-target strand in the target dsDNA is barely visible in the present structure, owing to its flexibility. In the guide–target heteroduplex, (1G to 11G):(16dC to 6dC) of the TAM-proximal region are accommodated within the positively charged central channel and recognized by TnpB through electrostatic interactions with its sugar-phosphate backbone (Extended Data Fig. 8a). By contrast, the TAM-distal region in the heteroduplex (12C:3dG) is exposed to the solvent, and the four terminal base pairs (13G to 16C):(4dC to 1dG) are disordered (Fig. 4f). These structural observations suggested that the base pairs in the TAM-distal region are not recognized by TnpB. Indeed, our in vitro cleavage assays

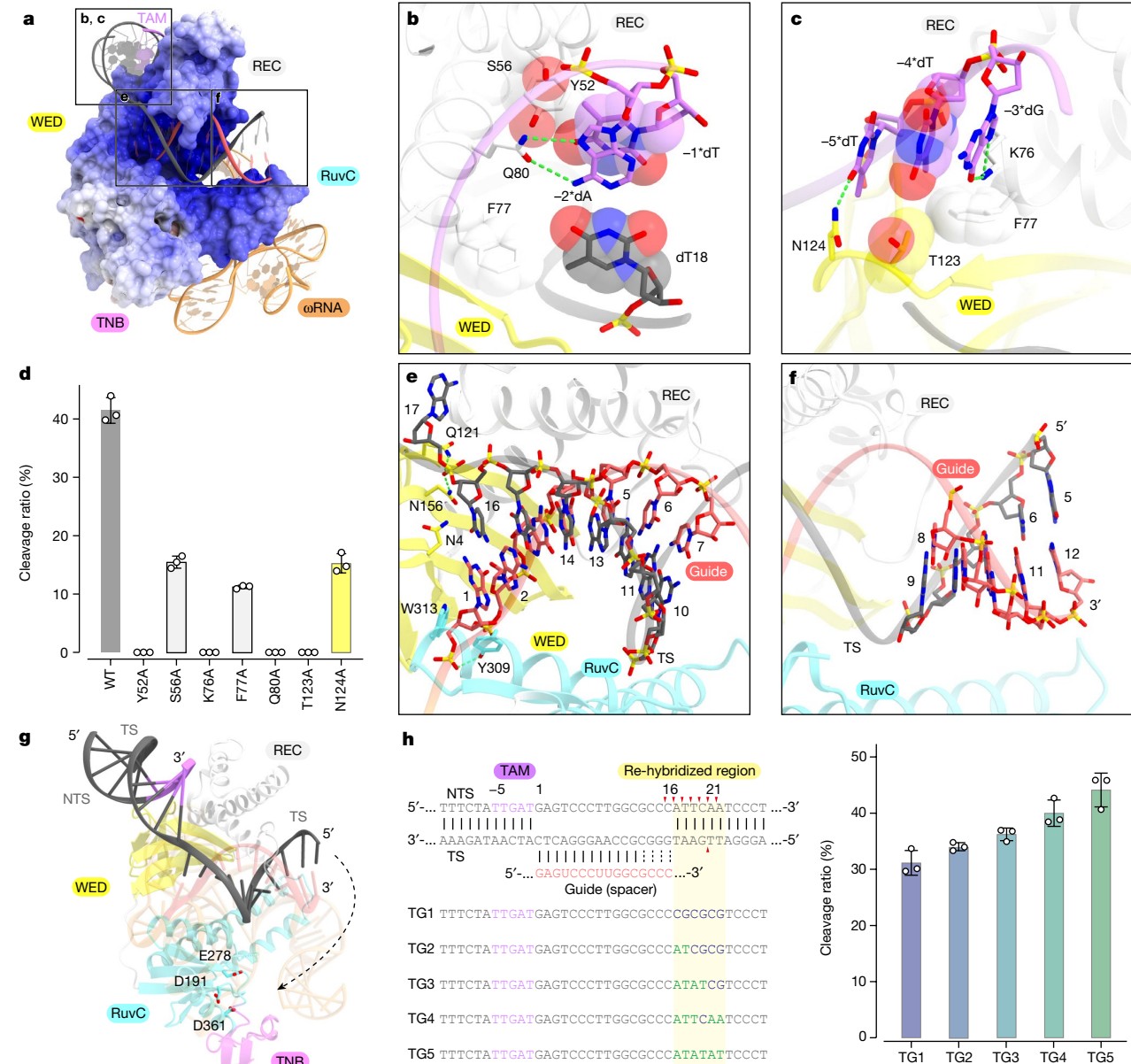

**Fig. 4 | Target DNA recognition and loading. a**, Recognition of the target DNA. TnpB is shown as an electrostatic surface potential model. The TAM duplex is bound to the cleft formed by the WED and REC domains. The guide RNA–target DNA heteroduplex is accommodated within a positively charged central channel formed by the REC and RuvC domains. **b,c**, Recognition of the TAM duplex. Nucleotides −1*dT, 18dT and −4*dT and residues Y52, S56, F77 and T123 are depicted by space-filling models. Hydrogen bonding and electrostatic interactions are shown as dashed lines. **d**, In vitro DNA cleavage activities of the wild-type TnpB and TAM recognition mutants with ωRNA with deleted 5′ region. The 3-kb linearized target DNA containing the 16-nt target sequence was incubated with the TnpB–ωRNA complex (250 nM) at 37 °C for 30 min. The reaction products were resolved, visualized and quantified with a MultiNA microchip electrophoresis device. Data are mean ± s.d. (*n* = 3 biologically independent samples). The experiments were repeated three times with similar results. **e,f**, Recognition of the TAM-proximal region of the guide RNA–target DNA heteroduplex (**e**) and the TAM-distal region of the heteroduplex (**f**). Hydrogen bonding and electrostatic interactions are shown as dashed lines. **g**, Positions of the active site and the target DNA. The possible trajectories of the target strand are shown by a dashed arrow. **h**, In vitro DNA cleavage activities of TnpB with five target DNAs with different sequences at the re-hybridized DNA duplex. Re-hybridized regions with altered sequences are highlighted with a yellow background. CG and AT sequences are coloured blue and green, respectively. Data are mean ± s.d. (*n* = 3 biologically independent samples). The experiments were repeated three times with similar results.

demonstrated that TAM-proximal double mismatches (positions 1–12) abolish the TnpB-mediated target DNA cleavage, whereas TAM-distal double mismatches (positions 13–16) reduce but still allow the target DNA cleavage by TnpB (Extended Data Fig. 8b). In addition, we analysed the target DNA specificity by performing a genome-wide off-target analysis for TnpB in human cells, and found many off-target sites with mismatches in the TAM-distal region[25] (Extended Data Fig. 8c). These results indicated that the TAM-proximal 12 bp (approximately) of a guide RNA–target DNA heteroduplex is important for the specificity of the RNA-guided target DNA cleavage by TnpB proteins.

## DNA cleavage mechanism

Previous studies revealed that TnpB cleaves the target and non-target strands at 21 nt and 15–21 nt downstream of the TAM, respectively, using a single RuvC active site[3]. Cas12 enzymes recognize the

PAM-distal region of the guide RNA–target DNA heteroduplex and the re-hybridized DNA duplex by their TNB domains, which facilitate the DNA unwinding and loading into the RuvC active site[19–21,26] (Extended Data Fig. 8d). By contrast, in the TnpB structure, the TAM-distal region of the heteroduplex is located far from the TNB domain, and the end of the TAM-distal region of the heteroduplex and the re-hybridized DNA duplex is disordered (Fig. 4g). These observations suggested that TnpB, unlike Cas12 enzymes, does not interact with these regions. Thus, we hypothesized that TnpB is unable to unwind the re-hybridized DNA by itself, but instead relies on spontaneously unwound DNA. To test this hypothesis, we performed an in vitro cleavage assay with five target DNAs with different sequences at the site of the DNA duplex that should be re-hybridized (Fig. 4h). We found that TnpB cleaves the target DNAs with AT-rich sequences more efficiently than those with GC-rich sequences (Fig. 4h). These results support the idea that TnpB spontaneously cleaves the unwound target DNA and does not unwind the target DNA by itself.

## Discussion

In this study, we determined the cryo-EM structure of the TnpB–ωRNA–dsDNA ternary complex, revealing how the relatively small TnpB protein recognizes its cognate ωRNA to form a compact effector complex that can cleave a dsDNA target that is complementary to the ωRNA guide sequence. Our structural and functional analyses revealed that TnpB requires the formation of a heteroduplex between an approximately 12-bp guide RNA and target DNA to mediate DNA cleavage and tolerates TAM-distal mismatches, suggesting that TnpB has several target sites in its own host genome. These observations indicated that TnpB may be involved in transposon propagation as well as transposon homing, although further biochemical analyses are needed to fully characterize the function of TnpB in transposition.

A structural comparison of TnpB with Cas12 enzymes highlighted the conservation of the molecular mechanism during their evolution from a key role in guided transposition to one in adaptive immunity[19–21] (Extended Data Fig. 9). TnpB and Cas12 enzymes share a bilobed architecture consisting of the REC and NUC lobes. By contrast, the Cas9 ancestors IscB and IsrB lack a REC domain and instead use their bulky cognate ωRNAs for an analogous role to the REC domains in Cas9[16,17,27] (Extended Data Fig. 10a). TnpB and Cas12 enzymes also recognize the TAM–PAM duplex in the groove formed by the WED and REC domains, facilitating the initial DNA unwinding and the guide RNA–target DNA heteroduplex formation. The structural comparison also revealed mechanistic differences for the target DNA loading, although the TnpB and Cas12 enzymes cleave the target and non-target strands at the single RuvC site. Upon cleavage of the non-target strand, Cas12 enzymes unwind the re-hybridized DNA duplex and load the target strand into the RuvC active site, whereas TnpB is unable to unwind the re-hybridized DNA duplex and captures spontaneously unwound DNA near the RuvC active site. Furthermore, the structural comparisons with two different types of compact Cas12 enzyme, UnCas12f1 and MmCas12m2 (also known as Cas12U-1), which show relatively high sequence similarity to TnpB proteins, provided insights into the evolution of type V CRISPR–Cas12 effectors[15,19] (Fig. 5 and Extended Data Fig. 10b). In the present structure, TnpB uses its REC and RuvC domains to recognize a relatively short approximately 12-bp heteroduplex (Fig. 5 and Extended Data Fig. 9). Although UnCas12f1 has a similar domain organization to TnpB, it functions as a dimer to interact with an approximately 20-bp heteroduplex, with the second molecule recognizing the terminal 6 bp of the heteroduplex (Fig. 5 and Extended Data Fig. 9). By contrast, although MmCas12m2 functions as a monomer, it uses a characteristic kinked coiled-coil insertion in the REC domain (referred to as the REC2 domain), in addition to the REC and RuvC domains, to recognize an approximately 17-bp heteroduplex (Fig. 5). All Cas12 enzymes, except for Cas12f and Cas12k (which are associated with one

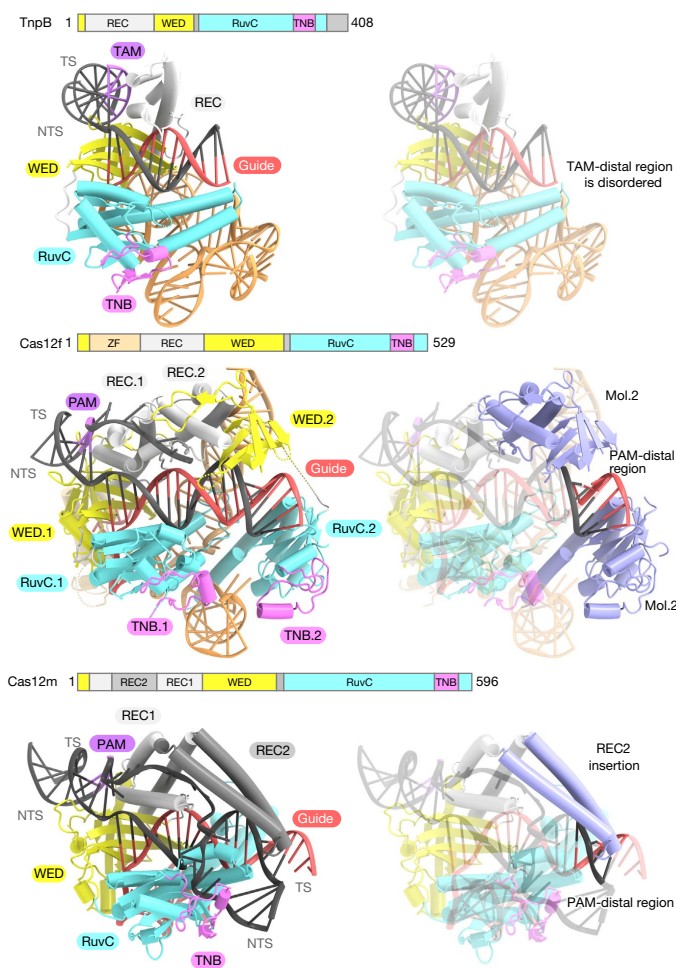

**Fig. 5 | Comparison of TnpB with Cas12f and Cas12m.** Structural comparison of TnpB with Cas12f (Cas12f from an uncultured archaeon) (PDB ID: 7C7L) and Cas12m (Cas12m from *Mycobacterium mucogenicum*) (PDB ID: 8HHL). Cas12f and Cas12m share high sequence similarity with TnpB proteins. Cas12f mol.2-and Cas12m-specific insertions (REC2) are highlighted in blue. These regions have a crucial role for the recognition of the PAM-distal region of the guide RNA–target DNA heteroduplex, suggesting that type V Cas12 enzymes acquired an ability to recognize the PAM-distal end of the guide RNA–target DNA heteroduplex in order to engage in CRISPR–Cas adaptive immunity. ZF, zinc-finger domain.

or more additional protein molecules), have the REC2 insertion to allow the recognition of the PAM-distal region (Extended Data Fig. 9). These structural findings suggest that Cas12 enzymes acquired the ability to recognize the PAM-distal end of the guide RNA–target DNA heteroduplex through two distinct strategies, either by dimerization or by REC2 insertion, to achieve the target specificity required for CRISPR–Cas adaptive immunity via the increased length of the effective guide (Fig. 5). The development of these different mechanistic strategies from a common minimal TnpB scaffold for the same ultimate function suggested the distinct evolutionary origins for the different Cas12 lineages[1]. Therefore, these findings indicate that the evolutionary path of Cas12 enzymes from TnpB is distinct from that of Cas9 enzymes from IscB, as Cas12 may have arisen from TnpB on multiple independent occasions, in contrast to the single evolutionary event that probably gave rise to all extant Cas9 variants.

Our structure also revealed that the ωRNA contains the guide–PK–stem structure, which is highly conserved among all guide RNAs of Cas12 enzymes (Extended Data Fig. 7a). This structural similarity strongly suggests that (1) the single guides of the tracrRNA-independent Cas12

variants (Cas12m to Cas12a) may have evolved from a condensed and duplicated version of the ωRNA ancestor to allow the formation of a CRISPR array (with palindromic repeats), and (2) the tracrRNA–crRNA pair present in some Cas12 systems (Cas12f to Cas12b) may have evolved from a split version of the ωRNA ancestor to allow the formation of a standalone tracrRNA gene, and a CRISPR RNA that later expanded into a full array (with non-palindromic repeats). A similar splitting hypothesis was proposed for the evolution of the Cas9 dual crRNA–tracrRNA guide from ωRNA[2].

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

## Methods

### Sample preparation

The genes encoding ISDra2 TnpB (TnpB from *D. radiodurans* ISDra2; residues 1 to 408) and the 247-nt ωRNA were synthesized by Eurofin Genomics and cloned into the modified pETDuet vector (Novagen). The N-terminally MBP-tagged TnpB and ωRNA were co-expressed in *Escherichia coli* Rosetta2 (DE3). The *E. coli* cells were cultured at 37 °C until the $A_{600}$ reached 0.8, and protein expression was then induced by the addition of 0.2 mM isopropyl β-ᴅ-thiogalactopyranoside (Nacalai Tesque). The *E. coli* cells were further cultured at 20 °C overnight, collected by centrifugation, resuspended in buffer A (20 mM Tris-HCl, pH 8.0, 0.5 M NaCl, 5% glycerol, and 1 mM DTT), and then lysed by sonication. The lysates were centrifuged, and the supernatant was mixed with 3 ml of amylose resin (New England Biolabs). The mixture was loaded into a Poly-Prep column (Bio-Rad), and the TnpB–ωRNA complex was eluted with buffer B (20 mM Tris-HCl, pH 8.0, 40 mM maltose, 0.2 M NaCl, 5% glycerol, and 1 mM DTT). The complex was incubated with HRV3C protease overnight, and then loaded onto a 5 ml HiTrap Heparin column (GE Healthcare) equilibrated with buffer C (20 mM Tris-HCl, pH 8.0, and 0.2 M NaCl). The peak fractions were collected and stored at −80 °C in buffer D (20 mM Tris-HCl, pH 8.0, 0.2 M NaCl and 20% glycerol) until use. Mutations were introduced by a PCR-based method, and sequences were confirmed by DNA sequencing (Supplementary Tables 1 and 2). Since the 3′ end of the *tnpB* gene overlapped with part of the ωRNA, it was difficult to perform PCR with the plasmid containing the full-length ωRNA. Thus, all TnpB mutants were created by introducing mutations on the DNA plasmid containing ωRNA with deleted 5′ region (−231 to −117).

### Cryo-EM analysis

The TnpB–ωRNA–target DNA ternary complex was prepared for cryo-EM analysis according to the following procedure. A double-stranded DNA with phosphorothioate modifications at the cleavage sites was prepared by annealing a 35-nt target DNA strand and a 35-nt non-target strand containing a TTGAT TAM, at 95 °C for 2 min. The purified TnpB–ωRNA complex was incubated with the target DNA at room temperature for 30 min. The TnpB–ωRNA–target DNA ternary complex was purified on a Superdex 200 Increase 10/300 column (GE Healthcare), equilibrated with buffer E (20 mM Tris-HCl, pH 8.0, 150 mM NaCl, 2 mM $MgCl_2$, 10 μM $ZnCl_2$, and 1 mM DTT). The purified complex solution (0.5 mg ml$^{-1}$ final concentration) was applied to freshly glow-discharged Au 300 mesh R1.2/1.3 grids (Quantifoil) after adding 3 μl of amylamine, using a Vitrobot Mark IV (FEI) at 4 °C, with a waiting time of 10 s and a blotting time of 4 s under 100% humidity conditions. The grids were then plunge-frozen in liquid ethane cooled to the temperature of liquid nitrogen.

Cryo-EM data were collected using a Titan Krios G3i microscope (Thermo Fisher Scientific), running at 300 kV and equipped with a Gatan Quantum-LS Energy Filter (GIF) and a Gatan K3 Summit direct electron detector in the electron counting mode (The University of Tokyo, Japan). Movies were recorded at a nominal magnification of 105,000×, corresponding to a calibrated pixel size of 0.83Å, with a total dose of approximately 50 electrons per Å$^2$ per 48 frames. The data were automatically acquired using the EPU software (Thermo Fisher Scientific), with a defocus range of −0.8 to −1.6 μm, and 3,570 movies were obtained.

### Image processing

The data processing was performed with the cryoSPARC v3.3.2 software platform[28]. The dose-fractionated movies were aligned using patch motion correction, and the contrast transfer function (CTF) parameters were estimated using the Patch-based CTF estimation. From the 3,570 motion-corrected and dose-weighted micrographs, 2,136,853 particles were automatically picked using blob picker in cryoSPARC.

The particles were subjected to several rounds of reference-free 2D classifications to create particle sets. The particles were curated by cryoSPARC heterogenous refinement (*n* = 4), using the map derived from the cryoSPARC ab initio reconstruction as a template. The selected particles were subjected to 3D variability analysis, and the resulting maps with different conformations were used for subsequent heterogeneous refinement. The best class containing 98,483 particles was refined using non-uniform refinement[29] after CTF refinement, yielding a map at 3.21 Å resolution, according to the Fourier shell correlation (FSC) = 0.143 criterion[30]. The local resolution was estimated by cryoSPARC.

### Model building and validation

The model was built using the predicted model of the ISDra2 TnpB protein created by AlphaFold2 as the reference[31], followed by manual model building with COOT[32]. The model was refined using phenix.real_space_refine ver. 1.20.1[33], with secondary structure and metal coordination restraints. The metal coordination restraints were generated using ReadySet, as implemented in PHENIX. The structure validation was performed using MolProbity in the PHENIX package[34]. The EMRinger score[35] and 3DFSC sphericity[36] were calculated by PHENIX and the 3DFSC processing Server (https://3dfsc.salk.edu/upload/info/), respectively. The statistics of the 3D reconstruction and model refinement are summarized in Extended Data Table 1. The cryo-EM density map figures were generated using UCSF ChimeraX[37]. Molecular graphics figures were prepared using CueMol (http://www.cuemol.org).

### Northern blotting analysis

Total RNA was extracted from cells expressing ωRNA or ωRNA-MBP or ωRNA-TnpB with TRIzol LS (Thermo Fisher Scientific) according to the manufacturer's instructions. TnpB-interacting ωRNA was extracted from the purified TnpB–ωRNA complex with TRIzol LS (Thermo Fisher Scientific) according to the manufacturer's instructions. Three micrograms of total RNA, 140 ng of in vitro-transcribed ωRNA and 140 ng of TnpB-interacting ωRNA were resolved by electrophoresis on a 10% polyacrylamide gel containing 7 M urea, followed by staining with GelGreen (Biotium). Fluorescence was visualized by an FLA-7000 imaging analyser (Fujifilm). The RNAs were transferred to a Hybond N$^+$ membrane (Cytiva) by electroblotting for 1 h at 1.5 mA cm$^{-2}$ in 1× TBE using a Transblot Turbo (Bio-Rad), and crosslinked by two rounds of UV irradiation (254 nm, 120 mJ cm$^{-2}$; CL-1000, UVP). The membrane was treated with hybridization buffer (5% PEG 6000 (w/v), 7.5% SDS (w/v), 0.5% casein (w/v), 1 mM EDTA (pH 8.0), and 282 mM sodium phosphate buffer (pH 7.4)) at 52 °C for 1 h, and then subjected to hybridization with 2 pmol of the 5′-$^{32}$P-labelled DNA probes at 50 °C overnight. The sequences of DNA probes are as follows: 5′-TTCTTCACTTCGGGATTCTTGAATC-3′ (probe I), 5′-CGTCTCGGTCATGGGTTTCCCCACA-3′ (probe II), 5′-GTCTGAGATTCCCGCAGCCACCAAC-3′ (probe III), 5′-GCAGACCATTGCCCGCCGAAGCAGG-3′ (probe IV), 5′-GGGCGCCAAGGGACTCTTGAACCTC-3′ (probe V) (Supplementary Table 2). The membrane was washed four times with 2× SSC (0.3 M NaCl, 30 mM sodium citrate (pH 7.0)), dried, and exposed to an imaging plate. Radioactivity was visualized by using an FLA-7000 imaging analyser. Uncropped images are available in the Source Data file.

### LC−MS analysis

Each band on the gel was cut into cubes smaller than 1 mm$^3$, soaked in 150 μl elution buffer (3 M sodium acetate (pH 5.3), 1 mM EDTA (pH 8.0), and 0.1% SDS), and shaken for 2 h at 37 °C. The buffer was transferred to a new tube. The gel fragments were then shaken with another 150 μl of elution buffer at 37 °C overnight. The elution buffers were combined, and after glycogen addition, the RNA was recovered by ethanol precipitation. The RNA precipitate was dissolved in water and digested by RNase A (Thermo Fisher Scientific), and then analysed by LC−MS. RNA fragment analysis was performed with an UltiMate 3000 RSLCnano

system coupled with an Orbitrap Eclipse Tribrid (Thermo Fisher Scientific). One picomole of the RNA digest was diluted with 10 mM triethylammonium acetate, and loaded on a trap column (Acclaim PepMap 100 C18, 100 μm ID × 20 mm, Thermo Fisher Scientific). RNA fragments were separated on an ODS column (HiQ sil C18W-3, 100 μm ID × 100 mm, Techno Alpha) at a 300 nl min⁻¹ flow rate. Separation was started with 99% mobile phase A (0.4 M hexafluoroisopropanol in water) and 1% B (0.4 M hexafluoroisopropanol in 50% methanol) for 10 min, and then B % was increased to 60% by linear gradient over 32 min. Eluents were injected into the ESI source through a nanoESI emitter (LOTUS emitters, FOSSILIONTECH), and ions were scanned by MS in the negative polarity mode.

### In vitro DNA cleavage assay

For the in vitro cleavage assay, the TnpB–ωRNA complexes (wild-type or mutants) were purified in a similar manner to that for the complex prepared for the cryo-EM analysis. Protein concentrations were measured using a Bradford Protein Assay Kit (TAKARA). The DNA cleavage activity of TnpB was measured by in vitro DNA cleavage assays. The TnpB–ωRNA complex (2 μl, final concentration 250 nM) was mixed with the 3-kb linearized plasmid target containing the 16-nt target sequence and the TTGAT TAM (8 μl, 100 ng) (Supplementary Table 1), and incubated at 37 or 50 °C for 30 min in 10 μl reaction buffer (20 mM HEPES, pH 7.5, 50 mM KCl, 2 mM MgCl₂, 1 mM DTT, and 5% glycerol). The reaction was stopped by the addition of quench buffer, containing EDTA (20 mM final concentration) and Proteinase K (40 ng). The reaction products were resolved, visualized, and quantified with a MultiNA microchip electrophoresis device (Shimadzu). In vitro cleavage experiments were performed at least three times.

### Mammalian genome editing assays

Mammalian cell culture experiments were performed in the HEK293FT cell line, grown in Dulbecco's modified Eagle's medium with high glucose, sodium pyruvate, and GlutaMAX (Thermo Fisher), supplemented with 1× penicillin–streptomycin (Thermo Fisher) and 10% fetal bovine serum (VWR Seradigm). All cells were maintained at confluency below 80%. All transfections were performed with Lipofectamine 3000 (Thermo Fisher) in 96-well plates, unless otherwise noted. Cells were plated at approximately 20,000 cells per well 16–20 h prior to transfection, to ensure 90% confluency at the time of transfection. To evaluate indel efficiencies, transfection plasmids were combined with OptiMEM I Reduced Serum Medium (Thermo Fisher) to a total volume of 20 μl per well. Separately, 18.8 μl of OptiMEM was combined with 1.2 μl of Lipofectamine 3000. The plasmid and Lipofectamine solutions were then combined, and 10 μl was pipetted onto each well. Genomic DNA was collected 96 h after transfection by removing the supernatant and resuspending each well in 50 μl of QuickExtract DNA Extraction Solution (Lucigen). Cells were lysed by cycling at 65 °C for 15 min, 68 °C for 15 min, and 95 °C for 10 min. A 3 μl portion of lysed cells was used as the input in each PCR reaction for deep sequencing, and indel frequencies were quantified by CRISPResso2[38]. Genome-wide off-target analysis was performed using tagmentation-based tag integration site sequencing (TTISS) as described previously[25], and sites identified by TTISS were subjected to quantification of indel frequencies in a separate experiment (Supplementary Tables 1 and 2).

### Reporting summary

Further information on research design is available in the Nature Portfolio Reporting Summary linked to this article.

## Data availability

The atomic models have been deposited in the Protein Data Bank under the accession code 8H1J. The cryo-EM density map has been deposited in the Electron Microscopy Data Bank under the accession code EMD-34428. Source data are provided with this paper.

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

**Acknowledgements** The authors thank K. Yamashita for assistance with the model building and K. S. Makarova and E. V. Koonin for assistance with the creation of the UPGMA dendrogram. S.N. was supported by award nos. T32GM007753 and T32GM144273 from the National Institute of General Medical Sciences. The content is solely the responsibility of the authors and does not necessarily represent the official views of the National Institute of General Medical Sciences or the National Institutes of Health. J.v.d.O. was supported by the Dutch Research Council (NWO Spinoza grant SPI 93-537, and NWO Gravitation grant 024.003.019), and the European Research Council (ERC-AdG-834279). T.S. was supported by Exploratory Research for Advanced Technology (ERATO, JPMJER2002) from the Japan Science and Technology Agency (JST). F.Z. is supported by an NIH grant (2R01HG009761-05); Howard Hughes Medical Institute; Poitras Center for Psychiatric Disorders Research at MIT; Hock E. Tan and K. Lisa Yang Center for Autism Research at MIT; K. Lisa Yang and Hock E. Tan Molecular Therapeutics Center at MIT; K. Lisa Yang Brain–Body Center at MIT; Broad Institute Programmable Therapeutics Gift Donors; The Pershing Square Foundation, William Ackman, and Neri Oxman; James and Patricia Poitras; BT Charitable Foundation; Asness Family Foundation; the Phillips family; David Cheng; and Robert Metcalfe. O.N. was supported by AMED grant no. JP223fa627001 and JP19am0401005, the Platform Project for Supporting Drug Discovery and Life Science Research (Basis for Supporting Innovative Drug Discovery and Life Science Research (BINDS)) from AMED under grant nos. JP22ama121002 and JP22ama121012, and the Cabinet Office, Government of Japan, Public/Private R&D Investment Strategic Expansion Program (PRISM) grant no. JPJ008000.

**Author contributions** R.N. performed biochemical and structural analyses with assistance from H.H., S.N.O., Y. Shuto, T.T., F.K.S., T.K., Y.K. and Y.I. R.N., H.H. and S.N.O. performed model building and structural refinement. S.N. and S.K. performed cell biological experiments. H.A.-T. performed bioinformatics analysis. X.Y., Y. Sakoguchi, T.O., H.N., N.D. and T.S. designed and performed northern blotting and LC–MS analyses. W.Y.W. and J.v.d.O. created the UPGMA dendrogram. O.N. and F.Z. conceived the project. R.N. and O.N. wrote the manuscript with help from all authors. O.N. and F.Z. supervised the research.

**Competing interests** F.Z. is a co-founder of Editas Medicine, Beam Therapeutics, Pairwise Plants, Arbor Biotechnologies and Sherlock Biosciences. J.v.d.O. is a co-founder of NTrans Technologies and a scientific advisor for NTrans Technologies, Scope Biosciences and Hudson River Biotechnology. O.N. is a co-founder, board member and scientific advisor for Curreio. The remaining authors declare no competing interests.

**Additional information**
**Correspondence and requests for materials** should be addressed to Osamu Nureki.

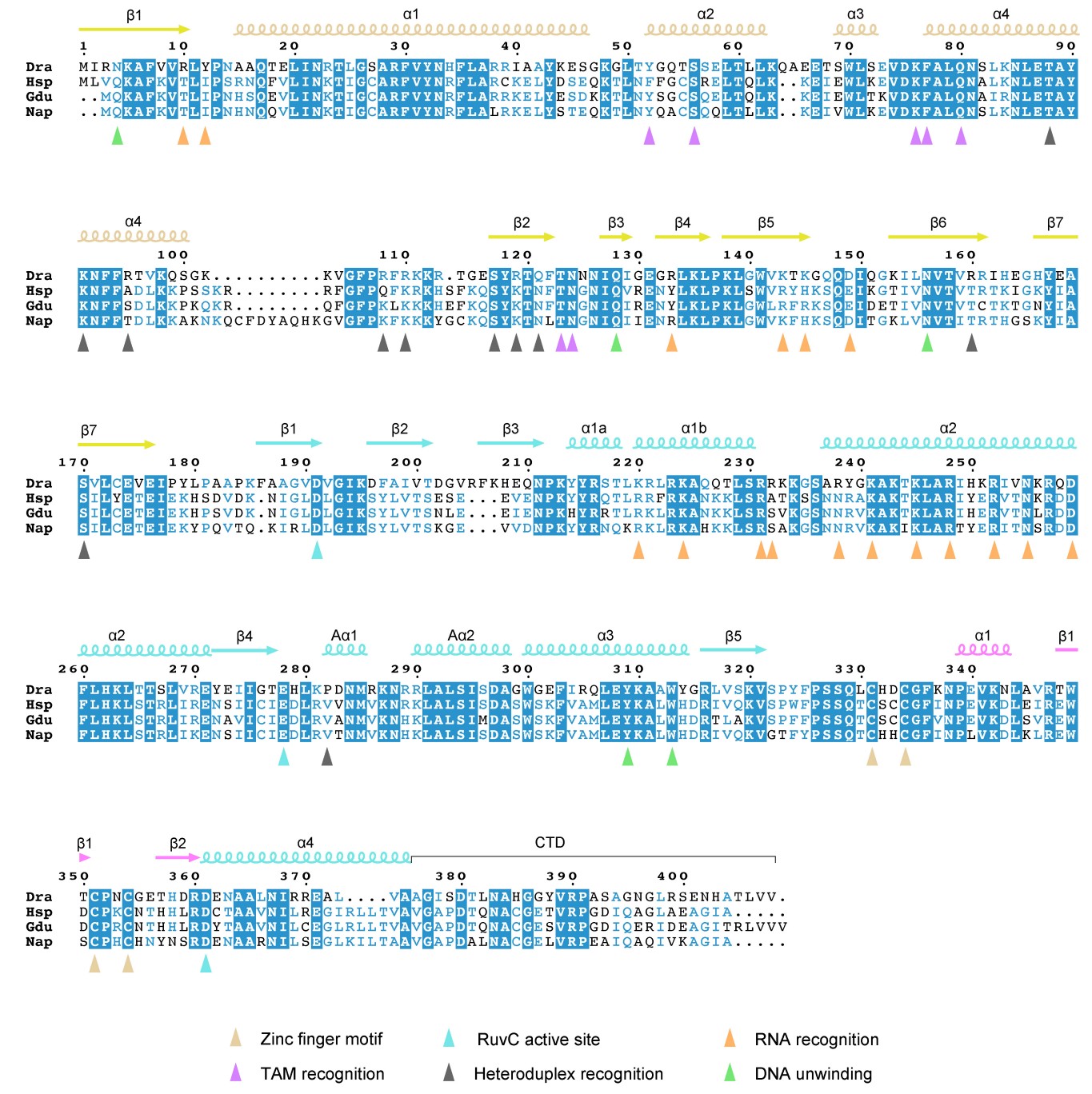

**Extended Data Fig. 1 | Multiple sequence alignment of TnpB orthologs.**
Dra, TnpB from *Deinococcus radiodurans* (WP_010887311.1); Hsp, TnpB from *Hydrococcus* sp.RU_2_2 (NJM87737.1); Gdu, TnpB from *Gloeocapsopsis dulcis* (WP_105220324.1); Nap, TnpB from *Nodularia sphaerocarpa* (WP_239728827.1). The secondary structure of TnpB is indicated above the sequences. Key residues of TnpB are marked below the sequences by triangles. The figure was prepared using Clustal Omega (http://www.ebi.ac.uk/Tools/msa/clustalo) and ESPript3 (http://espript.ibcp.fr/ESPript/ESPript).

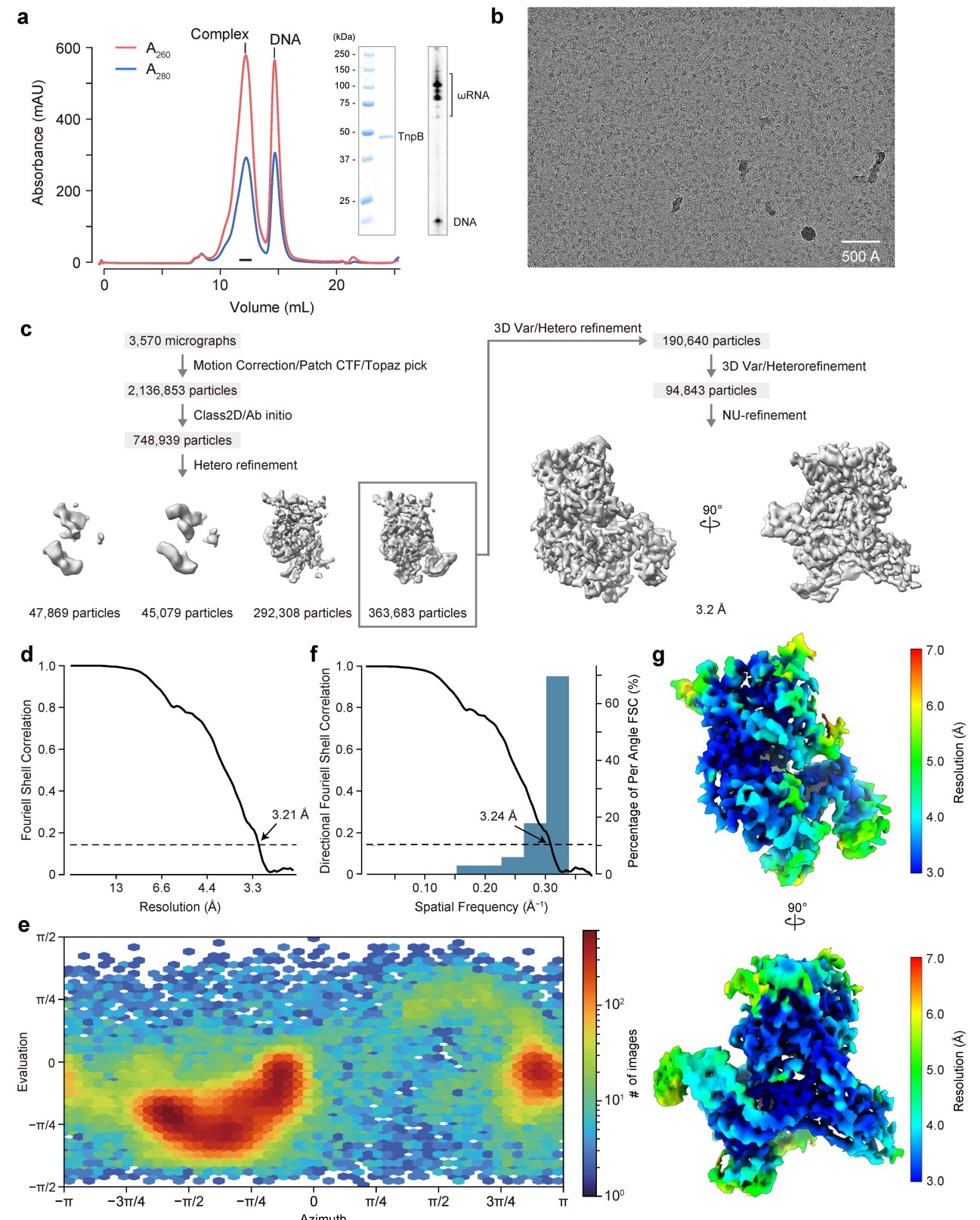

**Extended Data Fig. 2 | Single-particle cryo-electron microscopy analysis.**
(**a**) Size-exclusion chromatography profile of the TnpB–ωRNA–target DNA complex. The peak fraction (indicated by a black bar) was analyzed by SDS-PAGE and urea-PAGE, and then used for cryo-EM analysis. (**b**) A representative cryo-EM image of the TnpB–ωRNA–target DNA complex, recorded on a 300 kV Titan Krios with a K3 camera. (**c**) Single-particle cryo-EM image processing workflow.

(**d**) Fourier shell correlation (FSC) curve for the 3D reconstruction. The gold-standard cutoff (FSC = 0.143) is marked with the black dotted line. (**e**) Direct distribution plot (Viewing distribution plot). (**f**) Direction 3DFSC plots calculated by 3DFSC processing Server (https://3dfsc.salk.edu/upload/info/). (**g**) Local-resolution cryo-EM density map.

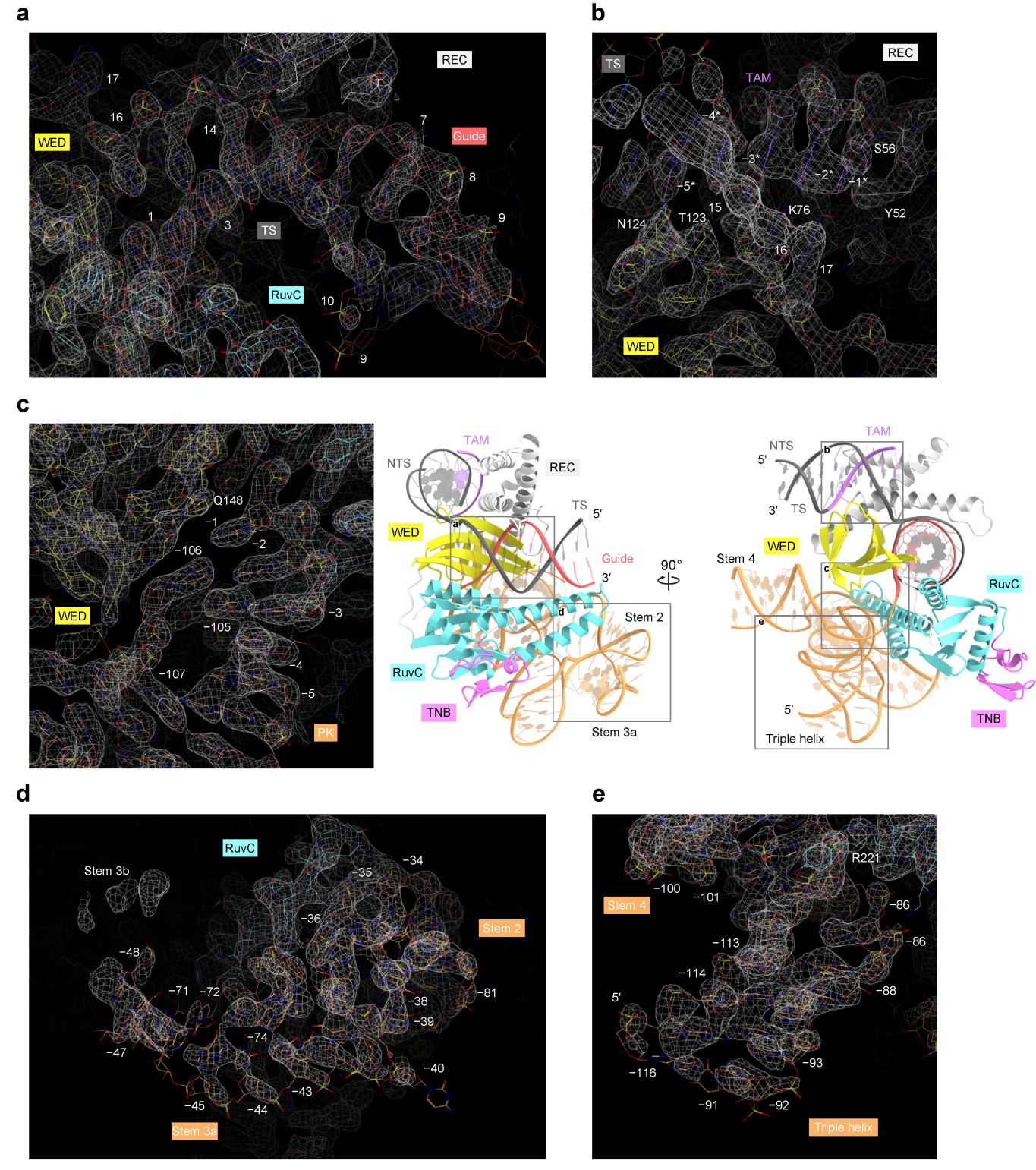

**Extended Data Fig. 3 | Cryo-EM density map.** Cryo-EM density maps for the guide RNA–target DNA heteroduplex (**a**), the TAM duplex (**b**), the pseudoknot structure (**c**), stems 2 and 3 (**d**), and stem 1 and the triple helix structure (**e**). The ambiguous density in (**d**) corresponds to stem 3b.

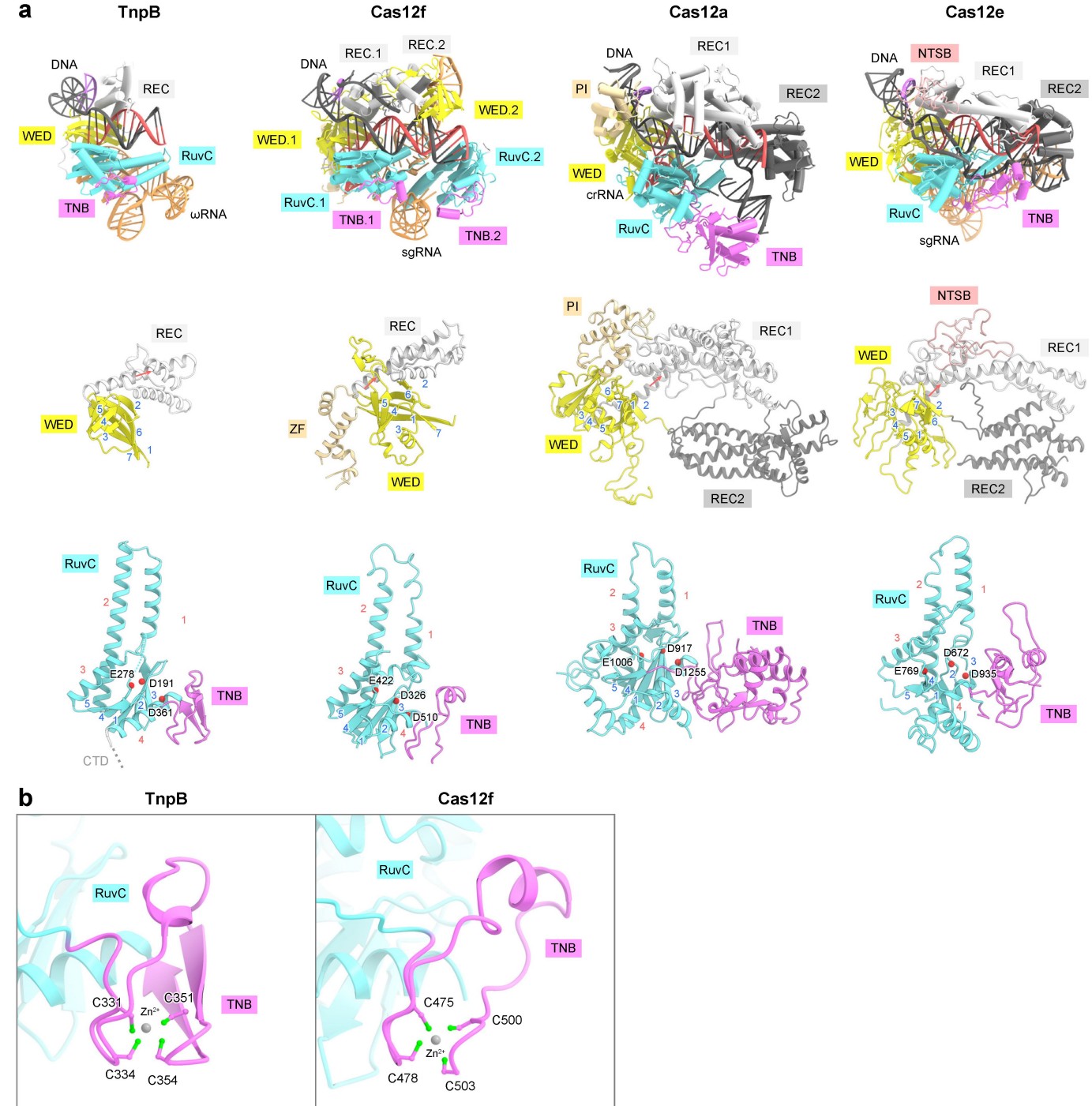

**Extended Data Fig. 4 | Domain structures.** (**a**) The structures of TnpB, Cas12f (Cas12f from an uncultured archaeon) (PDB ID: 7C7L), Cas12a (Cas12a from *Francisella novicida*) (PDB ID: 6I1K), and Cas12e (Cas12e from a Deltaproteobacterium, also known as CasX) (PDB ID: 6NY2) were aligned, based on the guide RNA–target DNA heteroduplex. TnpB and these type V Cas12 enzymes commonly adopt a bilobed architecture containing the REC and NUC lobes, which is structurally similar to that of the WED and RuvC domains, despite their limited sequence identity (the conserved α helices (red) and β strands (blue) are numbered). The REC lobes commonly consist of the WED and REC domains, and the WED domain comprises an OB fold (the conserved α helices (red) and β strands (blue) are numbered). The first α helices in their REC domains are located at similar positions (as indicated by red arrows).

Cas12f has the zinc finger (ZF) domain inserted between the WED and REC domains. Cas12a has the PAM-interacting (PI) domain inserted into the WED domain. Cas12e has the non-target-strand binding (NTSB) domain inserted into the REC1 domain. The NUC lobes of TnpB consist of the RuvC, TNB, and CTD domains, although the CTD domain is disordered. The RuvC domains comprise an RNase H fold (the conserved α helices (red) and β strands (blue) are numbered). The TNB domains are inserted between the conserved strand β5 and helix α4 in the RuvC domains. The TNB domains share low sequence similarity and adopt distinct structures. (**b**) The TNB domains of TnpB and Cas12f. Both TnpB and Cas12f contain a typical CXXC---CXXC zinc finger motif, and each of which binds zinc ions.

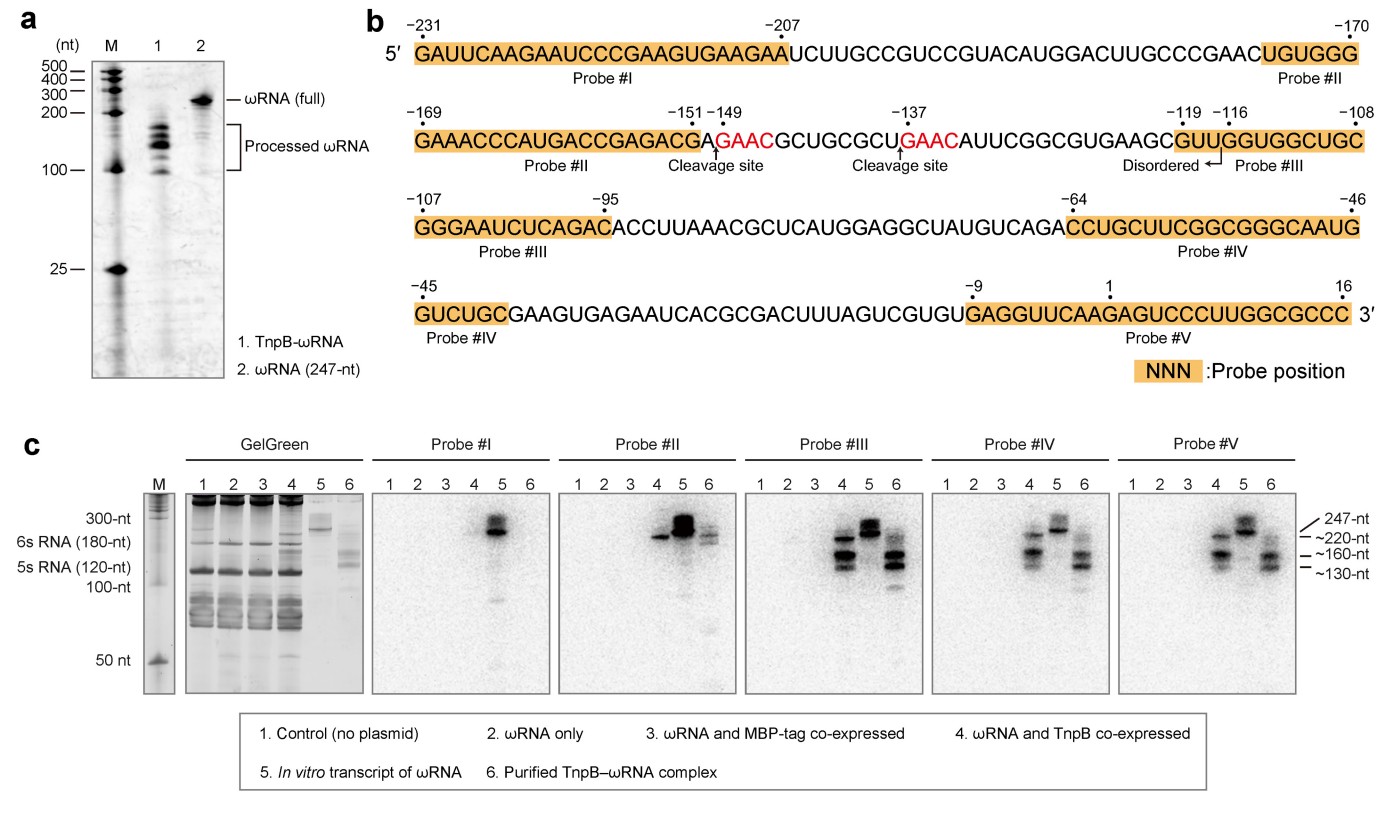

**a**

(nt) M 1 2

500
400
300 ← ωRNA (full)
200
] Processed ωRNA
100

1. TnpB-ωRNA
2. ωRNA (247-nt)

**b**

−231                                          −207                                      −170
5′ GAUUCAAGAAUCCCGAAGUGAAGAAUCUUGCCGUCCGUACAUGGACUUGCCCGAACUGUGGG
   Probe #I                                                              Probe #II

−169                −151 −149      −137                      −119 −116        −108
GAAACCCAUGACCGAGACGAGAACGCUGCGCUGAACAUUCGGCGUGAAGCGUUGGUGGCUGC
Probe #II            Cleavage site    Cleavage site              Disordered ←  Probe #III

−107              −95                                              −64                      −46
GGGAAUCUCGACACCUUAAACGCUCAUGGAGGCUAUGUCAGACCUGCUUCGGCGGGCAAUG
Probe #III                                                        Probe #IV

−45            −9        1              16
GUCUGCGAAGUGAGAAUCACGCGACUUUAGUCGUGUGAGGUUCAAGAGUCCCUUGGCGCCC 3′
Probe #IV                              Probe #V

NNN :Probe position

**c**

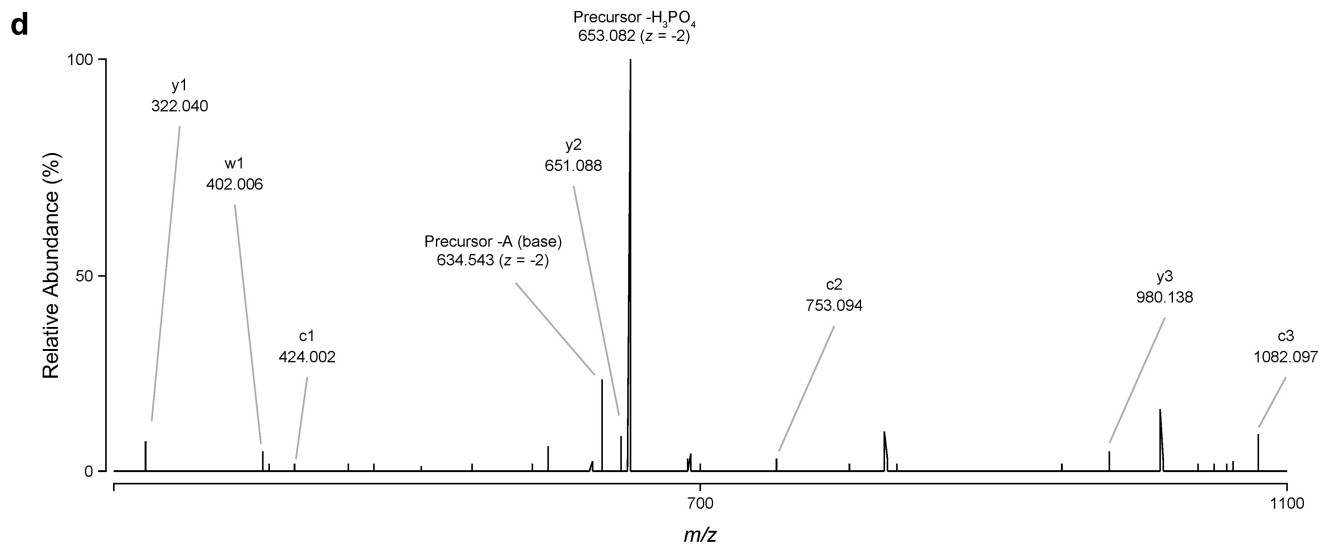

M  1 2 3 4 5 6

300-nt
6s RNA (180-nt)
5s RNA (120-nt)
100-nt

50 nt

GelGreen | Probe #I | Probe #II | Probe #III | Probe #IV | Probe #V

247-nt
~220-nt
~160-nt
~130-nt

1. Control (no plasmid)    2. ωRNA only    3. ωRNA and MBP-tag co-expressed    4. ωRNA and TnpB co-expressed

5. *In vitro* transcript of ωRNA    6. Purified TnpB–ωRNA complex

**d**

Precursor -H₃PO₄
653.082 (z = -2)

y1
322.040

w1
402.006

y2
651.088

c1
424.002

Precursor -A (base)
634.543 (z = -2)

c2
753.094

y3
980.138

c3
1082.097

Relative Abundance (%)

100

50

700                    1100

*m/z*

**Extended Data Fig. 5 | Biochemical characterization of ωRNA. (a)** Urea-PAGE analysis of the purified TnpB–ωRNA complex. Although we co-expressed TnpB with a 247-nt ωRNA, the purified TnpB was bound to heterogenous 100–160-nt parts of the ωRNA. **(b)** The sequence of full-length ωRNA (−231G to 16C). The probe positions used in the northern blotting analysis are shown in orange. Two GAAC sites that could generate a pGAACp fragment by RNase A digestion are indicated in red. **(c)** Northern blotting of ωRNA. Total RNAs prepared from *E. coli* wild-type cells (lane 1), ωRNA expressing cells (lane 2), ωRNA and MBP co-expressing cells (lane 3), ωRNA and TnpB co-expressing cells (lane 4),

the *in vitro* transcript of ωRNA (lane 5), and the ωRNA extracted from TnpB (lane 6) were resolved by 10% denaturing PAGE and stained with GelGreen (left panel) or subjected to northern blotting (right panels, Probes I–V). 6S RNA (180-nt), 5S rRNA (120-nt), tRNAs (76 to 93-nt) in total RNA and 50-nt, 100-nt, and 300-nt RNA markers are indicated (lane M). **(d)** Collision-induced dissociation spectrum of the pGAACp fragment from ωRNA digested by RNase A. The divalent negatively charged ion of pGAACp was used as the precursor ion for CID. The product ions in the CID spectrum are assigned on the sequence.

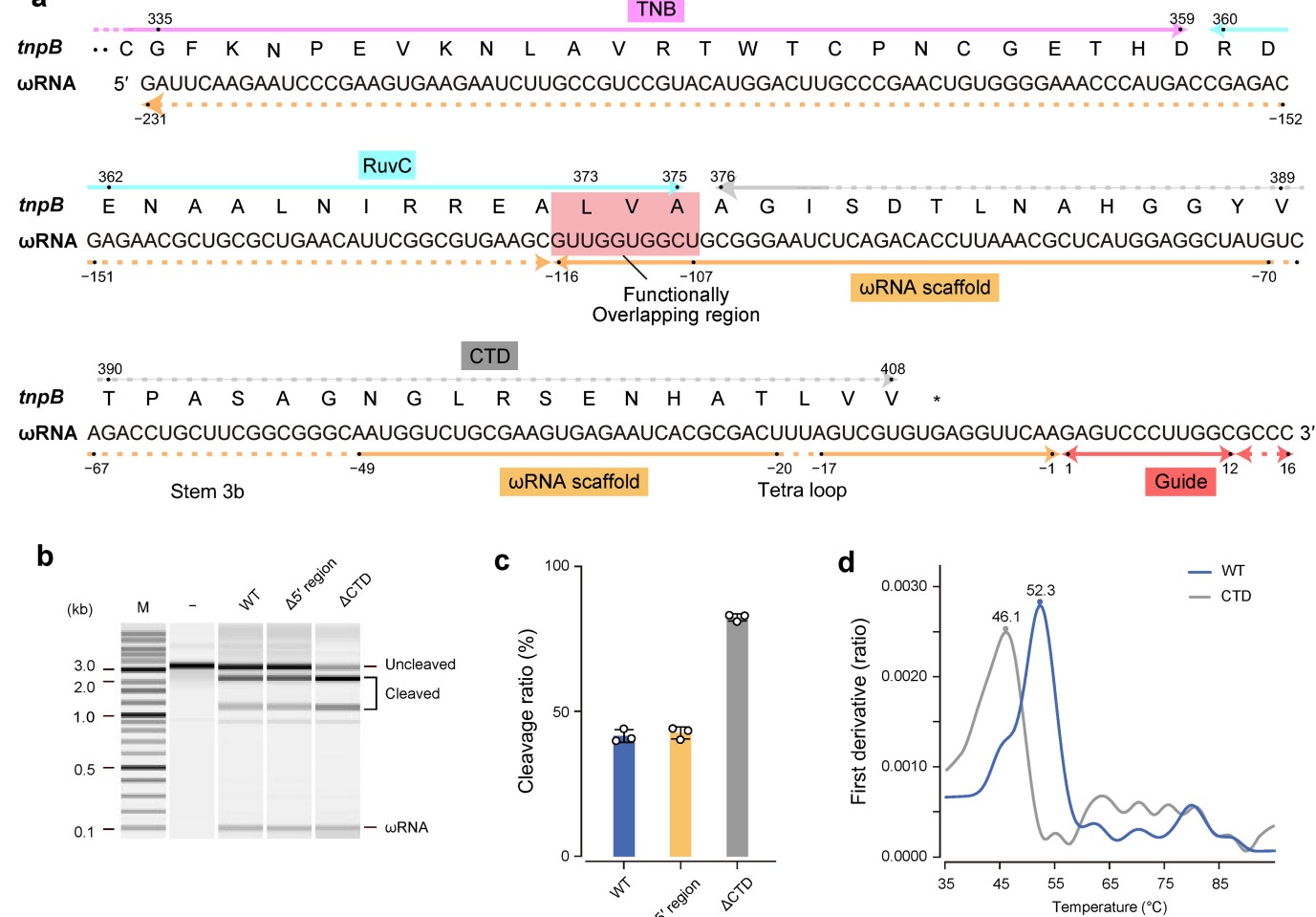

**Extended Data Fig. 6 | Overlapping region between *tnpB* gene and ωRNA.**
(**a**) Schematic illustrating the overlapping region between the *tnpB* gene (residues 335 to 408) and ωRNA (−231G to −10U). The disordered regions of TnpB and ωRNA are shown as dotted arrows. Except for a few nucleotides (indicated by the red box), the functionally important regions of the *tnpB* gene and ωRNA do not overlap. (**b**) *In vitro* DNA cleavage assay of the wild-type (WT) TnpB, the 5′ region of the ωRNA-deleted (Δ−231G to −117T; Δ5′ region) mutant, and the C-terminal domain-deleted (Δ376 to 408; ΔCTD) mutant. The linearized plasmid target, containing a 16-nt target sequence and a TTGAT TAM sequence,

was incubated with the TnpB–ωRNA complex at 37 °C for 30 min. The cleavage products were then analyzed by a MultiNA microchip electrophoresis system. (**c**) Quantification of the DNA cleavage data in (b). Data are mean ± s.d. (*n* = 3, biologically independent samples). The experiments were repeated three times with similar results. Source data are provided as a Source data file. (**d**) Thermal shift assay of the WT TnpB and the ΔCTD mutant, calculated by a NanoTemper Tycho NT.6 Differential Scanning Fluorimeter, which determines the inflection temperature ($T_i$) of samples.

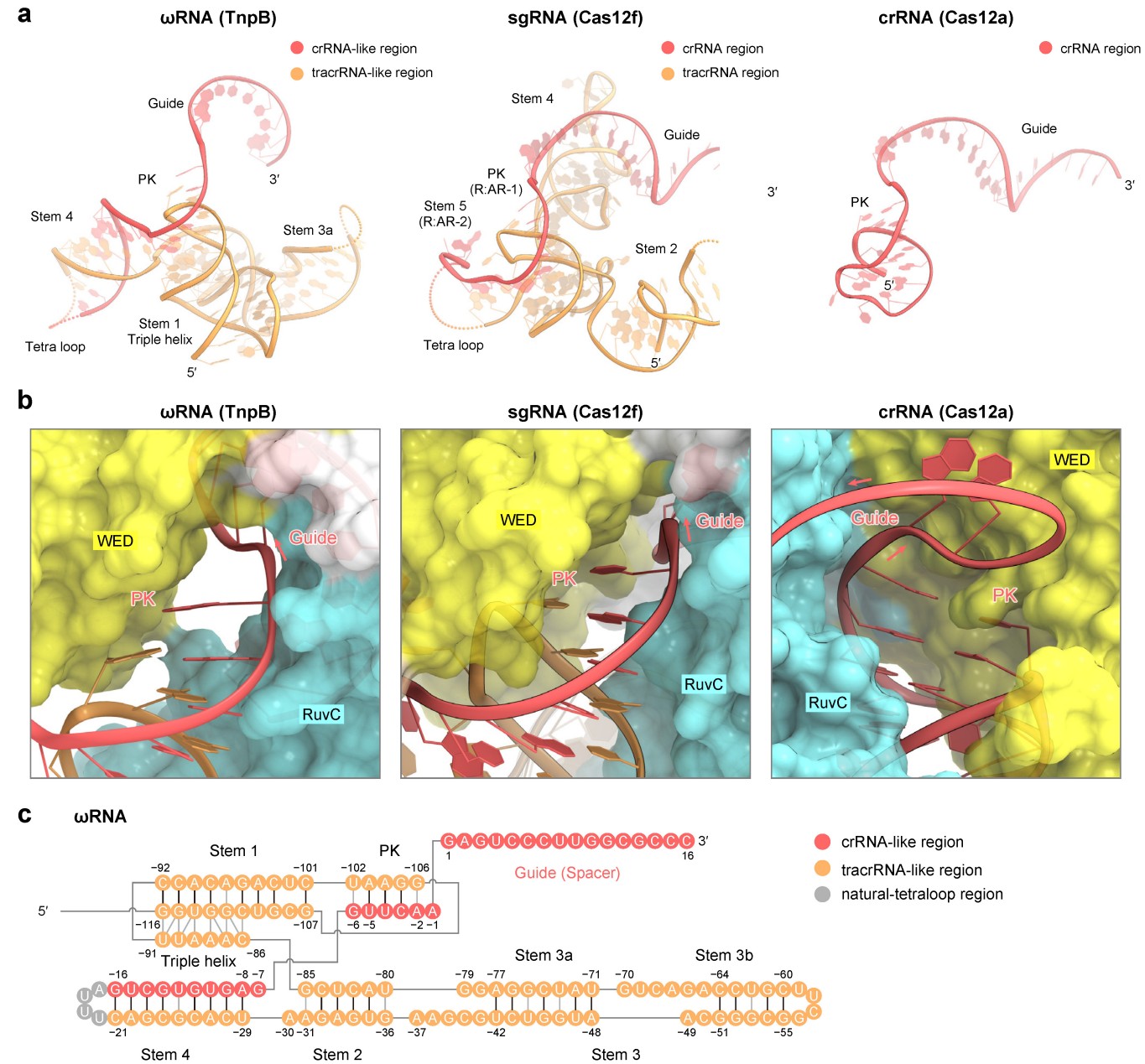

**Extended Data Fig. 7 | ωRNA architecture and recognition. (a)** Structural comparison of the ωRNA scaffold of TnpB with the guide RNA scaffolds of Cas12f (PDB ID: 7C7L) and Cas12a (PDB ID: 6I1K). Cas12f associates with its cognate RNA scaffold formed by crRNA and tracrRNA, whereas Cas12a uses only crRNA. Although these RNAs lack sequence similarity, they contain conserved PK structures. PK, pseudoknot; R:AR, repeat-antirepeat. **(b)** Recognitions of the PK architectures by TnpB, Cas12f, and Cas12a. The PK architectures are recognized by their cognate proteins in similar manners. **(c)** Schematic of the ωRNA. The crRNA-like region, tracrRNA-like region, and natural tetraloop are colored red, orange, and grey, respectively. The disordered regions are enclosed in dashed boxes.

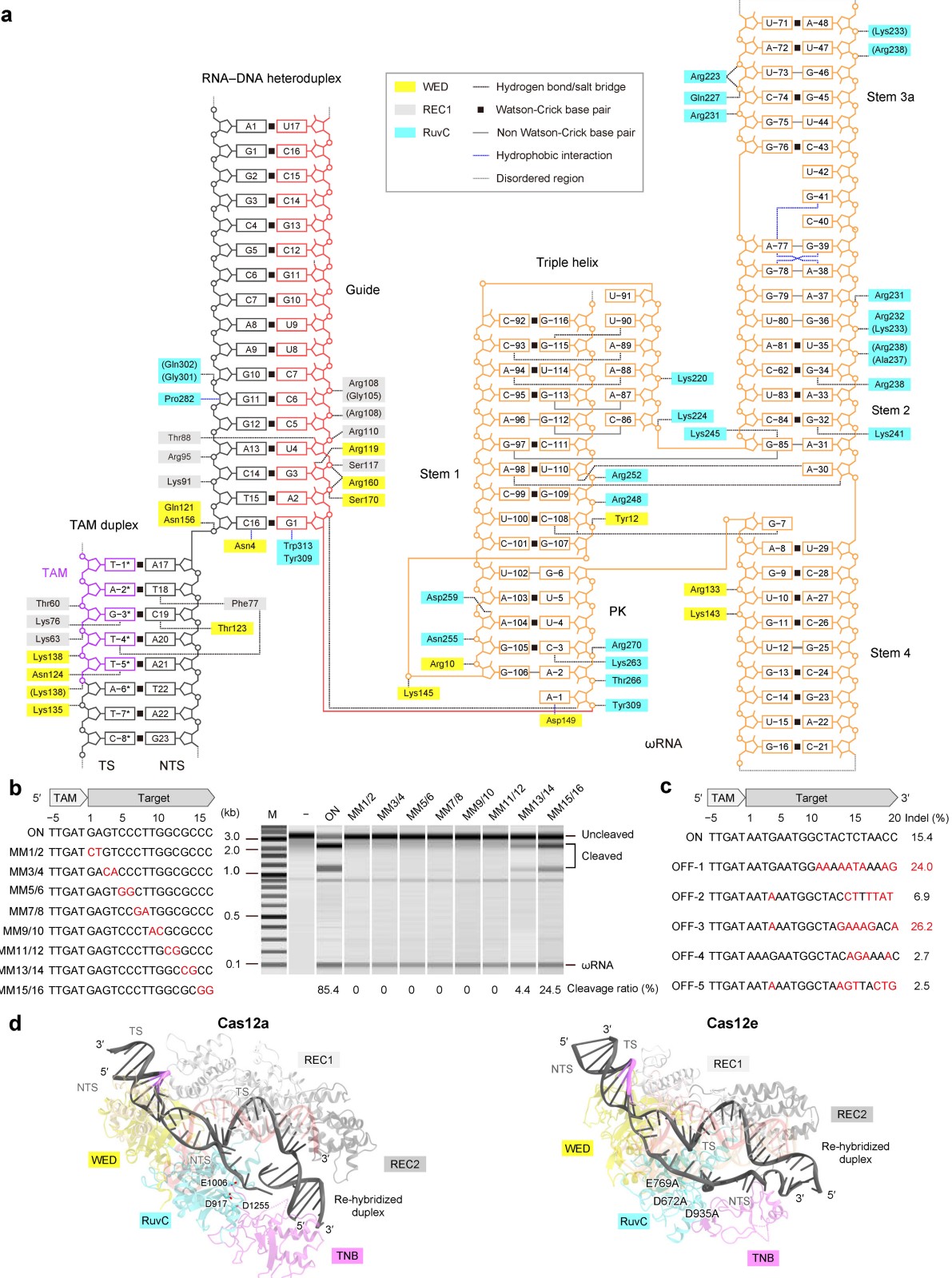

**Extended Data Fig. 8** | See next page for caption.

**Extended Data Fig. 8 | Schematic of nucleic-acid recognition and target DNA cleavage.** (**a**) The residues that interact with the nucleic acids through their main chains are shown in parentheses. The disordered regions are indicated by dashed gray lines. (**b**) Effects of mismatches between the ωRNA guide sequence and the target DNA on TnpB-mediated DNA cleavage. The 3-kb linearized target DNA, containing a 16-nt target sequence or 2-nt mismatches at positions 1–16, was incubated with the TnpB–ωRNA complex (250 nM) at 50 °C for 30 min. The reaction products were resolved, visualized, and quantified with a MultiNA microchip electrophoresis device (SHIMADZU). Data are mean ± s.d.

($n$ = 3, biologically independent samples). The experiments were repeated three times with similar results. Source data are provided as a Source data file. (**c**) Indel formation induced by TnpB at the on-target (AGBL1 gene) and off-target sites in HEK293FT cells. ON, on-target site; OFF, off-target site. (**d**) Positions of the active sites and the target DNAs of Cas12a (Cas12a from *Francisella novicida*) (PDB ID: 6I1K) and Cas12e (Cas12e from Deltaproteobacteria, also known as CasX) (PDB ID: 6NY2). The re-hybridized DNA duplexes are recognized by the TNB domain, thereby facilitating the DNA unwinding and loading into the RuvC active site.

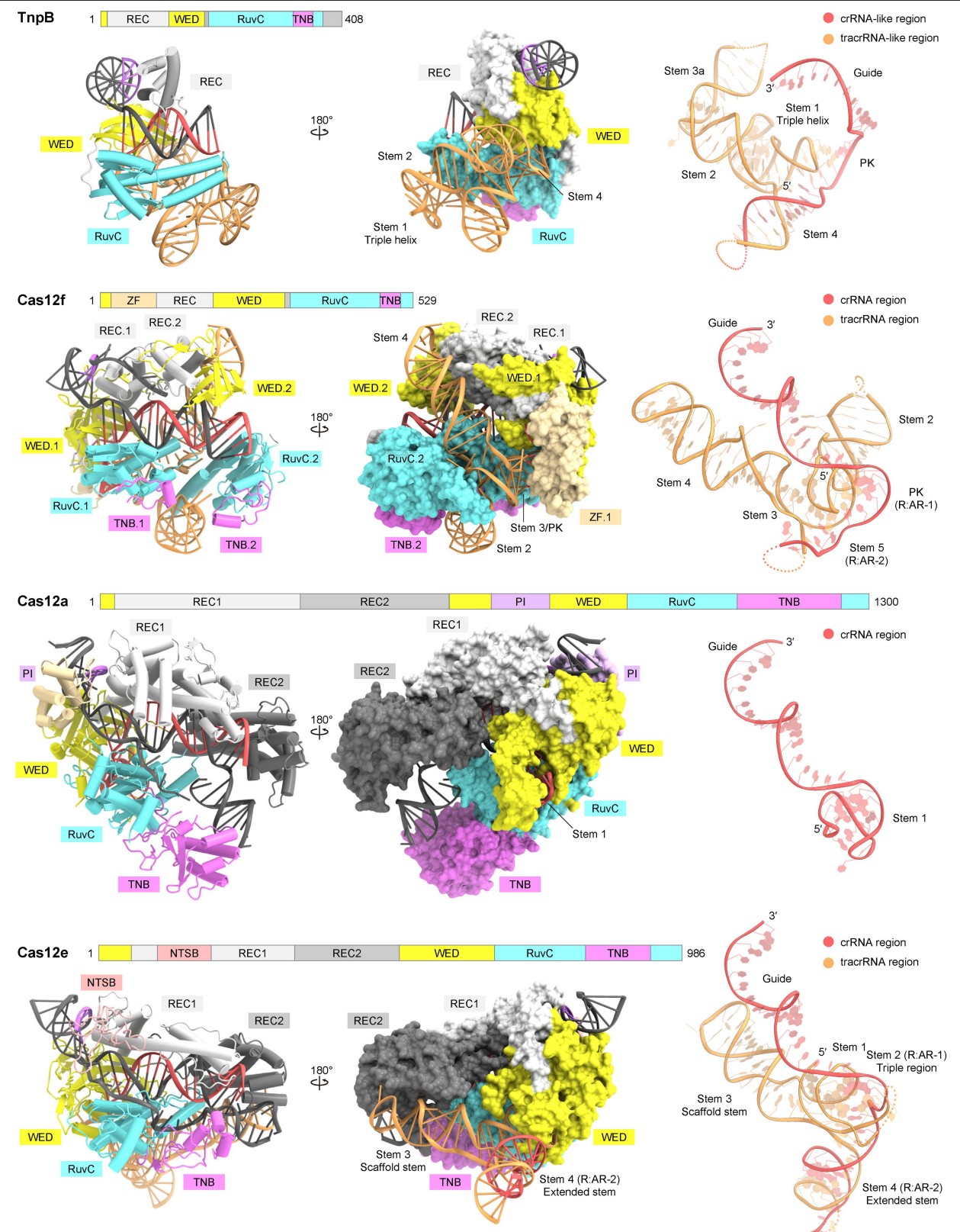

**Extended Data Fig. 9 | Comparison of TnpB with diverse type V CRISPR-Cas12 enzymes.** Structural comparison of TnpB with Cas12f (Cas12f from an uncultured archaeon) (PDB ID: 7C7L), Cas12a (Cas12a from *Francisella novicida*) (PDB ID: 6I1K), and Cas12e (Cas12e from Deltaproteobacteria, also known as CasX) (PDB ID: 6NY2).

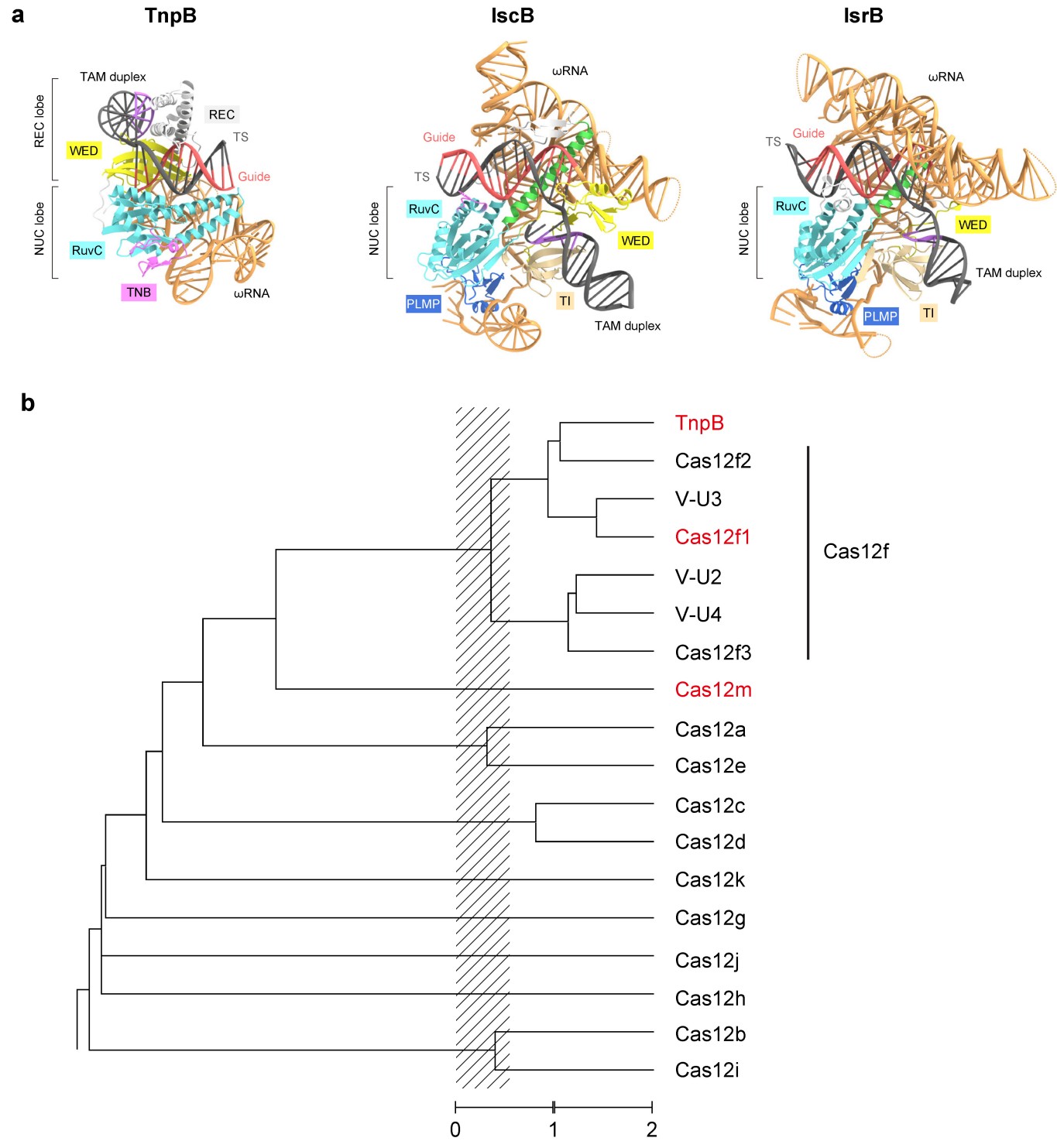

**Extended Data Fig. 10 | Comparison of TnpB with IscB and IsrB, and UPGMA dendrogram of Cas12 enzymes.** (**a**) Structural comparison of TnpB with IscB (IscB from the human gut metagenome) (PDB ID: 7UTN) and IsrB (IsrB from *Desulfovirgula thermocuniculi*) (PDB ID: 8DMB). (**b**) UPGMA dendrogram showing similarities between different families of Type V effectors. The dendrogram was built using the UPGMA (unweighted pair group method with arithmetic mean) method and is based on the matrix of HHalign[39] scores calculated for all against all pairwise alignments, with length coverage of at least 33%. The alignments for the respective families were taken from a previous report[1], except for the Cas12m family for which an updated alignment (104 proteins) was used. The striped rectangle corresponds to the tree depth D between 1.5 and 2 (D = 2 roughly corresponds to the pairwise HHsearch similarity score of exp(2D) ≈ 0.02 relative to the self-score), and reflects the tree depth where the subtype assignment is uncertain and a subject for additional consideration.

**Extended Data Table 1 | Cryo-EM data collection, refinement, and validation statistics**

| Data collection and processing | |
| --- | --- |
| Sample | TnpB–ωRNA–DNA |
| EMDB ID | EMD-34428 |
| PDB ID | 8H1J |
| Microscope | Titan Krios G3i |
| Detector | Gatan K3 Camera |
| Magnification | 105,000 |
| Voltage (kV) | 300 |
| Electron exposure (e⁻/Å²) | 50.0 |
| Defocus range (μm) | −0.8 to −1.6 |
| Pixel size (Å/px) | 0.83 |
| Symmetry imposed | $C1$ |
| Number of movies | 3570 |
| Initial particle images | 2,136,853 |
| Final particle images | 98,483 |
| Global resolution (Å) | |
| FSC 0.143 (masked/unmasked) | 3.21/3.82 |
| FSC 0.5 (masked/unmasked) | 3.99/6.25 |
| Resolution range (Å) | 2.96–7.95 |
| Map-sharpening $B$ factor (Å²) | −50 |
| **Model building and refinement** | |
| Refinement package | PHENIX |
| **Model composition** | |
| Protein atoms | 2,931 |
| Nucleic acid atoms | 2,842 |
| Ligands | 1 |
| **Model refinement** | |
| Model-Map CC ($CC_{mask}/CC_{box}/CC_{peaks}/CC_{volume}$) | 0.80/0.75/0.61/0.79 |
| Resolution (Å) by model to map FSC threshold 0.5 (masked/ unmasked) | 3.42/3.56 |
| Average $B$ factor (Å²) (protein/RNA/metal ion) | 127.00/161.48/277.62 |
| R.M.S. deviations from ideal | |
| Bond lengths (Å) | 0.005 |
| Bond angles (°) | 0.521 |
| **Validation** | |
| MolProbity score | 1.99 |
| CaBLAM outliers (%) | 0.85 |
| Clashscore | 8.21 |
| Rotamer outliers (%) | 5.19 |
| Cβ outliers (%) | 0.00 |
| EMRinger score | 2.02 |
| 3DFSC sphericity | 0.836 |
| Ramachandran plot | |
| Favored (%) | 98.32 |
| Allowed (%) | 1.68 |
| Outlier (%) | 0 |

The structure validation was performed using MolProbity in the PHENIX package. The EMRinger score and 3DFSC sphericity were calculated by PHENIX and the 3DFSC processing Server (https://3dfsc.salk.edu/upload/info/), respectively.

# Reporting Summary

## Statistics

For all statistical analyses, confirm that the following items are present in the figure legend, table legend, main text, or Methods section.

| n/a | Confirmed | |
|---|---|---|
| ☐ | ☒ | The exact sample size ($n$) for each experimental group/condition, given as a discrete number and unit of measurement |
| ☐ | ☒ | A statement on whether measurements were taken from distinct samples or whether the same sample was measured repeatedly |
| ☒ | ☐ | The statistical test(s) used AND whether they are one- or two-sided *Only common tests should be described solely by name; describe more complex techniques in the Methods section.* |
| ☒ | ☐ | A description of all covariates tested |
| ☒ | ☐ | A description of any assumptions or corrections, such as tests of normality and adjustment for multiple comparisons |
| ☐ | ☒ | A full description of the statistical parameters including central tendency (e.g. means) or other basic estimates (e.g. regression coefficient) AND variation (e.g. standard deviation) or associated estimates of uncertainty (e.g. confidence intervals) |
| ☒ | ☐ | For null hypothesis testing, the test statistic (e.g. $F$, $t$, $r$) with confidence intervals, effect sizes, degrees of freedom and $P$ value noted *Give P values as exact values whenever suitable.* |
| ☒ | ☐ | For Bayesian analysis, information on the choice of priors and Markov chain Monte Carlo settings |
| ☒ | ☐ | For hierarchical and complex designs, identification of the appropriate level for tests and full reporting of outcomes |
| ☒ | ☐ | Estimates of effect sizes (e.g. Cohen's $d$, Pearson's $r$), indicating how they were calculated |

*Our web collection on statistics for biologists contains articles on many of the points above.*

## Software and code

Policy information about availability of computer code

| Data collection | EPU (version 2.12) |
|---|---|
| Data analysis | cryoSPARC (version 3.3.2), COOT (version 0.9), UCSF ChimeraX (version 1.1.1), CueMol2 (http://www.cuemol.org/ version 2.2.3.443), PHENIX (version 1.20.1). |

For manuscripts utilizing custom algorithms or software that are central to the research but not yet described in published literature, software must be made available to editors and reviewers. We strongly encourage code deposition in a community repository (e.g. GitHub). See the Nature Portfolio guidelines for submitting code & software for further information.

## Data

Policy information about availability of data

All manuscripts must include a data availability statement. This statement should provide the following information, where applicable:
- Accession codes, unique identifiers, or web links for publicly available datasets
- A description of any restrictions on data availability
- For clinical datasets or third party data, please ensure that the statement adheres to our policy

The atomic models have been deposited in the Protein Data Bank under the accession code 8H1J. The cryo-EM density map has been deposited in the Electron Microscopy Data Bank under the accession code EMD-34428.

## Human research participants

Policy information about studies involving human research participants and Sex and Gender in Research.

| | |
|---|---|
| Reporting on sex and gender | n/a |
| Population characteristics | n/a |
| Recruitment | n/a |
| Ethics oversight | n/a |

Note that full information on the approval of the study protocol must also be provided in the manuscript.

# Field-specific reporting

Please select the one below that is the best fit for your research. If you are not sure, read the appropriate sections before making your selection.

☒ Life sciences        ☐ Behavioural & social sciences        ☐ Ecological, evolutionary & environmental sciences

For a reference copy of the document with all sections, see nature.com/documents/nr-reporting-summary-flat.pdf

# Life sciences study design

All studies must disclose on these points even when the disclosure is negative.

| | |
|---|---|
| Sample size | No statistical method was used to determine the sample size. For cryo-EM analyses, sample sizes were determined by the availability of microscope time and the number of particles on electron microscopy grids enough to obtain a structure at the reported resolution. For biochemical analysis, sample size were determined based on the previous reports of this type of study (Kato et al., Nat. Commun. 2022) and the reproducibility of results across independent experiments. |
| Data exclusions | For cryo-EM analyses, particles that did not contribute to improving map quality were excluded following the standard classification procedures in cryoSPARC. This is standard practice for structure determination by cryo-EM. For biochemical analyses, no data was excluded. |
| Replication | Biochemical experiments were performed at least three times. |
| Randomization | For cryo-EM analyses, particles were randomly assigned to half-maps for resolution determination following the standard procedures in cryoSPARC. For biochemical analyses, randomization is not relevant to this study, since no experimental group was assigned in all experiments. |
| Blinding | Blinding is not applicable to this study, since neither structural nor functional experiments included subjective assignments. |

# Reporting for specific materials, systems and methods

We require information from authors about some types of materials, experimental systems and methods used in many studies. Here, indicate whether each material, system or method listed is relevant to your study. If you are not sure if a list item applies to your research, read the appropriate section before selecting a response.

### Materials & experimental systems

| n/a | Involved in the study |
|---|---|
| ☒ | ☐ Antibodies |
| ☐ | ☒ Eukaryotic cell lines |
| ☒ | ☐ Palaeontology and archaeology |
| ☒ | ☐ Animals and other organisms |
| ☒ | ☐ Clinical data |
| ☒ | ☐ Dual use research of concern |

### Methods

| n/a | Involved in the study |
|---|---|
| ☒ | ☐ ChIP-seq |
| ☒ | ☐ Flow cytometry |
| ☒ | ☐ MRI-based neuroimaging |

# Eukaryotic cell lines

Policy information about cell lines and Sex and Gender in Research

| | |
|---|---|
| Cell line source(s) | HEK293FT cell line was obtained from ThermoFisher Scientific (R70007). |
| Authentication | The HEK293FT cell line was not authenticated as it was purchased commercially. |
| Mycoplasma contamination | The HEK293FT cell line was not tested for mycoplasma contamination. |
| Commonly misidentified lines (See ICLAC register) | HEK293FT is not in the ICLAC database of misidentified cell lines. |

