## [Peer Review File · Nature]

Manuscript Title: Cryo-EM structure of the transposon-associated TnpB enzyme

Reviewer Comments & Author Rebuttals

Reviewer Reports on the Initial Version:

Referees' comments:

Referee #1:

We ask reviewers the following questions, to provide an assessment of the various aspects of a manuscript: Please summarise what you consider to be the outstanding features of the work.

The authors determined a structure of the ω RNA-guided ISDra2 (TnpB) protein bound to target DNA. The authors perform structure-guided mutations in TnpB and the target DNA to reveal how this RNA-guided nuclease binds and cuts DNA in a programmable manner.

The authors make the following claims and conclusions:

- 1) ω RNA adopts a structure that was not expected based on its primary sequence.
- 2) The structure and structure-guided mutations reveals how TnpB recognizes ω RNA and its target DNA, providing insights into the DNA cleavage mechanism.
- 3) A structural comparison of TnpB with Cas12 enzymes suggests an “evolutionary path from TnpB to larger type V effectors.

However, text explaining how well the atomic model fits the density, seems to be missing. In particular, density for regions of the structure critical to the authors claims and conclusions should be modeled into the density so that the reader can verify what the authors claim. There is a lack of validation statistics provided for the cryo-EM map and its interpretation and the PDB validation report appears to reveal unmodelled electron density that is not explained in the text. Conversely, protein has appeared to have been built into a region of the map for which there is no density. Further, the biochemical data for the structure-guided mutants is noisy enough to make a conclusion about the residues of interest difficult to evaluate (i.e., one replicate of a reaction shows zero cleavage activity, and the next shows >50% cleavage). What conclusion is to be drawn from this and what explains that type of variability between reps? Finally, one of the Cas12 enzyme structures that they compare TnpB to is unpublished. In places, the text and legends are missing information necessary to understand the experiments, the data, and interpretations. The structure is important but the work presented here is not sufficiently to critically evaluate the paper.

- **Validity:** Does the manuscript have flaws which should prohibit its publication? If so, please provide details.

The cryo-EM and biochemical methodology lack sufficient detail and rigor. Details are provided below.

- **Originality and significance:** If the conclusions are not original, please provide relevant references. On a more subjective note, do you feel that the results presented are of immediate interest to many people in your own discipline, and/or to people from several disciplines?

The work is original and significant, but seemly premature in places.

- **Data & methodology:** Please comment on the validity of the approach, quality of the data and quality of presentation. Please note that we expect our reviewers to review all data, including any extended data and supplementary information. Is the reporting of data and methodology sufficiently detailed and transparent to enable reproducing the results?

Main Figure concerns:

The electron density map around modelled regions of interest critical to conclusions of the paper should be displayed in the main text or extended data figure (i.e., for Fig 1d, 2f, 2g, 2h, 3b, 3d, 3e). Showing the reader how the model and density fit together is fundamental.

Fig. 1 and 2. It looks like most of the omega RNA, target DNA and some of the TnpB protein could not be resolved? That should be made clear in Fig 1, where the atomic model is shown. A scheme to show what was included in the complex that was imaged, and what portions of this complex could not be resolved would be helpful.

Fig. 1b. It would be helpful to include the length of the guide RNA and target DNA in the legend or figure. Fig. 2b and 2c. The legends are flipped.

Fig. 3c. Five TnpB mutants were tested for their cleavage activity. For three TnpB mutants, two replicate reactions showed no cleavage, but one replicate from these reactions showed ~20-40% cleavage. For the last TnpB mutant, one replicate showed no cleavage, whereas another showed >50% cleavage. What mechanism could result in such a range of cleavage activities between replicates? This data does not support the authors claim that certain TnpB mutations “abolished” DNA cleavage activity”. How was the cleavage ratio calculated? Please update the methods with relevant information throughout. Gels or other results used to generate cleaves data must be provided.

Fig. 4. The structure of TnpB is compared to Cas12f (PDB: 7C7L) and to Cas12m (unpublished). The reader cannot determine the validity of a comparison made to an unpublished structure.

Extended Data concerns:

Extended Data Fig. 2. Please indicate what fractions were used for cryo-EM. “*the peak fraction*” is ambiguous. No fractions are shown, and there are two peaks. One of the peaks has a shoulder. An evaluation of the work requires that the authors include gels (protein and nucleic acid) of the “TnpB– ω RNA–target DNA complex” that was imaged by cryo-EM. Do all species of heterogeneous mixture of ω RNA (Fig. 2c) support TnpB protein binding to dsDNA?

Extended Data Fig. 2e. The legend indicates this is a “Direct distribution plot”. Is this synonymous with “Viewing distribution plot” or “Angular distribution”?

Extended Data Fig. 5a. Please define the mixture of solid and dotted lines (top). Please provide total number and positions of the RNA are deleted in the “ $\Delta 5'$ region”. The figure suggests that entire 3' is ordered but the text suggests otherwise. Please clarify. A detailed legend would be most helpful.

Extended Data Fig. 9. What does the scale with numbers 0, 1 and 2, found below the phylogenetic tree, represent?

Extended Data Table 1. Include additional reconstruction, refinement and validation statistics to help assess the quality of the cryo-EM map and model. Please add Global resolution at FSC=0.143 and 0.5 (unmasked and masked), Resolution range, 3DFSC sphericity, Refinement package used (this is included in the method but should be here to for ease-of-access), Cross Correlation of the map with the model, MolProbity score, Poor rotamers (%), C-beta deviations (%), CaBLAM Outliers (%), and EMRinger score. Several of these validation practices have been previously recommended in the following 2021 Nature Protocols paper: <https://www.nature.com/articles/s41592-020-01051-w> .

Extended Data Table 1. The EMDB ID and PDB ID rows appear to be switched. Presumably the PDB ID is 8H1J, and the EMDB ID is EMD-34428.

A Preliminary Validation Report was submitted. However, these reports state it is “Not For Manuscript Review”. This report has a couple limitations and should not be submitted to journals (https://www ww pdb.org/validation/FAQs#what_to_submit). The report produced at the annotation stage after a PDB has been issued includes a pink watermark that states “For Manuscript Review”. Please include a valid report.

The PDB validation report shows that there are regions of the electron density map that have not been modelled and regions of the model that appear to have been built without density. This is concerning and has not been made clear in the text.

Methods concerns:

Lines 415-416 state “For structural studies, the target DNA was purchased from Eurofin Genomics.”. Please explicitly state whether the DNA was pre-ordered as a duplex, or if two single strands were ordered and annealed *in vitro*. If the latter, please describe the annealing protocol. Further, the sequences of the target DNA strands are not provided in the main text or extended data, prohibiting the reproduction of these results.

Line 427 states “The purified complex solution (A260 = 5) was applied to...”. This is an odd way to report the concentration. The extinction coefficient for the complex and the path length of the device used to measure the A260 of the sample, should be used to calculate the molar concentration so that the readers can understand how much sample was applied to the grids. Further why is the A260 alone reported? The complex is composed of both protein and nucleic acid. The reader cannot discern the ratio of nucleic acid to protein from the A260 alone.

Line 467 also reports the concentration of the TnpB-ωRNA complex as an A260 value, which prevents the reader from replicating this experiment, as described above.

Line 473, “Except for the *in vitro* cleavage, experiments were performed at least three times.” This sentence is found in the methods subsection titled “*In vitro* DNA cleavage assay”. It might be clearer to state that the *in vitro* cleavage was performed once (which is a guess because the quoted sentence), and experiments X, Y and Z were performed at least three times. If this interpretation is correct, then what is the meaning of the 3 dots plotted in Figure 3C? Again, the legend for that figure is non-descriptive and does not mention the 3 dots shown for each mutant and what the 3 dots represent.

The sequences of the target DNAs are not included in the main text or the extended data. Their exclusion prohibits readers from reproducing these results.

Mutations of the target DNA “were introduced by a PCR-based method, and sequences were confirmed by DNA sequencing” (Lines 416-417). Consider providing the sequencing data.

- Appropriate use of statistics and treatment of uncertainties: All error bars should be defined in the corresponding figure legends; please comment if that’s not the case. Please include in your report a specific comment on the appropriateness of any statistical tests, and the accuracy of the description of any error bars and probability values.

This is impossible to accurately evaluate. See above comments.

- Conclusions: Do you find that the conclusions and data interpretation are robust, valid and reliable? The presented data lacks rigor, which precludes an assessment of validity or reliability.
- Suggested improvements: Please list additional experiments or data that could help strengthening the work in a revision.

There appears to be an orientation bias of the ω RNA-TnpB-DNA complex, which may be limiting the resolution and quality of the density map. Although additional validation parameters (e.g., 3DFSC sphericity) would help with the assessment. A collection of additional data at a tilted angle could improve the structure. The cryoEF tool (<https://www.mrc-lmb.cam.ac.uk/crusso/cryoEF/>) can assist in measuring the impact of particle orientation on the structure, and in suggesting optimal tilts angles for data collection.

The *in vitro* DNA cleavage assays are too noisy to draw conclusions. Further the methods are confusing. The “amount” of ω RNA-TnpB complex used in the assays is reported as A260 5. Therefore the concentration of complex used, or the proportion of RNA:protein cannot be discerned. Further it is not clear what steps were performed in triplicate. There is a general lack of rigor that precludes objective peer-review.

- References: Does this manuscript reference previous literature appropriately? If not, what references should be included or excluded?

Consider citing the authors that discovered this RNA (Weinberg et al).

- Clarity and context: Is the abstract clear, accessible? Are abstract, introduction and conclusions appropriate?

The text is not clear. Aside from the lack of rigor, there are many incorrect uses of tense and prepositions. The data do not sufficiently support the conclusions.

- Inflammatory material: Does the manuscript contain any language that is inappropriate or potentially libelous?

no.

- Springer Nature is committed to diversity, equity and inclusion; please raise any concerns that may in your view have an impact on this commitment.

none.

Referee #2:

Nakagawa and Hirano et al. describe the first cryo-EM structure of TnpB, an RNA-guided DNA endonuclease that is thought to be an evolutionary ancestor of Cas12 nucleases. TnpB, along with two other transposon-associated RNA-programmable endonucleases, IscB and IsrB, were reported last year (ref. 14). These proteins are of great interest because they provide insight into the extensive evolution of Cas12 nucleases, and also because they are extremely compact and could potentially be useful tools for genome editing. Recent studies have reported the structure of the IscB and IsrB. Thus, this manuscript provides the remaining structural piece in understanding the relationship between these proteins and various well-studied Cas12 proteins.

The authors describe the structure of TnsB and the associated omegaRNA. The structural comparisons between TnsB and Cas12 proteins are quite striking, and the structures are very well illustrated to show these similarities. These figures make a compelling argument of the structural evolution from TnsB to Cas12 variants, which contain analogous features with insertions that may extend the functionality of Cas12 as an immune effector.

The authors also functionally characterize some of the structural features, including important residues for recognition of a target-adjacent motif (TAM), which is analogous to a PAM for Cas12, and for the lack of observed interactions between TnsB and the TAM-distal region of the target-guide RNA duplex. The authors test off-target indel formation in the presence of TnsB, and observe a high rate of indel formation at sites with very low homology to the guide RNA in the TAM-distal region. This result is interesting, but preliminary, and could be further validated as described below.

Overall, this is an excellent manuscript that describes a very interesting structure and begins to characterize the functional mechanism of TnsB. I have some concerns about presentation in a few places, and feel that the result on TAM-distal mismatch targeting could be supported by further experiments. I hope the authors can address the following concerns.

Major concerns:

1. Line 142-143: The authors speculate that TnpB may process omegaRNA, but there is not really any evidence to support this. It is possible that any cellular RNase may have cleaved/degraded RNA that is not protected by the protein during expression of the RNP. This caveat should be included, or the authors could provide experimental evidence that TnpB has RNA processing activity. This would likely require purification of TnpB in the absence of omegaRNA, which may not be possible.
2. The promiscuity of TnpB, as demonstrated in Fig. 3f, is interesting and somewhat surprising, in comparison to the relatively high fidelity of Cas12 effectors. Have the authors tested this type of cleavage in vitro? While indel frequency suggests that TnpB is highly tolerant of TAM-distal mismatches, a direct observation of cleavage of such targets would more precisely define this tolerance.
3. Regarding the cleavage of TAM-distally mismatched targets, the cleavage mechanism that the authors propose seems to suggest that DNA unwinding of the TS and NTS would have to proceed into the TAM-distal region, to expose the cut sites 21- or 15- to 21-nt upstream of the TAM and present ssDNA in the RuvC active site. It is unclear how DNA unwinding might occur for some of the off-targets observed in Fig. 3f, given that there is very little capacity for the guide RNA to base pair with the TAM-distal region of these targets. The additional in vitro cleavage assays suggested above may clarify whether these sequences can be cut efficiently. If this is the case, do the authors have an explanation for how TnpB may expose cleavage sites in the absence of complementary base pairing

between the guide RNA and the TAM-distal region of the TS strand? Does the target get cleaved in the same location when such mismatches are present?

4. The authors mention the structure of IscB in the Discussion, but there is no comparison of this structure with TnpB. The authors could include a figure comparing TnpB with IscB and the recently published structure of IsrB (PMID: 36224386).

5. Lines 68-70: The authors mention that TAM is required for to “cleave and target” the DNA, and that PAM is required to avoid self-targeting by Cas12. This wording is unclear. The current model is that PAM recognition by a Cas effector is required to initiate DNA unwinding and facilitate strand invasion by the crRNA. The lack of a PAM in the CRISPR allows Cas effectors to differentiate self and non-self, but this is not a function of the PAM, but rather the lack of a PAM in the CRISPR repeat. Presumably, the TAM plays a similar role to the PAM, in facilitating TnpB binding to dsDNA by helping to initiate unwinding. The authors do not talk extensively about DNA unwinding in the manuscript, but this is an important concept in light of some of the results mentioned above.

Minor concerns:

1. Lines 155-177: Can the authors include a figure showing the RNA structural model fit in the EM density for the RNA, perhaps as part of ED Fig. 6? It would be helpful to see the density for the triple helix and the disordered stem 3 region.

2. Lines 128-129, “The C-terminal region (residues 376-408) relatively low sequence homology...”: Is there a word missing in this sentence?

3. Lines 213-214: The interactions between Y52 and S56 and the -1 residue of the TAM are described as van der Waals interactions. From the angle shown in Fig. 3b, it looks like the hydroxyl groups of Y52 and S56 are potentially interacting with phosphates in the DNA backbone (assuming there would be a phosphate bonded to the 3'-oxygen). Should these be described as hydrogen bonds or ion-dipole interactions rather than van der Waals interactions?

4. In ED Fig. 7, the TAM is called PAM. Please update.

5. Lines 218-221: The wording of this sentence is a bit unclear. Is the interaction that is described not actually observed in the EM density map, and therefore not built in the structural model? If it is part of the structural model, can it be included in a figure? If there is not EM density for this region, how was the interaction that is described between the -4 and -5 nucleotides and the P122-N126 loop inferred?

Referee #3:

The manuscript reports the structure of a member of the TnpB superfamily of transposon-associated proteins from the well-studied ISDra2 mobile element. TnpB proteins are frequently encoded in IS200/IS605 insertion sequences (or nonautonomous versions of the element), but long mysterious because they play no direct role in transposition of the genetic element. Interest in these enzymes has increased dramatically with the appreciation that they are guide RNA-directed endonucleases, explaining why they are related to the diverse Cas12 effector proteins. It was also recently appreciated that TnpB proteins very likely act as a type of addiction system that breaks the chromosome following loss of the element - breaking the chromosome will prompt recombinational repair from a sister chromosome thereby reestablishing the lost element.

Structural information helps clarify the biology of TnpB proteins and could help with future work engineering small and efficient effectors based on this information. However, the paper provides no context about the extensive literature with the ISDra2 element and instead focuses on the potential functional relationship to contemporary Cas12 effectors. This makes for a confused and misguided analysis because it assumes that TnpB and Cas12 proteins evolved to do the same thing which they did not. This analysis is further confused when it talks about intermediate forms of the protein as though the sole “purpose” of TnpB proteins were to provide an eventual role as immune systems or genome editing tools. This misrepresentation of TnpB leads to multiple critical flaws in logic throughout the paper (some of these are indicated below).

The authors need to first discuss and analyze the functional characteristics of TnpB considering its role in IS200/IS605 elements. This is the only meaningful way to discuss the structure. Many readers will likely also be interested in insights into how this information can be used for engineering new compact gene editors. However, they will not be served by conflating the evolutionary forces guiding the evolution of TnpB proteins with that of Cas12 immunity effector proteins.

Specific considerations

Line 54 (and elsewhere): The last formalized Cas12 protein subgroup is Cas12k, pending a comprehensive phylogenetic analysis of the newer suggested subgroups.

Lines 68-70, “In addition, TnpB requires...sequence to avoid self-targeting”: This is incorrect; the TnpB TAM site is not used “...just as Cas enzymes...” to avoid self-targeting. The motif is analogous, but the function is not the same because the role of the TnpB is to cause self-targeting once the element is lost. This sentence should also have references for the TAM system.

Lines 70-74, “TnpB from...evolutionary intermediate between TnpB and larger Cas12 enzymes.”: “Evolutionary intermediate” is the incorrect term because it suggests it was a mid-step in evolution. The authors probably mean that it shows features common to TnpB and larger Cas12 proteins, but as stated is incorrect.

Line 88: “REC” and “NUC” should be “REC lobe” and “NUC lobe” as they are listed in the figure.

Lines 131-135, “The truncation mutant of the...target DNA cleavage.”: Sentence needs to be clarified to explain what is meant by ‘yield’ and how the low-resolution readout in ED Fig. 5c supports the finding. The appropriate active site mutant control is missing, and it is unclear if anything significant can be assessed from the variability across the lanes.

Lines 141-144, “These results indicated...TnpB may possess RNA processing activity”: Processing could have occurred during purification from a host endonuclease or by a direct activity of TnpB. Was the cleaved RNA also found with the active site mutants?

Lines 147-151: Need to clarify what is meant by “...., the co-evolution of these two elements is less constrained than previously predicted escaping overlap of functionally essential gene regions.”

Lines 249-250, “This feature could provide an advantage for the transposon to integrate its own sequence into ectopic sites, although further studies are needed to clarify the role of TnpB in transposition.”: The extensive literature with IS200/IS605 indicates there is no role for TnpB in transposition. What is known about the transposition process should be explained and referenced as it is crucial to put the findings with TnpB in the correct context. After this point is made it might be

interesting to discuss how this relates to the application in human cells with off-site cutting.

Lines 283-287, "Furthermore, the structural...thought to be intermediates on the evolutionary path..": As mentioned above, this kind of discussion does not make sense because it assumes that the TnpB proteins and Cas12 effector proteins are under the same evolutionary pressures.

Referee #4:

In this manuscript, Nakagawa et al. reported a 3.1Å cryo-EM structure of the TnpB enzyme complexed with ω RNA and a short dsDNA. ω RNA is transcribed from the C-terminal region of TnpB gene and associated with TnpB to cleave dsDNA upon TAM recognition. The crystal structure reveals that 1) the full-length ω RNA contains several disordered region, particularly at the 5'-region and stem 3; 2) the CTD of TnpB is also disordered; and 3) the hypercompact protein architecture only recognized the relatively short TAM-proximal target sequence (~10 nt) as a monomer. The cryo-EM data show several important regions as disordered, and the limited structural information is backed up Alpha Fold prediction. Despite these several unique features, TnpB shares too many properties with the previous type V Cas proteins in that: 1) the overall Cas structure is bilobed; 2) a single RuvC is involved in both TS and NTS cleavage; 3) TNB regions is involved in the loading of target DNA strand; and 4) similar to Cas12f1, ω RNA (or gRNA for Cas12f1) contains a KT and disordered stem regions, etc. Although the authors provided some insights of TnpB and other type V CRISPR systems from an evolutionary perspective, this manuscript provides only marginal new findings on the mechanism underlying how this hypercompact TnpB is implicated in the sequence-specific DNA cleavage. Plus, several questions remain that need to be addressed as follows:

Major points:

1. The elucidation of structural features in this manuscript is informative but not sufficient for understanding the TnpB- ω RNA organization. The information is somewhat fragmented and does not integrate well with a focal point to address some of the key issues of the action mechanism. In particular, the structure of this paper does not fully include molecular information on target DNA recognition and DNA cleavage and detailed comparison of how the TnpB is different from other Cas proteins with respect to those molecular actions.
2. The authors claimed that the truncation of the C-terminal region of TnpB is responsible for the substantially decreased formation of TnpB- ω RNA complex. If it is true, the altered complex formation would result in the changes in DNA cleavage from a kinetic or thermodynamic point of view. It's because the complex formation is essential for the DNA cleavage. I am wondering why there's no change in DNA cleavage. This issue needs to be clarified with respect to kinetics and/or thermodynamics.
3. In Fig. 1c, the TnpB architecture indicates that RuvC is separated into two distinct domains. However, all Type V Cas proteins share RuvC domains, which are separated into RuvC I-III. How do you explain this discrepancy? Don't you think that the first long RuvC domain can be divided into two separate domains by, say, a beta-strand? If the RuvC of TnpB consists of two domains, how do you explain this discrepancy from an evolutionary perspective?
4. In Fig. 2c, could you explain why the processed ω RNA shows heterogenous band pattern? Was it that the cleavage shows a partial cleavage activity? In ED Fig. 5c, it appears that only the 15 min incubation is enough for full cleavage of ω RNA, considering that the ω RNA is exclusively seen below a 0.1 kb ladder. More importantly, which domain is involved in the ω RNA processing?

5. In Fig. 2d, the authors claim that Trim1 shows comparable to or higher indel efficiency to TnpB. However, no statistical analysis was conducted on this data and thus it is impossible to assess the significance. Moreover, I don't know whether Trim2 further improves indel efficiency.
6. Related to Fig. 2d, the authors claimed that Stem 3a and 3b are all disordered. Then, I am wondering why the authors did not test the Stem 3a-truncated ω RNA for indel tests with only Stem 3b truncation investigated.
7. In relation to Fig. 2c,d, the authors argue that TnpB cleaves a full-length ω RNA into a truncated one upon binding. Then why is it that the Trim1 ω RNA is responsible for the increased indel efficiency? The full-length ω RNA would be processed into a truncated forms eventually.
8. In Fig. 3c, the mutant-cleavage data were presented as triplicated experiments, but the data points appears to include failed experiments, rather than to show a variation between experiments. The related in vitro data needs to be refined to clarify the involvement of the amino acids in the TAM recognition.
9. For the off-target experiments in Fig. 3f, several targets with the TAM-proximal mismatches should be monitored because the data were originally aimed to convince that the TAM-proximal region is essential for on-target recognition.
10. I know TnpB cleaves NTS and TS downstream of the TAM, not upstream of the TAM.
11. The structures identified in this paper provide pretty limited information related to target DNA cleavage. As shown in Fig. 3g, the NTS and TS are distantly placed from the catalytic residues and are mostly invisible due to the disordered property. The authors attempt to explain it by comparing with the structure of Cas12e, but does not rely on clear structural basis, thereby the obtainable information being marginal.

Minor points:

1. In Fig. 2c, what's the 250-nt ladder in the lowest position? Is it a 25-nt band?
2. The referencing is disordered throughout the manuscript. The reference number and sentences need to be properly aligned.
3. Related to ED Fig. 5c, what's the truncation region at the 5'-region of ω RNA? Is it G[-231]-T[-117]? Then it needs to be mentioned in the figure legend.

Author Rebuttals to Initial Comments:

Reviewer #1:

The authors determined a structure of the ω RNA-guided ISDra2 (TnpB) protein bound to target DNA. The authors perform structure-guided mutations in TnpB and the target DNA to reveal how this RNA-guided nuclease binds and cuts DNA in a programmable manner.

The authors make the following claims and conclusions:

- 1) ω RNA adopts a structure that was not expected based on its primary sequence.*
- 2) The structure and structure-guided mutations reveals how TnpB recognizes ω RNA and its target DNA, providing insights into the DNA cleavage mechanism.*
- 3) A structural comparison of TnpB with Cas12 enzymes suggests an “evolutionary path from TnpB to larger type V effectors.*

However, text explaining how well the atomic model fits the density, seems to be missing. In particular, density for regions of the structure critical to the authors claims and conclusions should be modeled into the density so that the reader can verify what the authors claim. There is a lack of validation statistics provided for the cryo-EM map and its interpretation and the PDB validation report appears to reveal unmodelled electron density that is not explained in the text. Conversely, protein has appeared to have been built into a region of the map for which there is no density. Further, the biochemical data for the structure-guided mutants is noisy enough to make a conclusion about the residues of interest difficult to evaluate (i.e., one replicate of a reaction shows zero cleavage activity, and the next shows >50% cleavage). What conclusion is to be drawn from this and what explains that type of variability between reps? Finally, one of the Cas12 enzyme structures that they compare TnpB to is unpublished. In places, the text and legends are missing information necessary to understand the experiments, the data, and interpretations. The structure is important but the work presented here is not sufficiently to critically evaluate the paper.

We thank the reviewer for the comments. As you mentioned, our cryo-EM data sets appeared to have orientation bias, which limited the quality of our density map. Previous studies reported that glow-discharging the grid by adding pentylamine (also known as amylamine) radically changed the particle orientations¹. Furthermore, a recent IscB structural paper reported that the target DNA binds to the active site via a magnesium ion, in a cryo-EM analysis performed using wild-type (catalytically active) IscB and a target DNA with phosphorothioate modifications in the backbone at the cleavage site². To improve the quality of our density map and capture the structure in which the target DNA accesses the RuvC active site, we applied a wild-type TnpB- ω RNA-phosphorothioate-modified target DNA complex to glow-discharged grids with

pentylamine and collected the cryo-EM data set. As a result, we determined the TnpB- ω RNA-target DNA ternary structure at an overall resolution of 3.21 Å. Although the overall resolution is similar to the previous data (at an overall resolution 3.08 Å), the 3DFSC sphericity, which indicates the degree of anisotropy present in the structure, is significantly improved (Fig. L1a), suggesting that our present data sets resolve the orientation bias. Consistent with this statistic, the local resolution of the present density map is improved as compared to that of the previous map (Fig. L1b). In addition, although the TAM-distal DNA remains missing from the present density map due to its intrinsic flexibility, we newly observed densities corresponding to the TNB domain and the loop region involved in the TAM recognition, which we could not detect in the previous density. Thus, we rebuilt the model using the present density map, and discussed the RNA-guided DNA cleavage mechanism of TnpB in detail based on this model.

As you also mentioned, our *in vitro* cleavage data were very noisy, probably because the TnpB protein is so unstable *in vitro* that it gradually aggregates during cleavage experiments.

Therefore, we carefully re-examined and optimised the purification and cleavage conditions, and found that TnpB should be purified in glycerol-containing buffer and cleaves the target DNA more efficiently at a lower salt concentration. We re-purified wild-type TnpB and all the mutants and performed *in vitro* cleavage assays under the optimal condition. These results (shown in Fig. L1c) are very solid, and refined our conclusion about the importance of the mutated residues.

Lastly, our latest paper on the Cas12m structure is under peer review, and structural comparisons with Cas12m could provide quite important insights into Cas12 evolution for readers. Therefore, we believe that this study provides valuable insights for the broad audience of *Nature*. We have addressed the comments raised by the reviewers, as follows.

Fig. L1 | Cryo-EM analysis and *in vitro* cleavage analysis.

(a) Revised (left) and original (right) direction 3DFSC plots calculated by the 3DFSC Processing Server³ (<https://3dfsc.salk.edu/upload/info/>). The global resolution and sphericity of the revised data are improved as compared to those of the original data.

(b) Revised (left) and original (right) local-resolution cryo-EM density maps. The densities corresponding to the TNB domain and TAM duplex (indicated by black circles) are well resolved in the revised density map, as compared to those in the original density map.

(c) *In vitro* DNA cleavage activities of the wild-type TnpB and TAM recognition mutants with the 5' region-deleted ω RNA. The 3 kb linearized target DNA, containing a 16-nt target sequence and a TTGAT TAM sequence, was incubated with the TnpB- ω RNA complex (250 nM) at 37°C for 30 min. The reaction products were resolved, visualized, and quantified with a MultiNA microchip electrophoresis device (SHIMADZU). Data are mean \pm s.d. ($n = 3$). The experiments were repeated three times with similar results. Source data are provided as a Source data file.

MAIN FIGURE CONCERNS:

1. *The electron density map around modelled regions of interest critical to conclusions of the paper should be displayed in the main text or extended data figure (i.e., for Fig 1d, 2f, 2g, 2h, 3b, 3d, 3e). Showing the reader how the model and density fit together is fundamental.*

According to the reviewer's comment, we now show the overall electron density map in Fig. 1d and the densities around the modelled regions crucial for our conclusions in Extended Data Fig. 3 in the revised manuscript.

2. *Fig. 1 and 2. It looks like most of the omega RNA, target DNA and some of the TnpB protein could not be resolved? That should be made clear in Fig 1, where the atomic model is shown. A scheme to show what was included in the complex that was imaged, and what portions of this complex could not be resolved would be helpful.*

Thank you for the helpful comments. Based on the revised density map, the TNB domain (residues 327–359) and stem 3a ([-48]–[-42]–[-71]–[-77]) could be newly resolved. However, residues 281–296 and 379–408 of TnpB, nucleotides (-231)–(-117), (-70)–(-49), (-20)–(-17), and 13–16 of the ω RNA, nucleotides (-8)–4, and 27 of the target DNA (TS), and nucleotides -11* and 1*–24* of the non-target DNA (NTS) remain unresolved. According to the reviewer's suggestion, we have added a brief statement in the main text and figure legend of Fig. 1, and showed the disordered regions of the nucleic acid in Fig. 2a of the revised manuscript.

3. *Fig. 1b. It would be helpful to include the length of the guide RNA and target DNA in the legend or figure.*

Thank you for this comment. The TS and NTS are 35-nt each, and the NTS contains a TTGAT TAM sequence. The 247-nt ω RNA, containing a 16-nt guide segment, was co-expressed and

co-purified with TnpB. According to the reviewer's comments, we have added detailed information about our constructs for cryo-EM analysis in Fig. 1b and its legend, and Supplementary Table 2.

4. Fig. 3c. Five TnpB mutants were tested for their cleavage activity. For three TnpB mutants, two replicate reactions showed no cleavage, but one replicate from these reactions showed ~20-40% cleavage. For the last TnpB mutant, one replicate showed no cleavage, whereas another showed >50% cleavage. What mechanism could result in such a range of cleavage activities between replicates? This data does not support the authors claim that certain TnpB mutations "abolished" DNA cleavage activity". How was the cleavage ratio calculated? Please update the methods with relevant information throughout. Gels or other results used to generate cleaves data must be provided.

Thank you for these insightful comments. As the reviewer pointed out, our *in vitro* cleavage data in Fig. 3c in the original manuscript were generally less reliable, probably due to two issues. First, as described above, since the TnpB protein is very unstable *in vitro* and gradually aggregates during cleavage experiments, the wild-type TnpB cleaved only ~15% of the target DNA at 37°C for 30 min under the original conditions (20 mM HEPES, pH 7.5, 100 mM KCl, 2 mM MgCl₂, 1 mM DTT, and 5% glycerol). Second, as you mentioned in the Methods concern and Suggested improvement sections, we calculated the concentration of the TnpB-ωRNA complex using the A₂₆₀ value, which may not be accurate because of the total concentration of the protein-nucleic-acid complex. Therefore, we re-examined and optimised the purification and cleavage conditions, and found that TnpB can be stably purified in glycerol-containing buffer and cleaves the target DNA much more efficiently at a lower salt concentration. Furthermore, we used the Bradford method to accurately measure the concentration of the TnpB-ωRNA complex. Finally, we re-performed the *in vitro* cleavage assay in the following manner. The TnpB-ωRNA complex (2 μL, final 250 nM) was mixed with the 3 kb linearized plasmid target containing a 16-nucleotide target sequence and a TTGAT TAM sequence (8 μL, 100 ng), and incubated at 37°C for 30 min in 10 μL low salt buffer (20 mM HEPES, pH 7.5, 50 mM KCl, 2 mM MgCl₂, 1 mM DTT, and 5% glycerol). The reaction products (2 μL) were stopped by the addition of 6 μL quench buffer, containing EDTA (20 mM final concentration) and Proteinase K (40 ng). The reaction products were resolved, visualized, and quantified with a MultiNA microchip electrophoresis device (SHIMADZU). The cleavage ratio was calculated by the molar concentration of cleaved product (1 kb) / the molar concentration of

cleaved product (1 kb) + that of un-cleaved product (3 kb). The *in vitro* cleavage experiments were performed three times. These results (shown in Fig. L1c) are much more reliable and clarified our conclusion about the importance of functional residues. According to the reviewer's comments, we have added the detailed information of our *in vitro* cleavage experiment in the Methods section and these results in the revised manuscript (Fig. 4d). Furthermore, the raw cleavage data have been uploaded as Source data.

5. Fig. 4. The structure of TnpB is compared to Cas12f (PDB: 7C7L) and to Cas12m (unpublished). The reader cannot determine the validity of a comparison made to an unpublished structure.

Our latest paper on the Cas12m structure is under peer review, and comparisons of these structures could provide important insights for readers.

EXTENDED DATA CONCERNS:

1. Extended Data Fig. 2. Please indicate what fractions were used for cryo-EM. "the peak fraction" is ambiguous. No fractions are shown, and there are two peaks. One of the peaks has a shoulder. An evaluation of the work requires that the authors include gels (protein and nucleic acid) of the "TnpB- ω RNA-target DNA complex" that was imaged by cryo-EM. Do all species of heterogenous mixture of ω RNA (Fig. 2c) support TnpB protein binding to dsDNA?

According to your helpful comments, we have indicated the peak fraction used for cryo-EM analysis by a black line below the trace, and added the SDS-PAGE and urea-PAGE gels in Extended Data Fig. 2a in the revised manuscript. In addition, to understand which part of the ω RNA sequence remains bound to the TnpB protein, we performed northern blotting analyses using five DNA probes (I-V) covering the entire ω RNA sequence (Fig. L2a). We did not observe any RNA band in a total RNA extract from cells only expressing ω RNA, strongly suggesting that the ω RNA was degraded by endogenous RNases in the absence of the TnpB protein (Fig. L2b). In contrast, in a total RNA extract from the ω RNA and TnpB co-expressing cells, three different RNAs, ~220-, ~160-, and ~130-nt, were detected (Fig. L2b). The ~220-nt RNA and the ~160-/~130-nt RNAs were detected by probes II-V and probes III-V, respectively, indicating that these RNAs commonly contain the guide sequences on their 3' ends. The ~160-/~130-nt RNAs were also observed in the RNA extracted from the purified TnpB- ω RNA complex, whereas the ~220-nt RNA was barely detectable, suggesting that the ~220-nt RNA

was degraded during the purification process or urea gel electrophoresis. We have added these northern blotting results in the revised manuscript (Extended Data Fig. 5b, c).

Fig. L2 | Biochemical characterization of ω RNA.

(a) The sequence of full-length ω RNA (G[−231]–C16). The probe positions used in the northern blotting analysis are shown in orange.

(b) Northern blotting of ω RNA. Total RNAs prepared from *E. coli* wild-type cells (lane 1), ω RNA-expressing cells (lane 2), ω RNA and MBP co-expressing cells (lane 3), ω RNA and TnpB co-expressing cells (lane 4), the *in vitro* transcript of ω RNA (lane 5), and the ω RNA extracted from the purified TnpB- ω RNA complex (lane 6) were resolved by 10% denaturing PAGE and stained with GelGreen (left panel) or subjected to northern blotting (right panels, Probes I–V). The 6S RNA (180 nt), 5S rRNA (120 nt), tRNAs (76–93 nt) in the total RNA and the 50 nt, 100 nt, and 300 nt RNA markers are indicated (lane M).

2. Extended Data Fig. 2e. The legend indicates this is a “Direct distribution plot”. Is this synonymous with “Viewing distribution plot” or “Angular distribution”?

Yes, the “Direct distribution plot” means the same as “Viewing distribution plot”. The term “Direct distribution plot” is commonly used in cryo-EM data analyses performed with the cryoSPARC software platform.

3. *Extended Data Fig. 5a. Please define the mixture of solid and dotted lines (top). Please provide total number and positions of the RNA are deleted in the “ $\Delta 5'$ region”. The figure suggests that entire 3' is ordered but the text suggests otherwise. Please clarify. A detailed legend would be most helpful.*

Thank you for the helpful comments. The solid and dotted lines indicate the ordered and disordered regions, respectively. As the reviewer pointed out, in addition to the CTD domain (residues 379–408) and the 5' region ([–231]–[–117]), Stem 3b ([–70]–[–49]) and tetraloop ([–20]–[–17]) are disordered. In the $\Delta 5'$ region mutant, we deleted nucleotides (–231)–(–117) in the ω RNA (total 115 nucleotides). According to the reviewer's comments, we have modified the previous Extended Data Fig. 5a into the new Extended Data Fig. 6a, and added detailed information in the legend.

4. *Extended Data Fig. 9. What does the scale with numbers 0, 1 and 2, found below the phylogenetic tree, represent?*

The dendrogram was built using the UPGMA (unweighted pair group method with arithmetic mean) method, and the scale means the HHalign score calculated for all against all pairwise alignments.

5. *Extended Data Table 1. Include additional reconstruction, refinement and validation statistics to help assess the quality of the cryo-EM map and model. Please add Global resolution at FSC=0.143 and 0.5 (unmasked and masked), Resolution range, 3DFSC sphericity, Refinement package used (this is included in the method but should be here to for ease-of-access), Cross Correlation of the map with the model, MolProbity score, Poor rotamers (%), C-beta deviations (%), CaBLAM Outliers (%), and EMRinger score. Several of these validation practices have been previously recommended in the following 2021 Nature Protocols paper: <https://www.nature.com/articles/s41592-020-01051-w>.*

According to the reviewer's comments, we have added the refinement and validation statistics to Supplementary Table 1, and made a brief statement about model refinement and validation in the Methods section in the revised manuscript. The 3DFSC sphericity was calculated by the 3DFSC processing Server³ (<https://3dfsc.salk.edu/upload/info/>).

6. *Extended Data Table 1. The EMDB ID and PDB ID rows appear to be switched. Presumably the PDB ID is 8H1J, and the EMDB ID is EMD-34428.*

We fixed them in the revised manuscript.

7. *A Preliminary Validation Report was submitted. However, these reports state it is “Not For Manuscript Review”. This report has a couple limitations and should not be submitted to journals (https://www.wwpdb.org/validation/FAQs#what_to_submit). The report produced at the annotation stage after a PDB has been issued includes a pink watermark that states “For Manuscript Review”. Please include a valid report.*

We have submitted a Validation Report for Manuscript Review.

8. *The PDB validation report shows that there are regions of the electron density map that have not been modelled and regions of the model that appear to have been built without density. This is concerning and has not been made clear in the text.*

Thank you for the insightful comments. There were relatively poor densities for the peripheral regions of the complex molecule, such as the loop region (residues 122–126) in the REC domain, the $\alpha 1$ helix (residues 214–230) in the RuvC domain, and Stem 3a ([–48]–[–42]–[–71]–[–77]) in the ω RNA scaffold in the original electron density map, due to the orientation bias. However, in the revised density map, we observed clear densities for these regions, and thus have built the model. Nonetheless, residues 281–296 and 379–408 of TnpB, nucleotides (–231)–(–117), (–70)–(–49), (–20)–(–17), and 13–16 of the ω RNA, nucleotides (–8)–4 and 27 of the TS, and nucleotides –11* and 1*–24* of the NTS remain unresolved, indicating the intrinsic flexibilities of these regions. According to the reviewer’s suggestions, we have added a brief statement about our model in the main text of the revised manuscript.

METHODS CONCERNS:

1. *Lines 415-416 state “For structural studies, the target DNA was purchased from Eurofin Genomics.”. Please explicitly state whether the DNA was pre-ordered as a duplex, or if two single strands were ordered and annealed in vitro. If the latter, please describe the annealing protocol. Further, the sequences of the target DNA strands are not provided in the main text or extended data, prohibiting the reproduction of these results.*

Thank you for the comments. We purchased two single-stranded DNAs from Eurofin Genomics. A double-stranded DNA was prepared by annealing a 35-nt target DNA strand and a 35-nt non-target strand containing a TTGAT TAM at 95°C for 2 min. According to the reviewer's comments, we have added a brief statement about the sample preparation for the structural analysis in the Methods section, and detailed information about the DNA sequences used for the cryo-EM analysis in Supplementary Table 2 in the revised manuscript.

2. Line 427 states “The purified complex solution ($A_{260} = 5$) was applied to...”. This is an odd way to report the concentration. The extinction coefficient for the complex and the path length of the device used to measure the A_{260} of the sample, should be used to calculate the molar concentration so that the readers can understand how much sample was applied to the grids. Further why is the A_{260} alone reported? The complex is composed of both protein and nucleic acid. The reader cannot discern the ratio of nucleic acid to protein from the A_{260} alone.

According to the reviewer's comments, we have changed the statement “The purified complex solution ($A_{260} = 5$) was applied to ...” to “The purified complex solution (0.5 mg/ml final concentration) was applied to ...” in the revised manuscript.

3. Line 467 also reports the concentration of the TnpB- ω RNA complex as an A_{260} value, which prevents the reader from replicating this experiment, as described above.

Thank you for the helpful comments. To precisely evaluate the TnpB protein concentration in the sample, we used the Bradford method. Specifically, we measured the concentrations of wild-type TnpB or TnpB mutants based on a calibration curve prepared using a dilution series of BSA proteins. According to the reviewer's suggestions, we have added information about the measurement of the TnpB- ω RNA complex concentrations in the Methods section of the revised manuscript.

4. Line 473, “Except for the in vitro cleavage, experiments were performed at least three times.” This sentence is found in the methods subsection titled “In vitro DNA cleavage assay”. It might be clearer to state that the in vitro cleavage was performed once (which is a guess because the quoted sentence), and experiments X, Y and Z were performed at least three times. If this interpretation is correct, then what is the meaning of the 3 dots plotted in Figure 3C? Again, the

legend for that figure is non-descriptive and does not mention the 3 dots shown for each mutant and what the 3 dots represent.

We are sorry that the main text was incorrect. We performed the *in vitro* cleavage reactions at least three times, and the three dots shown for each mutant in Fig. 3c in the original manuscript represented the cleavage ratio, based on the wild-type TnpB, for each of the three *in vitro* cleavage assays. As described above, we performed the *in vitro* cleavage assays again under the modified conditions, using re-purified constructs, at least three times. The three dots shown in Fig. 4d also represented the cleavage ratio for each of the three *in vitro* cleavage assays. We have replaced Fig. 3c in the original manuscript with the new Fig. 4d and have modified the statement about the *in vitro* cleavage assay in the Methods section of the revised manuscript.

5. The sequences of the target DNAs are not included in the main text or the extended data. Their exclusion prohibits readers from reproducing these results.

According to the reviewer's comments, we have added the target DNA sequence to Supplementary Table 2 in the revised manuscript.

6. Mutations of the target DNA "were introduced by a PCR-based method, and sequences were confirmed by DNA sequencing" (Lines 416-417). Consider providing the sequencing data.

We have added information about the primers used in this study in Supplementary Table 3 in the revised manuscript.

SUGGESTED IMPROVEMENTS:

1. There appears to be an orientation bias of the ω RNA-TnpB-DNA complex, which may be limiting the resolution and quality of the density map. Although additional validation parameters (e.g., 3DFSC sphericity) would help with the assessment. A collection of additional data at a tilted angle could improve the structure. The cryoEF tool (<https://www.mrc-lmb.cam.ac.uk/crusso/cryoEF/>) can assist in measuring the impact of particle orientation on the structure, and in suggesting optimal tilts angles for data collection.

Thank you very much for the insightful comments. As you mentioned, the 3DFSC sphericity calculated by the 3DFSC processing Server³ revealed that the original data sets had a terrible

orientation bias (Fig. L1a), limiting the quality of the density map. Previous studies reported that adding pentylamine (also known as amylamine) to the glow-discharged grid resulted in a radical change in the particle orientation¹. To improve the quality of our density map, we applied the wild-type TnpB- ω RNA-target DNA complex to glow-discharged grids with pentylamine and collected the cryo-EM data set. As a result, we determined the TnpB- ω RNA-target DNA ternary structure at an overall resolution of 3.21 Å. Although the overall resolution is similar to that of the previous structure (at an overall resolution 3.08 Å), the 3DFSC sphericity is significantly improved, as compared to that of the previous data (Fig. L1a), suggesting that our present data set resolved the orientation bias problem. Consistent with this statistic, the local resolution of the present density map is apparently improved as compared to that of the previous map (Fig. L1b). Thus, we rebuilt the model using the revised density map, and have replaced the TnpB- ω RNA-target DNA ternary structure with the new one in the original manuscript. We have added detailed information about the cryo-EM analysis to the Methods section, and the refinement and validation statistics to Supplementary Table 1.

2. The in vitro DNA cleavage assays are too noisy to draw conclusions. Further the methods are confusing. The “amount” of ω RNA-TnpB complex used in the assays is reported as A260 5. Therefore the concentration of complex used, or the proportion of RNA:protein cannot be discerned. Further it is not clear what steps were performed in triplicate. There is a general lack of rigor that precludes objective peer-review.

Thank you for the helpful comments. We re-examined and optimised the purification and cleavage conditions, and found that TnpB can be stably purified in glycerol-containing buffer and cleaves the target DNA more efficiently at a lower salt concentration. Furthermore, we used the Bradford method to precisely calculate the concentration of the TnpB- ω RNA complex. Finally, we performed *in vitro* cleavage assays three times in the following manner. The TnpB- ω RNA complex (2 μ L, final 250 nM) was mixed with the 3 kb linearized plasmid target containing a 16-nucleotide target sequence and a TTGAT TAM sequence (8 μ l, 100 ng), and incubated at 37/50°C for 30 min in 10 μ l low salt buffer (20 mM HEPES, pH 7.5, 50 mM KCl, 2 mM MgCl₂, 1 mM DTT, and 5% glycerol). The reaction (2 μ l) was stopped by the addition of 6 μ l quench buffer, containing EDTA (20 mM final concentration) and Proteinase K (40 ng). The reaction products were resolved, visualized, and quantified with a MultiNA microchip electrophoresis device (SHIMADZU). The cleavage ratio was calculated by the molar concentration of cleaved products (1 kb)/the molar concentration of cleaved products (1 kb) +

that of un-cleaved products (3 kb). The *in vitro* cleavage experiments were performed three times. These results shown in Fig. 4d, h and Extended Data Figs. 6b, c and 8c in the revised manuscript are sufficiently reliable to confirm our conclusion. Source data are provided in Source data files.

CLARITY AND CONTEXT:

The text is not clear. Aside from the lack of rigor, there are many incorrect uses of tense and prepositions. The data do not sufficiently support the conclusions.

According to the reviewer's suggestions, we re-performed the structural analysis and the *in vitro* cleavage assays. Furthermore, we have revised the main text and figure legends to clarify our conclusion. Therefore, we believe that this study provides valuable insights for the broad audience of *Nature*.

Reviewer #2:

Nakagawa and Hirano et al. describe the first cryo-EM structure of TnpB, an RNA-guided DNA endonuclease that is thought to be an evolutionary ancestor of Cas12 nucleases. TnpB, along with two other transposon-associated RNA-programmable endonucleases, IscB and IsrB, were reported last year (ref. 14). These proteins are of great interest because they provide insight into the extensive evolution of Cas12 nucleases, and also because they are extremely compact and could potentially be useful tools for genome editing. Recent studies have reported the structure of the IscB and IsrB. Thus, this manuscript provides the remaining structural piece in understanding the relationship between these proteins and various well-studied Cas12 proteins.

The authors describe the structure of TnsB and the associated omegaRNA. The structural comparisons between TnsB and Cas12 proteins are quite striking, and the structures are very well illustrated to show these similarities. These figures make a compelling argument of the structural evolution from TnsB to Cas12 variants, which contain analogous features with insertions that may extend the functionality of Cas12 as an immune effector.

The authors also functionally characterize some of the structural features, including important residues for recognition of a target-adjacent motif (TAM), which is analogous to a PAM for Cas12, and for the lack of observed interactions between TnsB and the TAM-distal region of the

target-guide RNA duplex. The authors test off-target indel formation in the presence of TnsB, and observe a high rate of indel formation at sites with very low homology to the guide RNA in the TAM-distal region. This result is interesting, but preliminary, and could be further validated as described below.

We thank the reviewer for the positive comments. Our responses to the comments raised by the reviewer are as follows.

MAIN CONCERNS:

1. Line 142-143: The authors speculate that TnpB may process omegaRNA, but there is not really any evidence to support this. It is possible that any cellular RNase may have cleaved/degraded RNA that is not protected by the protein during expression of the RNP. This caveat should be included, or the authors could provide experimental evidence that TnpB has RNA processing activity. This would likely require purification of TnpB in the absence of omegaRNA, which may not be possible.

Thank you for the insightful comments. To understand which part of the ω RNA sequence remains bound to the TnpB protein, we performed northern blotting analyses using five DNA probes (I–V) covering the entire ω RNA sequence (Fig. L2a). In a total RNA extract from cells only expressing ω RNA, we did not observe any RNA band, strongly suggesting that the ω RNA was degraded by endogenous RNases in the absence of the TnpB protein (Fig. L2b). In contrast, in a total RNA extract from the ω RNA and TnpB co-expressing cells, three different RNAs, ~220-, ~160-, and ~130-nt, were detected (Fig. L2c). The ~220-nt RNA and the ~160-/~130-nt RNAs were detected by probes II–V and probes III–V, respectively, indicating that these RNAs commonly contain the guide sequences on the 3' end. The ~160-/~130-nt RNAs were also observed in the RNA extracted from the purified TnpB– ω RNA complex, whereas the ~220-nt RNA was hardly observed, suggesting that the ~220-nt RNA was degraded during the purification process or urea gel electrophoresis (Fig. L2b). Next, we isolated the ~160-nt and ~130-nt RNAs, and performed a Liquid Chromatography-Mass spectrometry (LC/MS) analysis to determine the processing sites of these RNAs. The ~160-nt RNAs treated with RNase A yielded a pGAACp fragment (Fig. L2c), suggesting that this RNA was cleaved between A(–150) and G(–149) or U(–138) and G(–137) by an RNase H-like enzyme, such as TnpB. Unfortunately, we could not identify the cleavage site in the ~130-nt RNA, since the region around (–110)–(–120) is difficult to analyze due to its GU-rich sequence. However, we did

observe a clear density for G(-116), but not T(-117), suggesting that the ~130-nt RNA is cleaved between T(-117) and G(-116). We concluded that the ω RNA is processed by either TnpB or endogenous RNases at multiple sites at its 5' end, and that at least a 130 nt fragment including the guide at the 3' end of the ω RNA remains stably bound to the TnpB protein. Nonetheless, as pointed out by the reviewer, we could not determine whether TnpB or the endogenous RNase is involved in the ω RNA processing, since TnpB cannot be purified without ω RNA. Thus, we have modified the statement “TnpB may have ω RNA processing activity, analogous to the crRNA processing activity observed for most Cas12 enzymes” to “the ω RNA is processed by TnpB or endogenous RNases at multiple sites at the 5' end” and added these northern blotting and LC/MS analysis results in Extended Data Fig. 5b–d in the revised manuscript.

Fig. L2 | Biochemical characterization of ω RNA

(a) The sequence of full-length ω RNA (G[−231]–C16). The probe positions used in the northern blotting analysis are highlighted in orange. Two GAAC sites, which could generate the pGAACp fragment by RNase A digestion, are indicated in red.

(b) Northern blotting of ω RNA. Total RNAs prepared from *E. coli* wild-type cells (lane 1), ω RNA-expressing cells (lane 2), ω RNA and MBP co-expressing cells (lane 3), ω RNA and TnpB co-expressing cells (lane 4), the *in vitro* transcript of ω RNA (lane 5), and the ω RNA extracted from TnpB (lane 6) were resolved by 10% denaturing PAGE and stained with GelGreen (left panel) or subjected to northern blotting (right panels, Probes I–V). The 6S rRNA (180 nt), 5S rRNA (120 nt), and tRNAs (76–93 nt) in the total RNA, and the 50 nt, 100 nt, and 300 nt RNA markers are indicated (lane M).

(c) Collision-induced dissociation spectrum of pGAACp fragment from ω RNA digested by RNase A. The divalent negatively charged ion of pGAACp was used as the precursor ion for CID. The product ions in the CID spectrum are assigned on the sequence.

2. The promiscuity of TnpB, as demonstrated in Fig. 3f, is interesting and somewhat surprising, in comparison to the relatively high fidelity of Cas12 effectors. Have the authors tested this type of cleavage *in vitro*? While indel frequency suggests that TnpB is highly tolerant of TAM-distal mismatches, a direct observation of cleavage of such targets would more precisely define this tolerance.

Thank you for the insightful comments. According to the reviewer's suggestions, we performed *in vitro* cleavage assays, using the TnpB- ω RNA complex and the double-mismatch-containing DNA targets, at 50°C for 30 min. TAM-proximal mismatches (positions 1–12) abolished the TnpB-mediated target DNA cleavage, whereas TAM-distal mismatches (positions 13–16) allowed the target DNA cleavage by TnpB (Fig. L3). These results indicate that the first ~12 nucleotides that guide RNA–target DNA heteroduplex formation are essential for the target DNA cleavage by TnpB, which is consistent with our structural and *in vivo* analyses. According to the reviewer's suggestions, we have added these results in Extended Data Fig. 8b in the revised manuscript.

Fig. L3 | Effects of mismatches

Effects of mismatches between the ω RNA guide sequence and target DNA on TnpB-mediated DNA cleavage. The 3 kb linearized target DNA, containing a 16-nt target sequence (ON) or 2-nt mismatches at positions 1–16 (MM1/2–MM15/16), was incubated with the TnpB- ω RNA complex (250 nM) at 50°C for 30 min. The reaction products were resolved, visualized, and quantified with a MultiNA microchip electrophoresis device (SHIMADZU). Data are mean \pm

s.d. ($n = 3$). The experiments were repeated three times with similar results. Source data are provided as a Source data file.

3. Regarding the cleavage of TAM-distally mismatched targets, the cleavage mechanism that the authors propose seems to suggest that DNA unwinding of the TS and NTS would have to proceed into the TAM-distal region, to expose the cut sites 21- or 15- to 21-nt upstream of the TAM and present ssDNA in the RuvC active site. It is unclear how DNA unwinding might occur for some of the off-targets observed in Fig. 3f, given that there is very little capacity for the guide RNA to base pair with the TAM-distal region of these targets. The additional in vitro cleavage assays suggested above may clarify whether these sequences can be cut efficiently. If this is the case, do the authors have an explanation for how TnpB may expose cleavage sites in the absence of complementary base pairing between the guide RNA and the TAM-distal region of the TS strand? Does the target get cleaved in the same location when such mismatches are present?

Thank you for the insightful comment. Cas12 enzymes recognize the PAM-distal region of the guide RNA-target DNA heteroduplex and the re-hybridized DNA duplex by their TNB domains, which facilitate the DNA unwinding and loading into the RuvC active site^{4,5} (Fig. L4a). In contrast, in the TnpB structure, the TAM-distal region of the heteroduplex is located far from the TNB domain, and the end of the TAM-distal region of the heteroduplex and the re-hybridized DNA duplex are disordered (Fig. L4b). These observations suggested that TnpB, unlike Cas12 enzymes, does not interact with these regions. Thus, we hypothesized that TnpB is unable to unwind the re-hybridized DNA by itself, and instead traps the spontaneously unwound DNA in the positively charged pocket formed by the TNB domain and the RuvC active site. To test this hypothesis, we performed an *in vitro* cleavage assay with five target DNAs with different sequences at the site of the DNA duplex that should be re-hybridized (Fig. L4c). We found that TnpB cleaves the target DNAs with AT-rich sequences more efficiently than those with GC-rich sequences (Fig. L4c). These results support our hypothesis that TnpB spontaneously cleaves the unwound target DNA, rather than that the target DNA was unwound by TnpB itself. We have added these results in Fig. 4h, and proposed the target DNA loading mechanism by TnpB, which is distinct from those of Cas12 enzymes, in the revised manuscript.

Fig. L4 | Target DNA loading mechanism.

(a) Positions of active sites and target DNAs of Cas12a (Cas12a from *Francisella novicida*) (PDB ID: 6I1K) and Cas12e (Cas12e from Deltaproteobacteria, also known as CasX) (PDB ID: 6NY2). The re-hybridized DNA duplexes are recognized by the TNB domain, thereby facilitating the DNA unwinding and loading into the RuvC active site.

(b) Positions of the active site and the target DNA of TnpB. The possible trajectories of the TS and NTS are shown by a dashed arrow.

(c) *In vitro* DNA cleavage activities of TnpB with five target DNAs with different sequences at the re-hybridized DNA duplex. Re-hybridized regions with altered sequences are highlighted with a yellow background. CG and AT sequences are coloured blue and green, respectively. Data are mean \pm s.d. ($n = 3$). The experiments were repeated three times with similar results. Source data are provided as a Source data file.

4. The authors mention the structure of *IscB* in the Discussion, but there is no comparison of this structure with TnpB. The authors could include a figure comparing TnpB with *IscB* and the recently published structure of *IsrB* (PMID: 36224386).

Thank you for the important comments. Accordingly, we compared the ternary structure of TnpB with those of IscB (PDB: 7XHT) and IsrB (PDB: 8DMB)^{2,6}. These proteins share neither sequence nor structural similarity, except for the RuvC domains, and their cognate ω RNA scaffolds lack sequence and structural similarities. The TnpB protein adopts a bilobed architecture consisting of the REC and NUC lobes, whereas the IscB and IsrB proteins lack REC lobes, and instead, a part of their ω RNA plays the equivalent role of the REC lobe. We have added statements about the structural comparisons of TnpB with IscB and IsrB in the revised manuscript and the new Extended Data Fig. 10a.

5. Lines 68-70: The authors mention that TAM is required for to “cleave and target” the DNA, and that PAM is required to avoid self-targeting by Cas12. This wording is unclear. The current model is that PAM recognition by a Cas effector is required to initiate DNA unwinding and facilitate strand invasion by the crRNA. The lack of a PAM in the CRISPR allows Cas effectors to differentiate self and non-self, but this is not a function of the PAM, but rather the lack of a PAM in the CRISPR repeat. Presumably, the TAM plays a similar role to the PAM, in facilitating TnpB binding to dsDNA by helping to initiate unwinding. The authors do not talk extensively about DNA unwinding in the manuscript, but this is an important concept in light of some of the results mentioned above.

Thank you for the helpful comments. As the reviewer pointed out, the PAM plays a crucial role in facilitating the target DNA unwinding and the guide RNA–target DNA heteroduplex formation, rather than avoiding self-targeting by a Cas effector. A structural comparison of TnpB and the Cas12 enzymes revealed that the TAM/PAM duplexes and the guide RNA–target DNA heteroduplexes are bound in similar positions and orientations relative to their cognate proteins, demonstrating that the TAM plays a similar role to the PAM, facilitating the target DNA unwinding by TnpB. According to the reviewer’s comments, we have modified the statement about the TAM function in TnpB.

MINOR CONCERNS:

1. Lines 155-177: Can the authors include a figure showing the RNA structural model fit in the EM density for the RNA, perhaps as part of ED Fig. 6? It would be helpful to see the density for the triple helix and the disordered stem 3 region.

Thank you for the comments. According to the reviewer's suggestions, we now show the densities for Stem 3 and the triple helix regions in Extended Data Fig. 3d, e, respectively, in the revised manuscript.

2. Lines 128-129, “The C-terminal region (residues 376-408) relatively low sequence homology...”: Is there a word missing in this sentence?

Thank you for the comments. We have changed the statement “The C-terminal region (residues 376–408) relatively low sequence homology among TnpB proteins, ...” to “The C-terminal domain (CTD, residues 376–408), which has little sequence homology among TnpB proteins, is disordered in the present structure except for its first three residues” in the revised manuscript.

3. Lines 213-214: The interactions between Y52 and S56 and the -1 residue of the TAM are described as van der Waals interactions. From the angle shown in Fig. 3b, it looks like the hydroxyl groups of Y52 and S56 are potentially interacting with phosphates in the DNA backbone (assuming there would be a phosphate bonded to the 3'-oxygen). Should these be described as hydrogen bonds or ion-dipole interactions rather than van der Waals interactions?

Thank you for the helpful comments. The 5'-methyl group of dT(-1*) in the TAM forms van der Waals interactions with the side chains of Y52 and S56 in the REC domain (Fig. L5). To avoid confusion, we have replaced Fig. 4b (the previous Fig. 3b) with a new one, which presents space-filling models of the nucleobase of dT(-1*) and the side chains of Y52 and S56 from different angles.

Fig. L5 | Recognition of dT(-1*).

Nucleotide dT(-1*) and residues Y52 and S56 are depicted by space-filling models.

4. In ED Fig. 7, the TAM is called PAM. Please update.

We fixed it in the revised manuscript.

5. Lines 218-221: The wording of this sentence is a bit unclear. Is the interaction that is described not actually observed in the EM density map, and therefore not built in the structural model? If it is part of the structural model, can it be included in a figure? If there is not EM density for this region, how was the interaction that is described between the -4 and -5 nucleotides and the P122-N126 loop inferred?

Thank you for the insightful comments. In the original electron density map, we could not fully explain the TAM preference of dT(-4*) and dT(-5*), due to the ambiguous densities of a loop region (P122–N126). However, in the revised electron density map, we observed the clear density in this region, and could build the model (Fig. L6a). The 5'-methyl group of dT(-4*) forms van der Waals interactions with the side chains of F77 in the REC domain and T123 in the WED domain (Fig. L6b). The nucleobase of dT(-5*) forms a hydrogen bond with N124 in the WED domain. Indeed, the F77A and N124A mutations substantially reduced, and T123A abolished, the DNA cleavage activity of TnpB, confirming the functional importance of these residues for the recognition of dT(-4*) and dT(-5*) in the TAM sequence (Fig. L6c). According to the reviewer's comments, we have added these results and explanations in the revised manuscript (Fig. 4c, d).

Fig. L6 | TAM recognition.

(a) The cryo-EM density map for the TAM duplex. T123 and N124 in the loop are well resolved.

(b) Recognition of the TAM duplex. Nucleotide dT(-4*) and residues F77 and T123 are depicted by space-filling models. Hydrogen bonds are depicted with green dashed lines.

(c) *In vitro* DNA cleavage activities of the wild-type TnpB and TAM recognition mutants with the 5' region-deleted ω RNA. The 3 kb linearized target DNA, containing a 16-nt target sequence and a TTGAT TAM sequence, was incubated with the TnpB- ω RNA complex (250 nM) at 37°C for 30 min. The reaction products were resolved, visualized, and quantified with a MultiNA microchip electrophoresis device (SHIMADZU). Data are mean \pm s.d. ($n = 3$). The experiments were repeated three times with similar results. Source data are provided as a Source data file.

Reviewer #3:

*The manuscript reports the structure of a member of the TnpB superfamily of transposon-associated proteins from the well-studied ISDra2 mobile element. TnpB proteins are frequently encoded in IS200/IS605 insertion sequences (or nonautonomous versions of the element), but long mysterious because they play no direct role in transposition of the genetic element. Interest in these enzymes has incr+**

eased dramatically with the appreciation that they are guide RNA-directed endonucleases, explaining why they are related to the diverse Cas12 effector proteins. It was also recently appreciated that TnpB proteins very likely act as a type of addiction system that breaks the chromosome following loss of the element - breaking the chromosome will prompt recombinational repair from a sister chromosome thereby reestablishing the lost element.

Structural information helps clarify the biology of TnpB proteins and could help with future work engineering small and efficient effectors based on this information. However, the paper provides no context about the extensive literature with the ISDra2 element and instead focuses on the potential functional relationship to contemporary Cas12 effectors. This makes for a confused and misguided analysis because it assumes that TnpB and Cas12 proteins evolved to do the same thing which they did not. This analysis is further confused when it talks about intermediate forms of the protein as though the sole “purpose” of TnpB proteins were to provide an eventual role as immune systems or genome editing tools. This misrepresentation of TnpB leads to multiple critical flaws in logic throughout the paper (some of these are indicated below).

The authors need to first discuss and analyze the functional characteristics of TnpB considering its role in IS200/IS605 elements. This is the only meaningful way to discuss the structure. Many readers will likely also be interested in insights into how this information can be used for engineering new compact gene editors. However, they will not be served by conflating the

evolutionary forces guiding the evolution of TnpB proteins with that of Cas12 immunity effector proteins.

We thank the reviewer for their comments. The role of TnpB was unknown until Altae-Tran *et al.* and Karvelis *et al.* identified TnpB as an RNA-guided DNA endonuclease^{7,8}. Although it is quite important to fully characterise the role of TnpB in transposition, we could not fully address this question due to our technical limitations and it is outside the scope of the current study. Instead, we determined the cryo-EM structure of the TnpB- ω RNA-target DNA complex and revealed how the compact TnpB protein recognizes its cognate ω RNA and cleaves dsDNA targets complementary to the ω RNA guide sequence. These structural findings will contribute to future research on the role of TnpB in transposition.

Previous biochemical and biological studies revealed that CRISPR-Cas12 enzymes have evolved through the integration of TnpB-encoding transposons near CRISPR arrays⁹. Thus, structural comparisons of TnpB with CRISPR-Cas12 enzymes may provide clues as to how CRISPR-Cas12 enzymes have acquired the ability to function as an adaptive immunity against foreign nucleic acids. However, as pointed out by the reviewer, we refrained from using the term “evolutionary intermediate” in the revised manuscript, since TnpB and Cas12f/Cas12m are not intermediates of the larger Cas12 enzymes. In addition, we have addressed the comments raised by the reviewer, as follows.

SPECIFIC CONSIDERATIONS:

1. Line 54 (and elsewhere): The last formalized Cas12 protein subgroup is Cas12k, pending a comprehensive phylogenetic analysis of the newer suggested subgroups.

Thank you for this comment. Since two papers on Cas12l¹⁰ and Cas12m¹¹ were published during this revision, Cas12 effectors are now classified as Cas12a–m. We have added these citations in the revised manuscript.

2. Lines 68-70, “In addition, TnpB requires...sequence to avoid self-targeting”: This is incorrect; the TnpB TAM site is not used “...just as Cas enzymes...” to avoid self-targeting. The motif is analogous, but the function is not the same because the role of the TnpB is to cause self-targeting once the element is lost. This sentence should also have references for the TAM system.

Thank you for the helpful comments. As the reviewer pointed out, TnpB does not avoid self-targeting, but cleaves its own genome after the transposon excision has occurred. Our present structure revealed that TnpB requires the TAM sequence to initiate the target DNA unwinding, just as Cas12 enzymes require the PAM sequence to initiate the target DNA unwinding. According to the reviewer's comments, we have modified the statement about the TAM function in TnpB.

3. Lines 70-74, "*TnpB from...evolutionary intermediate between TnpB and larger Cas12 enzymes.*": "*Evolutionary intermediate*" is the incorrect term because it suggests it was a mid-step in evolution. The authors probably mean that it shows features common to TnpB and larger Cas12 proteins, but as stated is incorrect.

Thank you for mentioning this. As you pointed out, Cas12f and Cas12m are not intermediates of larger Cas12 enzymes. Thus, we refrained from using the term "Evolutionary intermediate" in the revised manuscript.

4. Line 88: "*REC*" and "*NUC*" should be "*REC lobe*" and "*NUC lobe*" as they are listed in the figure.

We fixed them in the revised manuscript.

5. Lines 131-135, "*The truncation mutant of the...target DNA cleavage.*": Sentence needs to be clarified to explain what is meant by 'yield' and how the low-resolution readout in ED Fig. 5c supports the finding. The appropriate active site mutant control is missing, and it is unclear if anything significant can be assessed from the variability across the lanes.

Thank you for the helpful comments. The C-terminal domain (residues 376–408) has relatively low sequence homology among TnpB proteins and is largely disordered (residues 379–408) in our structure, suggesting that this region is not crucial for target DNA cleavage by TnpB. To confirm its function, we prepared the C-terminal domain (residues 376–408)-deletion TnpB mutant (referred to as Δ CTD). The purification yield of Δ CTD in was lower than that of wild-type TnpB, probably because the Δ CTD protein was less stable than wild-type TnpB. To examine the stabilities of these proteins, we performed a thermal shift assay using a NanoTemper Tycho NT.6 Differential Scanning Fluorimeter, which can determine the

inflection temperature (T_i) of samples. The T_i value of the wild-type TnpB- ω RNA complex was 52.3°C, whereas that of the Δ CTD- ω RNA complex was 46.1°C (Fig. L7a), indicating that the truncation of the C-terminal domain reduced the protein stability. Next, to assess the cleavage activity of Δ CTD, we performed *in vitro* cleavage assays using purified TnpB- ω RNA and Δ CTD- ω RNA complexes. We found that Δ CTD exhibited higher cleavage activity than wild-type TnpB (Fig. L7b, c). These results indicated that the C-terminal domain is not crucial for the TnpB-mediated target DNA cleavage, at least under our *in vitro* cleavage assay conditions (20 mM HEPES, pH 7.5, 50 mM KCl, 2 mM MgCl₂, 1 mM DTT, and 5% glycerol, at 37°C for 30 min), despite its involvement in the protein stability. To clarify the function of the C-terminal domain, we have added the results of the thermal shift assay and the *in vitro* cleavage experiment in Extended Data Fig. 6b–d, and modified the statement about the C-terminal domain in the revised manuscript.

Fig. L7 | Biochemical analysis of Δ CTD.

(a) Thermal shift assays of the wild-type TnpB and the Δ CTD mutant, calculated by a NanoTemper Tycho NT.6 Differential Scanning Fluorimeter, which can determine the inflection temperature (T_i) of samples.

(b) *In vitro* DNA cleavage assays of the wild-type (WT) TnpB and the C-terminal domain-deleted (Δ 376–408; Δ CTD) mutant. The linearized plasmid target, containing a 16-nt target sequence and a TTGAT TAM sequence, was incubated with the TnpB- ω RNA complex at 37°C for 30 min. The cleavage products were then analyzed by a MultiNA microchip electrophoresis system.

(c) Quantification of the DNA cleavage data in (b). Data are mean \pm s.d. ($n = 3$). The experiments were repeated three times with similar results. Source data are provided as a Source data file.

6. Lines 141-144, “These results indicated...TnpB may possess RNA processing activity”:
Processing could have occurred during purification from a host endonuclease or by a direct activity of TnpB. Was the cleaved RNA also found with the active site mutants?

Thank you for the insightful comments. To understand the ω RNA processing by TnpB, we performed northern blotting analyses using five DNA probes (I–V) covering the entire ω RNA sequence (Fig. L2a). In a total RNA extract from cells only expressing ω RNA, we did not observe any RNA band, strongly suggesting that the ω RNA was degraded by endogenous RNases in the absence of the TnpB protein (Fig. L2b). In contrast, in a total RNA extract from the ω RNA and TnpB co-expressing cells, three different RNAs, ~220-, ~160-, and ~130-nt, were observed (Fig. L2b). The ~220-nt RNA and the ~160-/~130-nt RNAs were detected by probes II–V and probes III–V, respectively, indicating that these RNAs commonly contain the guide sequences on the 3' end. The ~160-/~130-nt RNAs were also observed in the RNA extracted from the purified TnpB– ω RNA complex, whereas the ~220-nt RNA was barely detected, suggesting that the ~220-nt RNA was degraded during the purification process or urea gel electrophoresis (Fig. L2b). Next, we isolated the ~160-nt and ~130-nt RNAs, and performed Liquid Chromatography-Mass Spectrometry (LC/MS) analyses to determine the processing sites of these RNAs. The ~160-nt RNA treated with RNase A produced a pGAACp fragment (Fig. L2c), suggesting that this RNA was cleaved between A(–150) and G(–149) or U(–138) and G(–137) by an RNase H-like enzyme, such as TnpB. Unfortunately, we could not identify the cleavage site in the ~130-nt RNA, since the region around (–110)–(–120) is difficult to analyze due to its GU-rich sequence. However, we did observe a clear density for G(–116), but not T(–117), suggesting that the ~130-nt RNA is cleaved between T(–117) and G(–116). We concluded that the ω RNA is processed by either TnpB or endogenous RNases at multiple sites at the 5' end, and that at least a 130 nt fragment including the guide at the 3' end of the ω RNA remains stably bound to the TnpB protein. However, we could not determine whether TnpB or an endogenous RNase is involved in the ω RNA processing, since TnpB cannot be purified without ω RNA. We have added these northern blotting and LC-MS analysis results to Extended Data Fig. 5b–d in the revised manuscript.

Fig. L2 | Biochemical characterization of ω RNA.

(a) The sequence of full-length ω RNA (G[−231]–C16). The probe positions used in the northern blotting analysis are highlighted with an orange background. The two GAAC sites that could generate the pGAACp fragment by RNase A digestion are indicated in red.

(b) Northern blotting of ω RNA. Total RNAs prepared from *E. coli* wild-type cells (lane 1), ω RNA-expressing cells (lane 2), ω RNA and MBP co-expressing cells (lane 3), ω RNA and TnpB co-expressing cells (lane 4), the *in vitro* transcript of ω RNA (lane 5), and ω RNA extracted from TnpB (lane 6) were resolved by 10% denaturing PAGE and stained with GelGreen (left panel) or subjected to northern blotting (right panels, Probes I–V). The 6S RNA (180 nt), 5S rRNA (120 nt), and tRNAs (76–93 nt) in the total RNA and 50 nt, 100 nt, and 300 nt RNA markers are indicated (lane M).

(c) Collision-induced dissociation spectrum of the pGAACp fragment from ω RNA digested by RNase A. The divalent negatively charged ion of pGAACp was used as the precursor ion for CID. The product ions in the CID spectrum are assigned on the sequence.

7. Lines 147-151: *Need to clarify what is meant by “...., the co-evolution of these two elements is less constrained than previously predicted escaping overlap of functionally essential gene regions.”*

Previous studies revealed that many archaea and bacteria transcribe non-coding RNAs (ncRNAs) overlapping the 3' end of the *tnpB* gene, suggesting that these ncRNAs play conserved roles in prokaryotes¹². However, our structural and functional analyses revealed that, except for a few nucleotides, the functionally important regions of the *tnpB* gene and ω RNA do not overlap. This finding suggests that the co-evolution of the *tnpB* gene and ncRNAs overlapping the 3' end of *tnpB* gene is less constrained than previously predicted. According to the reviewer's suggestion, we have added an explanation in the revised manuscript.

8. Lines 249-250, *“This feature could provide an advantage for the transposon to integrate its own sequence into ectopic sites, although further studies are needed to clarify the role of TnpB in transposition.”: The extensive literature with IS200/IS605 indicates there is no role for TnpB in transposition. What is known about the transposition process should be explained and referenced as it is crucial to put the findings with TnpB in the correct context. After this point is made it might be interesting to discuss how this relates to the application in human cells with off-site cutting.*

Thank you for the insightful comments. The role of TnpB in transposition was unknown until Altae-Tran *et al.* and Karvelis *et al.* identified TnpB as an RNA-guided DNA endonuclease^{7,8}. Since TnpB targets only its own sequence after transposon excision, Karvelis *et al.* proposed the “copy” mechanism for TnpB. However, our structural and functional analyses revealed that TnpB requires ~12-bp guide RNA–target DNA heteroduplex formation to mediate DNA cleavage with TAM-distal mismatches tolerated, suggesting that TnpB has several target sites in its own host genome. These observations suggested that TnpB may be involved in transposon propagation as well as transposon homing, although further biochemical analyses are needed to fully characterize the function of TnpB in transposition. Thus, we moved the statement “This feature could provide an advantage for the transposon to integrate its own sequence into ectopic

sites, although further studies are needed to clarify the role of TnpB in transposition” from the main text to the Discussion section of the revised manuscript.

9. Lines 283-287, “Furthermore, the structural...thought to be intermediates on the evolutionary path..”: As mentioned above, this kind of discussion does not make sense because it assumes that the TnpB proteins and Cas12 effector proteins are under the same evolutionary pressures

As described above, previous biochemical and biological studies revealed that CRISPR-Cas12 enzymes have evolved through the integration of TnpB-encoding transposons near CRISPR arrays⁹. Thus, structural comparisons of TnpB with CRISPR-Cas12 enzymes may provide clues as to how CRISPR-Cas12 enzymes acquired the ability to function as an adaptive immunity system against foreign nucleic acids. However, as pointed out by the reviewer, we refrained from using the term “evolutionary intermediate” in the revised manuscript, since TnpB and Cas12f/Cas12m are not intermediates of larger Cas12 enzymes.

Reviewer #4:

In this manuscript, Nakagawa et al. reported a 3.1Å cryo-EM structure of the TnpB enzyme complexed with ω RNA and a short dsDNA. ω RNA is transcribed from the C-terminal region of TnpB gene and associated with TnpB to cleave dsDNA upon TAM recognition. The crystal structure reveals that 1) the full-length ω RNA contains several disordered region, particularly at the 5'-region and stem 3; 2) the CTD of TnpB is also disordered; and 3) the hypercompact protein architecture only recognized the relatively short TAM-proximal target sequence (~10 nt) as a monomer. The cryo-EM data show several important regions as disordered, and the limited structural information is backed up Alpha Fold prediction. Despite these several unique features, TnpB shares too many properties with the previous type V Cas proteins in that: 1) the overall Cas structure is bilobed; 2) a single RuvC is involved in both TS and NTS cleavage; 3) TNB regions is involved in the loading of target DNA strand; and 4) similar to Cas12f1, ω RNA (or gRNA for Cas12f1) contains a KT and disordered stem regions, etc. Although the authors provided some insights of TnpB and other type V CRISPR systems from an evolutionary perspective, this manuscript provides only marginal new findings on the mechanism underlying how this hypercompact TnpB is implicated in the sequence-specific DNA cleavage. Plus, several questions remain that need to be addressed as follows:

We thank the reviewer for the important comments. As you mentioned, the overall architecture of TnpB is similar to those of the Cas12 enzymes, which is the structural evidence suggesting that TnpB proteins are ancestors of CRISPR-Cas12 enzymes. Our structure revealed that the single ω RNA consists of crRNA- and tracrRNA-related regions linked by a natural tetra-loop, and adopts an unexpected architecture from its primary sequence, containing a pseudoknot structure, which is conserved among the guide RNAs of Cas12 enzymes. In addition, our structural and functional analyses revealed that, except for a few nucleotides, the functionally important regions of the *tnpB* gene and ω RNA do not overlap, although previous studies suggested that the ω RNA, overlapping the 3' end of the *tnpB* gene, plays conserved roles in prokaryotes¹². Furthermore, our new structure of TnpB, in combination with a new biochemical analysis, suggests that TnpB cleaves the spontaneously unwound target DNA, rather than unwinding the target DNA by itself, which is distinct from Cas12 enzymes. Therefore, we believe that this study provides valuable insights for the broad audience of *Nature*. We have addressed the comments raised by the reviewer as follows.

MAJOR POINTS:

1. The elucidation of structural features in this manuscript is informative but not sufficient for understanding the TnpB- ω RNA organization. The information is somewhat fragmented and does not integrate well with a focal point to address some of the key issues of the action mechanism. In particular, the structure of this paper does not fully include molecular information on target DNA recognition and DNA cleavage and detailed comparison of how the TnpB is different from other Cas proteins with respect to those molecular actions.

Thank you for these comments. Accordingly, we compared the ternary structure of TnpB with those of Cas12 enzymes, and found mechanistic conservations among them. On the one hand, the TnpB and Cas12 enzymes commonly have a bilobed architecture consisting of the REC and NUC lobes, with the guide RNA–target DNA heteroduplex accommodated within the central channel. In addition, they commonly recognize the TAM/PAM duplex in the groove formed by the WED and REC domains, facilitating the initial DNA unwinding and the guide RNA–target DNA heteroduplex formation. On the other hand, the structural comparison also revealed mechanistic differences between the TnpB and Cas12 enzymes for the target DNA loading into the RuvC active site, although they cleave both the TS and NTS at the single RuvC domain. Cas12 enzymes recognize the PAM-distal region of the guide–target heteroduplex and the re-hybridized DNA duplex by their TNB domains, which facilitate the DNA unwinding and

loading into the RuvC active site^{4,5} (Fig. L4a). In contrast, in the TnpB structure, the TAM-distal region of the heteroduplex is located far from the TNB domain, and the end of the TAM-distal region of the heteroduplex and the re-hybridized DNA duplex is disordered (Fig. L4b). These observations suggested that TnpB, unlike Cas12 enzymes, does not interact with these regions. Thus, we hypothesized that TnpB is unable to unwind the re-hybridized DNA by itself, but instead, it traps spontaneously unwound DNA within the positively charged pocket formed by the TNB domain and the RuvC active site. To test this hypothesis, we performed an *in vitro* cleavage assay with five target DNAs with different sequences at the site of the DNA duplex that should be re-hybridized (Fig. L4c). We found that TnpB cleaves target DNAs with AT-rich sequences more efficiently than those with GC-rich sequences (Fig. L4c). These results support our hypothesis that TnpB spontaneously cleaves the unwound target DNA, and does not unwind the target DNA by itself. We have added these results in Fig. 4g, h and Extended Data Fig. 8d, and proposed the target DNA loading mechanism by TnpB, which is distinct from those of Cas12 enzymes, in the revised manuscript.

Fig. L4 | Target DNA loading mechanism.

(a) Positions of active sites and target DNAs of Cas12a (Cas12a from *Francisella novicida* (PDB ID: 6I1K) and Cas12e (Cas12e from Deltaproteobacteria, also known as CasX) (PDB ID:

6NY2). The re-hybridized DNA duplexes are recognized by the TNB domain, thereby facilitating the DNA unwinding and loading into the RuvC active site.

(b) Positions of the active site and the target DNA of TnpB. The possible trajectories of the TS and NTS are shown by a dashed arrow.

(c) *In vitro* DNA cleavage activities of TnpB with five target DNAs with different sequences at the re-hybridized DNA duplex. Re-hybridized regions with altered sequences are highlighted with a yellow background. CG and AT sequences are coloured blue and green, respectively. Data are mean \pm s.d. ($n = 3$). The experiments were repeated three times with similar results. Source data are provided as a Source data file.

2. The authors claimed that the truncation of the C-terminal region of TnpB is responsible for the substantially decreased formation of TnpB- ω RNA complex. If it is true, the altered complex formation would result in the changes in DNA cleavage from a kinetic or thermodynamic point of view. It's because the complex formation is essential for the DNA cleavage. I am wondering why there's no change in DNA cleavage. This issue needs to be clarified with respect to kinetics and/or thermodynamics.

We are sorry that our main text was misleading. We think that the lower yield from the Δ CTD mutant compared to that from the wild-type is caused by the decreased Δ CTD protein stability, rather than the decrease in the interaction between TnpB protein and ω RNA. To examine the stabilities of the Δ CTD- ω RNA complex, we performed a thermal shift assay using a NanoTemper Tycho NT.6 Differential Scanning Fluorimeter, which can determine the inflection temperature (T_i) of samples. The T_i value of the wild-type TnpB- ω RNA complex was 52.3°C, whereas that of Δ CTD- ω RNA complex was 46.1°C (Fig. L7a), indicating that the truncation of the C-terminal domain reduced the protein stability. Next, to assess the cleavage activity of Δ CTD, we performed *in vitro* cleavage assays using purified TnpB- ω RNA and Δ CTD- ω RNA complexes. We found that Δ CTD exhibited higher cleavage activity than wild-type TnpB (Fig. L7b, c). These results indicated that the C-terminal domain is not crucial for the TnpB-mediated target DNA cleavage, at least under our *in vitro* cleavage assay conditions (20 mM HEPES, pH 7.5, 50 mM KCl, 2 mM MgCl₂, 1 mM DTT, and 5% glycerol, at 37°C for 30 min), despite its involvement in the protein stability. To clarify the function of the C-terminal domain, we have added the results of the thermal shift assay and the *in vitro* cleavage experiment in Extended Data Fig. 6b–d and modified the statement about the C-terminal domain in the revised manuscript.

Fig. L7 | Biochemical analysis of Δ CTD.

(a) Thermal shift assays of the WT TnpB and the Δ CTD mutant, calculated by a NanoTemper Tycho NT.6 Differential Scanning Fluorimeter, which can determine the inflection temperature (T) of samples.

(b) *In vitro* DNA cleavage assays of the wild-type (WT) TnpB, the 5' region of the ω RNA-deleted (Δ G[−231]–T[−117]; Δ 5' region) mutant, and the C-terminal domain-deleted (Δ 376–408; Δ CTD) mutant. The linearized plasmid target, containing a 16-nt target sequence and a TTGAT TAM sequence, was incubated with the TnpB– ω RNA complex at 37°C for 30 min. The cleavage products were then analyzed by a MultiNA microchip electrophoresis system.

(c) Quantification of the DNA cleavage data in (b). Data are mean \pm s.d. ($n = 3$). The experiments were repeated three times with similar results. Source data are provided as a Source data file.

3. In Fig. 1c, the TnpB architecture indicates that RuvC is separated into two distinct domains. However, all Type V Cas proteins share RuvC domains, which are separated into RuvC I-III. How do you explain this discrepancy? Don't you think that the first long RuvC domain can be divided into two separate domains by, say, a beta-strand? If the RuvC of TnpB consists of two domains, how do you explain this discrepancy from an evolutionary perspective?

The RuvC domains of both Cas12 enzymes and TnpB share an RNase H fold, consisting of a five-stranded mixed β sheet and four α helices (the conserved α helices (red) and β strands (blue) are numbered in Fig. L8). We consider these conserved four α helices as components of the RNase H fold, although parts of these helices have been defined as other elements in some papers, such as a bridge helix in Cas12e⁵ and a REC domain in Cas12f¹³. Thus, we did not divide the RuvC domain of TnpB into RuvC I-III in our manuscript.

Fig. L8 | Structural comparison of RuvC domains

The RuvC domains comprise an RNase H fold, consisting of a five-stranded mixed β sheet and four α helices, and the conserved α helices (red) and β strands (blue) are numbered.

4. In Fig. 2c, could you explain why the processed ω RNA shows heterogeneous band pattern? Was it that the cleavage shows a partial cleavage activity? In ED Fig. 5c, it appears that only the 15 min incubation is enough for full cleavage of ω RNA, considering that the ω RNA is exclusively seen below a 0.1 kb ladder. More importantly, which domain is involved in the ω RNA processing?

Thank you for these comments. Consistent with the previous research⁸, TnpB could not be purified without its cognate ω RNA. Therefore, we co-expressed and co-purified TnpB with ω RNA. The ω RNA in purified samples showed a heterogeneous band pattern, indicating that it has already been cleaved in *E. coli* cells, rather than during the incubation. To understand the ω RNA processing by TnpB, we performed a northern blotting analysis using five DNA probes (I–V) covering the entire ω RNA sequence (Fig. L2a). In a total RNA extract from cells only expressing ω RNA, we did not observe any RNA band, strongly suggesting that the ω RNA was degraded by endogenous RNases in the absence of the TnpB protein (Fig. L2b). In contrast, in a total RNA extract from the ω RNA and TnpB co-expressing cells, three different RNAs, ~220-, ~160-, and ~130-nt, were detected (Fig. L2c). The ~220-nt RNA and the ~160-/~130-nt RNAs were detected by probes II–V and probes III–V, respectively, indicating that these RNAs commonly contain the guide sequences on the 3' end. The ~160-/~130-nt RNAs were also observed in the RNA extracted from the purified TnpB– ω RNA complex, whereas the ~220-nt RNA was barely detected, suggesting that the ~220-nt RNA was degraded during the purification process or urea gel electrophoresis (Fig. L2b). Next, we isolated the ~160-nt and ~130-nt RNAs and performed a Liquid Chromatography-Mass Spectrometry (LC/MS) analysis

to determine the processing sites of these RNAs. The ~160-nt RNAs treated with RNase A yielded a pGAACp fragment (Fig. L2c), suggesting that this RNA was cleaved between A(-150) and G(-149) or U(-138) and G(-137) by an RNase H-like enzyme, such as TnpB. Unfortunately, we could not identify the cleavage site in the ~130-nt RNA, since the region around (-110)–(-120) is difficult to analyze due to its GU-rich sequence. However, we did observe a clear density for G(-116), but not T(-117), suggesting that the ~130-nt RNA is cleaved between T(-117) and G(-116). We concluded that the ω RNA is processed by either TnpB or endogenous RNases at multiple sites at the 5' end, and that at least a 130 nt fragment including the guide at the 3' end of the ω RNA remains stably bound to the TnpB protein. Nonetheless, we could not determine whether TnpB or an endogenous RNase is involved in the ω RNA processing, since TnpB cannot be purified without ω RNA. We have added these northern blotting and LC-MS analysis results in Extended Data Fig. 5b–d in the revised manuscript.

Fig. L2 | Biochemical characterization of ω RNA

(a) The sequence of full-length ω RNA (G[−231]–C16). The probe positions used in the northern blotting analysis are shown in orange. The two GAAC sites that could generate the pGAACp fragment by RNase A digestion are indicated in red.

(b) Northern blotting of ω RNA. Total RNAs prepared from *E. coli* wild-type cells (lane 1), ω RNA-expressing cells (lane 2), ω RNA and MBP co-expressing cells (lane 3), ω RNA and TnpB co-expressing cells (lane 4), the *in vitro* transcript of ω RNA (lane 5), and the ω RNA extracted from the purified TnpB– ω RNA complex (lane 6) were resolved by 10% denaturing PAGE and stained with GelGreen (left panel) or subjected to northern blotting (right panels, Probes I–V). The 6S RNA (180 nt), 5S rRNA (120 nt), tRNAs (76–93 nt) in the total RNA and the 50 nt, 100 nt, and 300 nt RNA markers are indicated (lane M).

(c) Collision-induced dissociation spectrum of the pGAACp fragment from ω RNA digested by RNase A. The divalent negatively charged ion of pGAACp was used as the precursor ion for CID. The product ions in the CID spectrum are assigned on the sequence.

5. In Fig. 2d, the authors claim that Trim1 shows comparable to or higher indel efficiency to TnpB. However, no statistical analysis was conducted on this data and thus it is impossible to assess the significance. Moreover, I don't know whether Trim2 further improves indel efficiency.

Thank you for these comments. We re-measured the formation of indels induced by TnpB with wild-type, Trim1, and Trim2 ω RNAs at seven target sites. Consistent with the results obtained with the four target sites tested in the original manuscript (the previous Fig. 2d), the Trim1 mutant induced indels at efficiencies comparable to or higher than those of the wild-type, and the Trim2 mutant exhibited further enhanced genome editing activity (two-way ANOVA $p < 0.0001$) (Fig. L9). We have included these results in the revised manuscript (Fig. 2c).

Fig. L9 | Indel efficiencies.

Indel formation efficiencies of TnpB with WT ω RNA, the 5' region-deleted ω RNA ($\Delta G[-231]-T[-117]$; Trim1), and the 5' region-deleted plus Stem 3b-deleted ω RNA ($\Delta G[-231]-T[-117]+G[-70]-A[-49]$; Trim2) at seven endogenous target sites in HEK293FT cells. Data are mean \pm s.d. ($n = 3$). Source data are provided as a Source data file.

6. Related to Fig. 2d, the authors claimed that Stem 3a and 3b are all disordered. Then, I am wondering why the authors did not test the Stem 3a-truncated ω RNA for indel tests with only Stem 3b truncation investigated.

Thank you for the helpful comments. Although we could not build the complete model of Stem 3a in the original model, we observed an ambiguous density corresponding to Stem 3a close to the $\alpha 1$ helix (residues 214–230) in the RuvC domain, suggesting that Stem 3a directly interacts with the TnpB protein. Therefore, we did not test the Stem 3a-truncated mutant in the original manuscript. We observed clear density for Stem 3a in the new electron density map and found that Stem 3a interacts with the $\alpha 1$ and $\alpha 2$ helices in the RuvC domain (Fig. L10a). Furthermore, we newly prepared the truncated mutant, in which Stem 3a and 3b (G[−79]–A[−37]) was deleted by connecting U(−80) and G(−36) with a GAAA linker (referred to as Trim3), and examined the indel formation efficiencies in HEK293FT cells. The Trim3 mutant showed lower indel efficiencies than the Trim2 mutant (Fig. L10b), indicating that Stem 3a is important for the TnpB-mediated DNA cleavage, which is consistent with our structural analysis.

Fig. L10 | Structural and biochemical analysis of Stem 3.

(a) Recognition of Stem 3a by TnpB. The U(−73) and C(−74) in Stem 3a form hydrogen-bonding interactions with Q227 and R231, respectively. Hydrogen bonds are shown as dashed lines.

(b) Indel formation efficiencies of TnpB with the Trim2 and Trim3 ω RNAs. At all seven target sites, the Trim3 mutant showed equivalent or lower indel efficiencies than the Trim2 mutant. Data are mean \pm s.d. ($n = 3$).

7. In relation to Fig. 2c,d, the authors argue that TnpB cleaves a full-length ω RNA into a truncated one upon binding. Then why is it that the Trim1 ω RNA is responsible for the increased indel efficiency? The full-length ω RNA would be processed into a truncated forms eventually.

Thank you for these comments. As pointed out by the reviewer, the full-length ω RNA is thought to be in the same form as the Trim1 ω RNA after processing. Indeed, our *in vitro*

cleavage assay revealed that the TnpB- ω RNA complex exhibits similar cleavage activity to the TnpB-Trim1 ω RNA complex (Fig. L7b, c). However, recent studies revealed that the truncation of the disordered region of the Cas12f sgRNA improved the genome editing efficiency in mammalian cells, probably because the shorter guide RNA is expressed more efficiently in mammalian cells^{14,15}. Consistent with these results, the Trim1 mutant induced indels at efficiencies comparable to or higher than those of the wild-type, and the Trim2 mutant exhibited further enhanced genome editing activity (Fig. L9).

8. In Fig. 3c, the mutant-cleavage data were presented as triplicated experiments, but the data points appears to include failed experiments, rather than to show a variation between experiments. The related *in vitro* data needs to be refined to clarify the involvement of the amino acids in the TAM recognition.

Thank you for the helpful comments. As the reviewer pointed out, our *in vitro* cleavage data in Fig. 3c in the original manuscript were generally less reliable. This is probably due to two issues. First, as described above, since the TnpB protein is very unstable *in vitro* and gradually aggregates during cleavage experiments, the wild-type TnpB cleaved only ~15% of the target DNA at 37°C for 30 min under the original conditions (20 mM HEPES, pH 7.5, 100 mM KCl, 2 mM MgCl₂, 1 mM DTT, and 5% glycerol). Second, as you mentioned in the Methods concern and Suggested improvement sections, we calculated the concentration of the TnpB- ω RNA complex using the A₂₆₀ value, which may not be accurate because of the total concentration of the protein-nucleic-acid complex. Therefore, we re-examined and optimised the purification and cleavage conditions, and found that TnpB can be stably purified in glycerol-containing buffer and cleaves the target DNA much more efficiently at a lower salt concentration. Furthermore, we used the Bradford method to accurately measure the concentration of the TnpB- ω RNA complex. Finally, we re-performed the *in vitro* cleavage assay in the following manner. The TnpB- ω RNA complex (2 μ L, final 250 nM) was mixed with the 3 kb linearized plasmid target containing a 16-nucleotide target sequence and a TTGAT TAM sequence (8 μ L, 100 ng), and incubated at 37°C for 30 min in 10 μ L low salt buffer (20 mM HEPES, pH 7.5, 50 mM KCl, 2 mM MgCl₂, 1 mM DTT, and 5% glycerol). The reaction products (2 μ L) were stopped by the addition of 6 μ L quench buffer, containing EDTA (20 mM final concentration) and Proteinase K (40 ng). The reaction products were resolved, visualized, and quantified with a MultiNA microchip electrophoresis device (SHIMADZU). The cleavage ratio was calculated by the molar concentration of cleaved product (1 kb) / the molar concentration of

cleaved product (1 kb) + that of un-cleaved product (3 kb). The *in vitro* cleavage experiments were performed three times. These results (shown in Fig. L11) are much more reliable and clarified our conclusion about the importance of functional residues. According to the reviewer's comments, we have added the detailed information of our *in vitro* cleavage experiment in the Methods section and these results in the revised manuscript (Fig. 4d). Furthermore, the raw cleavage data have been uploaded as Source data.

Fig. L11 | *In vitro* cleavage analysis.

In vitro DNA cleavage activities of the wild-type TnpB and TAM recognition mutants with the 5' region-deleted ω RNA. The 3 kb linearized target DNA, containing a 16-nt target sequence and a TTGAT TAM sequence, was incubated with the TnpB- ω RNA complex (250 nM) at 37°C for 30 min. The reaction products were resolved, visualized, and quantified with a MultiNA microchip electrophoresis device (SHIMADZU). Data are mean \pm s.d. ($n = 3$). The experiments were repeated three times with similar results. Source data are provided as a Source data file.

9. For the off-target experiments in Fig. 3f, several targets with the TAM-proximal mismatches should be monitored because the data were originally aimed to convince that the TAM-proximal region is essential for on-target recognition.

Thank you for this suggestion. We performed the Tagmentation-based tag integration site sequencing (TTISS) analysis¹⁶, which enables genome-wide profiling of the off-target cleavage, and found many more off-target sites with mismatches in the TAM-distal region than those with mutations in the TAM-proximal region. In the TAM-proximal region, only a single G>A mismatch seems to be tolerated, and this effect could be specific to the guide sequence tested. We then performed targeted amplicon sequencing for the top off-target sites found by the

TtISS analysis, and found that TnpB efficiently induced indels at target sites containing the TAM sequence and matching the ~12-bp guide.

10. I know TnpB cleaves NTS and TS downstream of the TAM, not upstream of the TAM.

We fixed it in the revised manuscript.

11. The structures identified in this paper provide pretty limited information related to target DNA cleavage. As shown in Fig. 3g, the NTS and TS are distantly placed from the catalytic residues and are mostly invisible due to the disordered property. The authors attempt to explain it by comparing with the structure of Cas12e, but does not rely on clear structural basis, thereby the obtainable information being marginal.

Thank you for the comments. As the reviewer pointed out, several important regions are disordered in our original electron density map. Previous studies reported that adding pentylamine (also known as amylamine) to the glow-discharged grid resulted in a radical change of the particle orientations¹. Furthermore, a recent IscB structural paper reported that the target DNA binds to the active site via a magnesium ion, in a cryo-EM analysis performed using wild-type (catalytically active) IscB and a target DNA with phosphorothioate modifications in the backbone at the cleavage site². To improve the quality of our density map and capture the structure of target DNA accessing the RuvC active site, we applied the wild-type TnpB-ωRNA-phosphorothioate-modified target DNA complex to glow-discharged grids with pentylamine, and collected the cryo-EM data set. As a result, we determined the TnpB-ωRNA-target DNA ternary structure at an overall resolution of 3.21 Å. In the new density map, we observed clear density for the TNB domain, whereas the NTS and TS remained invisible due to their intrinsic flexibility. The structure revealed that the TAM-distal region of the guide-target DNA duplex is located far from the TNB domain and exposed to the solvent, indicating that TnpB does not form any interactions with the end of the guide-target heteroduplex and the re-hybridized DNA duplex (Fig. L4b). Thus, we performed *in vitro* cleavage assays using five target DNAs with different sequences at the TAM-distal region. We found that TnpB more efficiently cleaves the target DNA with AT-rich sequences than GC-rich sequences at the TAM-distal region (Fig. L4c), suggesting that TnpB spontaneously cleaves the unwound target DNA, rather than the target DNA unwound by TnpB itself. This target DNA loading mechanism is distinct from those of the Cas12 enzymes, which recognize the PAM-distal region of the guide

RNA-target DNA heteroduplex and the re-hybridized DNA duplex by their TNB domains, thus facilitating the DNA unwinding and loading into the RuvC active site (Fig. L4a). We have added a statement about the target DNA loading mechanism by TnpB and these results in Fig. 4g, h in the revised manuscript.

Fig. L4 | Target DNA loading mechanism.

(a) Positions of active sites and target DNAs of Cas12a (Cas12a from *Francisella novicida*) (PDB ID: 6I1K) and Cas12e (Cas12e from Deltaproteobacteria, also known as CasX) (PDB ID: 6NY2). The re-hybridized DNA duplexes are recognized by the TNB domain, thereby facilitating the DNA unwinding and loading into the RuvC active site.

(b) Positions of the active site and the target DNA of TnpB. The possible trajectories of the TS and NTS are shown by a dashed arrow.

(c) *In vitro* DNA cleavage activities of TnpB with five target DNAs with different sequences at the re-hybridized DNA duplex. Re-hybridized regions with altered sequences are highlighted with a yellow background. CG and AT sequences are coloured blue and green, respectively. Data are mean \pm s.d. ($n = 3$). The experiments were repeated three times with similar results. Source data are provided as a Source data file.

MINOR POINTS:

1. In Fig. 2c, what's the 250-nt ladder in the lowest position? Is it a 25-nt band?

We fixed it in the revised manuscript.

2. The referencing is disordered throughout the manuscript. The reference number and sentences need to be properly aligned.

We fixed the citations in the revised manuscript.

3. Related to ED Fig. 5c, what's the truncation region at the 5'-region of ω RNA? Is it G[-231]-T[-117]? Then it needs to be mentioned in the figure legend.

Thank you for the helpful comments. The $\Delta 5'$ region mutant lacked G(-231)-T(-117) of the ω RNA. According to the reviewer's comments, we have added this information in the main text and figure legend in the revised manuscript.

References

1. Morris, E. P. & Da Fonseca, P. C. A. High-resolution cryo-EM proteasome structures in drug development. in *Acta Crystallographica Section D: Structural Biology* vol. 73 522–533 (International Union of Crystallography, 2017).
2. Schuler, G., Hu, C. & Ke, A. Structural basis for RNA-guided DNA cleavage by IscB- ω RNA and mechanistic comparison with Cas9. *Science* **376**, 1476–1481 (2022).
3. Zi Tan, Y. *et al.* Addressing preferred specimen orientation in single-particle cryo-EM through tilting. *Nat. Methods* **14**, 793–796 (2017).
4. Zhang, H., Li, Z., Xiao, R. & Chang, L. Mechanisms for target recognition and cleavage by the Cas12i RNA-guided endonuclease. *Nat. Struct. Mol. Biol.* **27**, 1069–1076 (2020).
5. Liu, J. J. *et al.* CasX enzymes comprise a distinct family of RNA-guided genome editors. *Nature* **566**, 218–223 (2019).
6. Hirano, S. *et al.* Structure of the OMEGA nickase IsrB in complex with ω RNA and target DNA. *Nature* **610**, 575–581 (2022).
7. Altae-Tran, H. *et al.* The widespread IS200/IS605 transposon family encodes diverse programmable RNA-guided endonucleases. *Science* **374**, 57–65 (2021).
8. Karvelis, T. *et al.* Transposon-associated TnpB is a programmable RNA-guided DNA endonuclease. *Nature* **599**, (2021).
9. Shmakov, S. *et al.* Diversity and evolution of class 2 CRISPR-Cas systems. *Nat. Rev. Microbiol.* (2017) doi:10.1038/nrmicro.2016.184.
10. Urbaitis, T. *et al.* A new family of CRISPR-type V nucleases with C-rich PAM recognition. *EMBO Rep.* (2022) doi:10.15252/embr.202255481.
11. Wu, W. Y. *et al.* The miniature CRISPR-Cas12m effector binds DNA to block transcription. *Mol. Cell* **82**, 4487–4502.e7 (2022).
12. Gomes-Filho, J. V. *et al.* Sense overlapping transcripts in IS1341-type transposase genes are functional non-coding RNAs in archaea. *RNA Biol.* **12**, 490–500 (2015).
13. Xiao, R., Li, Z., Wang, S., Han, R. & Chang, L. Structural basis for substrate recognition and cleavage by the dimerization-dependent CRISPR-Cas12f nuclease. *Nucleic Acids Res.* **49**, 4120–4128 (2021).
14. Kim, D. Y. *et al.* Efficient CRISPR editing with a hypercompact Cas12f1 and engineered guide RNAs delivered by adeno-associated virus. *Nat. Biotechnol.* **40**, 94–102 (2022).
15. Wang, Y. *et al.* Guide RNA engineering enables efficient CRISPR editing with a miniature *Syntrophomonas palmitatica* Cas12f1 nuclease. *Cell Rep.* **40**, (2022).
16. Schmid-Burgk, J. L. *et al.* Highly Parallel Profiling of Cas9 Variant Specificity. *Mol. Cell* **78**, 794–800.e8 (2020).

Reviewer Reports on the First Revision:

Referees' comments:

Referee #1:

The concerns I raised have been sufficiently addressed by the authors.

Referee #2:

The authors have performed several new experiments that address my and other reviewers' concerns. In particular, the new experiments in Fig. 4g and ED Fig. 8b provide more insight into how TnpB may cleave targets with several mismatches in the TAM-distal region. The authors also obtained a new cryo-EM reconstruction using grid conditions that reduced orientational bias. This current map appears to have more ordered regions and allowed for more unambiguous model building. Overall, I believe the authors have sufficiently addressed the concerns of reviewers and that the manuscript is suitable for publication. I do have one last minor concern that I hope the authors can address:

Lines 150-153: It is not clear how these cleavage sites are suggestive of cleavage by an RNase H-like enzyme, as opposed to any endoribonuclease. This statement needs to be updated with further rationale that explains why an RNase H-like enzyme is likely to have made these cuts, or to just state "suggesting that this RNA was cleaved by an endoribonuclease".

Referee #3:

I am satisfied with the responses by the authors and the changes made to the text.

Referee #4:

The authors strived to address most of the concerns raised previously. However, the result that a C-terminally truncated TnpB is vulnerable to stability is still confusing. The infection temperature may indicate the loss in protein stability, but the changes in binding states between ω RNA and TnpB, because T_i value, is an indicative of a transition state. Plus, the authors performed in vitro DNA cleavage assay at 37 °C, at which the status of the complex formed does not significantly differ between WT and Δ CD A. Finally, how could you explain the result that Δ CTD TnpB shows an improved DNA cleavage despite the almost the same complex status at 37 °C (Fig. L7a in the response to referees)? This issue needs to be clarified because this property is one of those supporting that TnpB is distinct from other Cas12a nucleases with respect to TnpB- ω RNA complex formation and cleavage.

Author Rebuttals to First Revision:

Responses to reviewers' comment

Reviewer #2: The authors have performed several new experiments that address my and other reviewers' concerns. In particular, the new experiments in Fig. 4g and ED Fig. 8b provide more insight into how TnpB may cleave targets with several mismatches in the TAM-distal region. The authors also obtained a new cryo-EM reconstruction using grid conditions that reduced orientational bias. This current map appears to have more ordered regions and allowed for more unambiguous model building. Overall, I believe the authors have sufficiently addressed the concerns of reviewers and that the manuscript is suitable for publication. I do have one last minor concern that I hope the authors can address:

We thank the reviewer for the positive comments.

Lines 150-153: It is not clear how these cleavage sites are suggestive of cleavage by an RNase H-like enzyme, as opposed to any endoribonuclease. This statement needs to be updated with further rationale that explains why an RNase H-like enzyme is likely to have made these cuts, or to just state "suggesting that this RNA was cleaved by an endoribonuclease".

According to the reviewer's comments, we have changed the statement "The ~160-nt RNA treated with RNase A had a pGAACp fragment (Extended Data Fig. 5d), suggesting that this RNA was cleaved between A(-150) and G(-149) or U(-138) and G(-137) by an RNase H-like enzyme, such as TnpB." to "The ~160-nt RNA treated with RNase A had a pGAACp fragment (Extended Data Fig. 5d), suggesting that this RNA was cleaved between A(-150) and G(-149) or U(-138) and G(-137) by TnpB and/or endogenous RNases." in the revised manuscript.

Reviewer #4: The authors strived to address most of the concerns raised previously. However, the result that a C-terminally truncated TnpB is vulnerable to stability is still confusing. The infection temperature may indicate the loss in protein stability, but the changes in binding states between ω RNA and TnpB, because T_i value, is an indicative of a transition state. Plus, the authors performed in vitro DNA cleavage assay at 37 °C, at which the status of the complex formed does not significantly differ between WT and Δ CDA. Finally, how could you explain the result that Δ CTD TnpB shows an improved DNA cleavage despite the almost the same complex

status at 37 °C (Fig. L7a in the response to referees)? This issue needs to be clarified because this property is one of those supporting that TnpB is distinct from other Cas12a nucleases with respect to TnpB-ωRNA complex formation and cleavage.

Thank you for the comments. Our biochemical experiments revealed that, although the ΔCTD-ωRNA complex is less stable than the wild-type TnpB-ωRNA complex, it cleaves the target DNA more efficiently than the wild-type at 37°C, at which both complexes are stable. This might be because the absence of CTD near the RuvC active site makes it easier for the target DNA to access the active site, but it is difficult to confirm this. Nonetheless, we believe that these results support our conclusion that the CTD is not functionally important for the TnpB-mediated target DNA cleavage.